**Observed trends in ground-level $O_3$ in Monterrey, Mexico during 1993-2014: Comparison with Mexico City and Guadalajara**

Iván Y. Hernández Paniagua[1,2], Kevin C. Clemitshaw[3], and Alberto Mendoza[1,*]

[1]Escuela de Ingeniería y Ciencias, Tecnologico de Monterrey, Campus Monterrey, Av. Eugenio Garza Sada 2501, Monterrey, N.L., México, 64849.
[2]Centro de Ciencias de la Atmosfera, Universidad Nacional Autónoma de México, Circuito Exterior de Ciudad Universitaria, Ciudad de México, 04510, México
[3]Department of Earth Sciences, Royal Holloway University of London, Egham, Surrey TW20 0EX, UK.
*Corresponding author: mendoza.alberto@itesm.mx

**Keywords**
Air quality, emissions inventory, odd oxygen, time series, wind-sector analysis

**Abstract**
The largest urban areas in Mexico have experienced historically high ambient $O_3$ levels. Here, we present an assessment of long-term trends in $O_3$ and odd oxygen ($O_3 + NO_2$) at the industrial Monterrey metropolitan area (MMA) in NE Mexico. High-precision and high-frequency UV-photometric measurements of ambient $O_3$ have been made since 1993 at 5 sites within the MMA. Diurnal amplitudes in $O_3$ ($AV_d$) are used as a proxy for net $O_3$ production, which is influenced by the $NO_2$ photolysis rate. No significant differences are observed in the $AV_d$ during weekdays when fossil fuel use and combustion process are higher than during weekends, although the largest $AV_d$ are observed at sites downwind of industrial areas. During weekdays, cycle troughs and peaks are typically recorded at 07:00 and 14:00 CDT, respectively, and during weekends, at 06:00 and 13:00 CDT, respectively.

The $O_3$ annual cycle is driven by changes in meteorology and photochemistry, with maximum $O_3$ mixing ratios recorded in spring and minimum values in winter. The largest annual variations in $O_3$ are typically observed downwind of the MMA, with the lowest variations generally recorded in highly populated areas and close to industrial areas. A wind sector analysis shows that, at all sites, the highest $O_3$ mixing ratios are typically recorded from the E and SE sectors, while the lowest values are recorded in air masses from the W and NW. A wind sector analysis of mixing ratios of $O_3$ precursors revealed that the dominant sources of emissions are located in the industrial regions within the MMA and the surrounding area. Significant increasing trends in $O_3$ in spring, summer and autumn are observed depending on site location, with trends in annual averages ranging between 0.19 and 0.33 ppb yr$^{-1}$. The largest annual increases in $O_3$ are for the E and SE sectors, 0.50 and 0.66 ppb yr$^{-1}$, respectively. Overall, during 1993 to 2014, within the MMA, $O_3$ has increased at an average rate of 0.22 ppb yr$^{-1}$ ($p<0.01$), which is in marked contrast with the decline of 1.15 ppb yr$^{-1}$ ($p<0.001$) observed in the Mexico City metropolitan

area (MCMA) for the same period. No clear trend is observed during 1996 to 2014 within the Guadalajara metropolitan area (GMA).

## 1.    Introduction

$O_3$ is a secondary air pollutant formed in the troposphere via the photo-oxidation of CO, methane ($CH_4$) and volatile organic compounds (VOCs) in the presence of NO and $NO_2$ (NO + $NO_2$ = $NO_X$) (Jenkin and Clemitshaw, 2000). The system of $O_3$ production is not linear, and is termed $NO_x$-limited, when $O_3$ production increases in response to increasing $NO_X$ emissions, and termed VOC-limited when it responds positively to emissions of VOCs (Monks et al., 2015; Pusede et al., 2015). Tropospheric $O_3$ is of concern to policy makers due to its adverse impacts on human health, agricultural crops and vegetation, and also due to its role as a greenhouse gas despite its relatively short lifetime of around $22.3 \pm 3.0$ days (Stevenson et al., 2006; IPCC, 2013; WHO, 2014; Lelieveld et al., 2015). As the predominant source of OH, tropospheric $O_3$ controls the lifetime of $CH_4$, CO, VOCs, among many other air pollutants (Revell et al., 2015). In polluted regions, increased levels of $O_3$ are prevalent during seasons with stable high-pressure systems and intense photochemical processing of $NO_X$ and VOCs (Dentener et al., 2005; Xu et al., 2008) with downward transport from the stratosphere of lesser importance (Wang et al., 2012). By contrast, the main removal processes for tropospheric $O_3$ are photochemical loss and dry deposition (Atkinson, 2000; Jenkin and Clemitshaw, 2000).

Tropospheric $O_3$ increased in the Northern Hemisphere (NH) during 1950-1980s due to rapid increases in precursor emissions during the industrialisation and economic growth of Europe and North America (NA) (Staehelin and Schmid, 1991; Guicherit and Roemer, 2000). Since the 1990s, reductions in $O_3$ precursor emissions in economically developed countries have resulted in decreases in tropospheric $O_3$ levels (Schultz and Rast, 2007; Butler et al., 2012; Pusede et al., 2012), however, in some regions, increases in $O_3$ have also been reported. For instance, from an analysis of $O_3$ data from 179 urban sites over France during 1999-2012, Sicard et al. (2016) reported an increasing trend in the annual averages of $0.14 \pm 0.19$ ppb $yr^{-1}$, and in the medians of $0.13 \pm 0.22$ ppb $yr^{-1}$, attributed to long-range transport and reduced $O_3$ titration by NO due to reductions in local $NO_X$ emissions. However, Sicard et al. (2016) also reported during the same period that at 61 rural sites, $O_3$ decreased in the annual averages by $0.12 \pm 0.21$ ppb $yr^{-1}$, and in the medians by $0.09 \pm 0.22$ ppb $yr^{-1}$.

In the US and Canada, $O_3$ levels have decreased substantially at different metrics during the last two decades in response to more stringent emission controls focused on on-road and industrial sources. In the Greater Area of Toronto from 2000 to 2012, $O_3$ levels decreased at urban sites by approximately 0.4 % $yr^{-1}$, and at sub-urban sites by approximately 1.1 % $yr^{-1}$, as a consequence of a reduction in the mid-day averages of $NO_2$ of 5.8 - 6.4 % $yr^{-1}$, and in the VOC reactivity of 9.3% $yr^{-1}$ (Pugliese et al., 2014). Emission estimates suggest an overall national scale decrease during 1980-2008 in US $NO_X$ and VOCs emissions of 40 % and 47 %, respectively, with city-to-city varaiablity (EPA, 2009; Xing et al., 2013).

Lefohn et al. (2010) reported that for 12 US major metropolitan areas, the $O_3$ US EPA exposure metrics of the annual 2nd highest 1-h average, and the annual 4th highest daily maximum 8-h average, decreased during 1980-2008 at 87 % and 71 % of the monitoring sites evaluated, respectively. However, Lefohn et al. (2010) observed an increase in the lower- and mid-$O_3$ mixing ratios in response to decreased titration by NO. More recently, Simon et al. (2015) assessed changes in the 1-h average $O_3$ mixing ratios at around 1400 sites across the US between 1998-2013, using the 5th, 25th, 50th 75th 95th percentiles, and the maximum daily 8-h average. Overall, Simon et al. (2015) observed increases at the lower end of the $O_3$ data distribution of 0.1-1 ppb yr$^{-1}$, mostly in urban and sub-urban areas, whereas $O_3$ decreased at the upper end of the data distribution between 1-2 ppb yr$^{-1}$ at less urbanised areas. Such changes were associated with the implementation of control strategies within the US to abate peak $O_3$ mixing ratios, as the $NO_X$ SIP Call and, tighter point and vehicle emission standards.

In Mexico, studies of long-term trends in $O_3$ have focused on the Mexico City Metropolitan Area (MCMA) (Molina and Molina, 2004; Jaimes et al., 2012; Rodriguez et al., 2016), with reports of a decrease in $O_3$ annual averages of ca. 33 % during the last two decades (Parrish et al., 2011; SEDEMA, 2016a). $O_3$ has received less consideration at other large metropolitan areas, where Mexican air quality standards are frequently exceeded (Table 1). Indeed, since 2000, recorded $O_3$ mixing ratios have exceeded Mexican official standards for $O_3$ 1-h average (110 ppb) and 8-h running average (80 ppb) by more than 50 % at the Guadalajara metropolitan area (GMA, the second most populated city) and at the Monterrey metropolitan area (MMA, the third most populated city (INE, 2011; SEMARNAT, 2015). To date, only Benítez-García et al. (2014) have addressed changes in ambient $O_3$ at the GMA and MMA during 2000-2011, reporting an increase in $O_3$ annual averages of around 47 % and 42 %, respectively. However, it should be noted that the ordinary linear regression analysis used by Benítez-García et al. (2014) may be biased by extreme values and is therefore not suitable to determine $O_3$ long-term trends with significant confidence.

To improve air quality, the Mexican government has introduced several initiatives to reduce primary pollutants emissions, with emission estimates reported in the Mexican National Emissions Inventories (NEI). The NEI suggest that from 1999 to 2008, anthropogenic $NO_X$ emissions decreased at the MCMA by 3.8 % yr$^{-1}$, but increased at the GMA and the MMA by 1.9 % yr$^{-1}$, and by 4.0 % yr$^{-1}$, respectively (Fig. S1) (SEMARNAT, 2006, 2011, 2014). These NEI $NO_X$ emission estimates agree with the decrease for the MCMA of 1.7 % yr$^{-1}$ in the $NO_2$ vertical column density during 2005-2014 reported by Duncan et al. (2016), but disagree for the GMA and the MMA where decreases of 2.7 % yr$^{-1}$ and of 0.3 % yr$^{-1}$, respectively, are reported. Similarly, Boersma et al. (2008) observed that $NO_X$ emissions over Mexico derived from $NO_2$ satellite observations were higher by a factor of 1.5 - 2.5 times than bottom-up emission estimates, which were lower by 1.6 - 1.8 times than data reported in the NEI 1999-base year. The NEI anthropogenic VOCs emissions estimates suggest a decrease at the MMA by 0.2 % yr$^{-1}$, but increases at the MCMA and at the GMA by 2.7 % yr$^{-1}$ and by 3.2 % yr$^{-1}$, respectively (Fig. S1) (SEMARNAT, 2006,

2011, 2014). However, as for NO$_X$, NEI trends in VOCs disagree with existing reports for average VOCs
decreases within the MCMA (Arriaga-colina et al., 2004; Garzón et al., 2015).
Local authorities have developed local emission inventories for the MCMA and the MMA, although only
for the MCMA the inventories have been compiled with a frequency of two years since 1996 (SEDEMA,
1999, 2001, 2003, 2004, 2006, 2008, 2010, 2012, 2014, 2016b; SDS, 2015). The accuracy of the MCMA
emission inventories has been also assessed during several field campaigns. For instance, during the
MCMA 2002-2003 campaign, Velasco et al. (2007) observed an overestimation in the 1998 inventory for
VOCs emissions of alkenes and aromatics, but an underestimation in the contribution of some alkanes.
By contrast, for the 2002 MCMA inventory, Lei et al. (2007) reported an underestimation in the VOCs
total emissions of around 65 %, based on a simulation of an O$_3$ episode occurred in 2003 within the
MCMA. Therefore, since these emission estimates are used to predict future air quality, and to design
clean air policies, it is imperative to examine the results of the policies implemented to control emissions
of O$_3$ precursors.
To our knowledge, no previous study has address trends in O$_3$ and odd oxygen in urban areas of Mexico.
In this study, we describe trends in ground-level O$_3$ within the MMA, and its response to changes in
precursor emissions during 1993-2014. Long-term and high-frequency measurements of O$_3$ were
recorded at 5 air quality monitoring stations evenly distributed within the MMA. In order to better assess
photo-chemical production of O$_3$, odd oxygen defined as ($[O_X] = [O_3] + [NO_2]$) was also considered, as
O$_3$ and NO$_2$ are rapidly interconverted. Diurnal and annual cycles of O$_3$ and O$_X$ are used to interpret net
O$_3$ production within the MMA. We show that air mass origin influences strongly the O$_3$ annual growth
rates. The trends in O$_3$, O$_X$ and precursor emissions are compared with those observed within the MCMA
and GMA. Finally, we describe that NEI emission estimates for NO$_X$ and VOCs disagree in the trend
magnitudes with ground-based NO$_X$ and VOCs measurements made at the urban areas studied here.
This paper is organised as follows: Section 2 presents the data quality and methodology used to derived
the different trends presented. Section 3 describes in detail the O$_3$ and O$_X$ diurnal and annual cycles,
and, annual and seasonally averaged trends. Section 4 discusses the origin of the O$_3$ and O$_X$ diurnal
variations and trends in the light of changes in precursor emissions. Finally, Section 5 provides some
conclusions regarding the trends observed at the studied urban areas.
**2.    Methodology**
**2.1    Monitoring of O$_3$ in the Monterrey Metropolitan Area (MMA).**
The MMA (25°40'N, 100°20'W) is located around 720 km N of Mexico City, some 230 km S of the US
border in the State of Nuevo Leon (Fig. 1a). It lies at an average altitude of 500 m above sea level (m
asl) and is surrounded by mountains to the S and W, with flat terrain to the NE (Fig. 1b). The MMA is the
largest urban area in Northern Mexico at around 4,030 km$^2$, and is the third most populous in the country
with 4.16 million inhabitants, which in 2010, comprised 88 % of the population of Nuevo Leon State
(INEGI, 2010). It is the second most important industrial area in Mexico and has the highest gross
domestic product per capita (Fig. 1c). Although the weather changes rapidly on a daily time-scale, the
climate is semi-arid with an annual average rainfall of 590 mm, and an annual average temperature of
25.0°C with hot summers and mild winters (ProAire-AMM, 2008; SMN, 2016).
Within the MMA, tropospheric $O_3$, 6 additional air pollutants (CO, NO, $NO_2$, $SO_2$, $PM_{10}$, and $PM_{2.5}$) and 7
meteorological parameters (wind speed (WS), wind direction (WD), temperature (Temp), rainfall, solar
radiation (SR), relative humidity (RH) and pressure) have been monitored continuously, with data
summarised as hourly averages, since November 1992 at 5 stations that form part of the Integral
Environmental Monitoring System (SIMA) of the Nuevo Leon State Government (Table 2; SDS, 2016).
From November 1992 to April 2003, and in accordance with EPA, EQOA-0880-047, Thermo
Environmental Inc. (TEI) model 49 UV photometric analysers were used to measure $O_3$ with stated
precision less than ±2 ppb $O_3$ and a detection limit of 2 ppb $O_3$. Similarly, in accordance with RFNA-1289-
074, TEI model 42 NO-$O_3$ chemiluminescence detectors were used to measure NO-$NO_2$-$NO_x$ with stated
precision less than ±0.5 ppb NO, and a detection limit of 0.5 ppb NO. In May 2003, replacement TEI
model 49C $O_3$ and model 42C NO-$NO_2$-$NO_x$ analysers were operated as above, with stated precision
better than ±1 ppb $O_3$ and ±0.4 ppb NO, respectively, and detection limits of 1 ppb $O_3$ and 0.4 ppb NO,
respectively. To rule out instrumentation influences on the determined air pollutants trends, long-term
trends based on annual averages were compared with those derived using 3-yr running averages, in
accordance with Parrish et al. (2011) and Akimoto et al. (2015) (Supplementary Information S1.1; Fig.
S2). Calibration, maintenance procedures and quality assurance/quality control (QA/QC) followed
protocols established in the Mexican standards NOM-036-SEMARNAT-1993 and NOM-156-
SEMARNAT-2012. The SIMA dataset has been validated by the Research Division of Air Quality of the
Secretariat of Environment and Natural Resources (SEMARNAT). The monitoring of $O_3$ and other air
pollutants at the MCMA and the GMA is detailed in the Supplementary Information S1.2-3.
**2.2 NEI data**
NEI data for estimated $NO_x$ and VOCs emissions for the 1999-, 2005- and 2008-base years were
obtained from the SEMARNAT website (http://sinea.semarnat.gob.mx). The data comprised emission
sources (mobile, point, area and natural) and air pollutants ($NO_X$, VOCs, $SO_X$, CO, $PM_{2.5}$ and $PM_{10}$), at
national, state and municipality scales. The NEI emission estimates are developed in accordance with
the Manual for the Emission Inventories Program of Mexico (Radian, 2000), which is based on the US
EPA AP-42 emission factors categorisation (EPA, 1995). The emission factors are regionalised for each
Mexican state, based upon on-site measurements and survey information. Updates to the emission
factors have been conducted for each released NEI, although no changes in the methodology were
implemented between the 1999- and 2008-base years. Overall, the mobile emissions were estimated
using the MOBILE6-Mexico model (EPA, 2003). The emissions from point sources were derived using
the annual operation reports submitted to the Environment Ministry. The emissions from area sources
were obtained using the categorisation of Mexican area sources and the regionalised AP-42 emission
factors.
The MCMA emissions inventories have been developed with a 2-year frequency since 1996, and were
obtained from the MCMA Environment Secretariat website (http://www.aire.cdmx.gob.mx/). The
methodology used to construct the MCMA inventories estimates is consistent with that used in the NEI
(SEDEMA, 2016a), which is based on the AP-42 EPA emission factors. However, more speciated
emission factors have been developed in each released version, considering updates in the local
industrial activity, survey information and field measurement campaigns. To date, the only significant
change in the methodology is the replacement of the Mobile6-Mexico model with the MOVES model to
obtain the 2014-base year mobile emissions (SEDEMA, 2016b). As for the MCMA inventories, more
speciated emission factors than those contained in the NEI were developed to produce the MMA
emissions inventory 2013-base year (SDS, 2015), although, mobile emissions estimates were obtained
with the Mobile6-Mexico model (EPA, 2003).

**2.3    Analytical methods**
SIMA, SIMAT (Atmospheric Monitoring System of the MCMA) and SIMAJ (Atmospheric Monitoring
System of the GMA) instrumentation recorded $O_3$ data every minute, which were then validated and
archived as 1-h averages. Total SIMA $O_3$ data capture by year and site are shown in Fig. S3. Data
capture averaged during 1993-2014 ranged from 82.6 % at GPE to 93.3 % at SNB, with data capture
<50 % during 1998-2000 at GPE, in 1998 at SNN, and in 1999 at OBI. A threshold of 75% data capture
was defined to consider data valid and representative (ProAire-MMA, 2008; Zellweger et al., 2009;
Wilson et al., 2012). All data were processed with hourly averages used to determine daily averages,
which were used to calculate monthly averages, from which yearly averages were obtained.

**2.4    Data analysis methods**
The SIMA, SIMAT and SIMAJ $O_3$ data sets were analysed extensively using the *openair* package v. 1.1-
4 (Carslaw and Ropkins, 2012) for R software v. 3.1.2 (R Core Team, 2013). In this study, the *openair*
functions windRose, timeVariation and TheilSen were used to analyse air pollution data. Briefly, the
windRose summarises wind speed and wind direction by a given time-scale, with proportional paddles
representing the percentage of wind occurrence from a certain angle and speed range. The timeVariation
function was used to obtain normalised daily cycles by season, and weekly cycles, with the 95 %
confidence intervals in the cycles calculated from bootstrap re-sampling, which accounts for better
estimations for non-normally distributed data (Carslaw, 2015). Finally, long-term trends of air pollutants
at the MCMA, GMA and MMA were computed with the TheilSen function, which is based on the non-
parametric Theil-Sen method (Carslaw, 2015; and references therein). The Theil-Sen estimate of the
slope is the median of all slopes calculated for a given *n* number of *x,y* pairs, while the regression

parameters, confidence intervals and statistical significance are determined through bootstrap re-sampling. It yields accurate confidence intervals despite the data distribution and heteroscedasticity, and is also resistant to outliers.

The trends computed with *openair* were contrasted with those calculated using the MAKESENS 1.0 macro (Salmi et al., 2002) as follows. Firstly, the presence of a monotonic trend was tested with the non-parametric Mann-Kendal test. For the MCMA, GMA and MMA, the available yearly data are $n>10$, hence positive values in the $Z$ parameter correspond to positive trends and vice-versa for negative values of $Z$. The significance of the estimated trend was tested at $\alpha=0.001$, 0.01, 0.05 and 0.1 using a two-tailed test. Secondly, slopes of linear trends were calculated with the non-parametric Sen's method, which assumes linear trends, with a $Q$ slope and a $B$ intercept. To calculate $Q$, first the slopes of all data values were calculated in pairs, with the Sen's estimator slope as the median of all calculated slopes. Finally, $100(1-\alpha)$ % two-sided confidence intervals about the slope estimate were obtained based on a normal distribution. Comparisons of estimated trends from both approaches are shown in the Supplementary information S1.4 (Fig. S4).

The $O_3$ and other air pollutant time-series were decomposed into trend, seasonal and residual components using the Seasonal-Trend Decomposition technique (STL; Cleveland et al., 1990). STL consists of two recursive procedures: an inner loop nested inside an outer loop, assuming measurements of $x_i$ (independent) and $y_i$ (dependent) for i = 1 to $n$. The seasonal and trend components are updated once in each pass through the inner loop; each complete run of the inner loop consists of $n_{(i)}$ such passes. Each pass of the outer loop consists of the inner loop followed by a computation of the robustness weights, which are used in the following run of the inner loop to minimise the influence of transient and aberrant behaviour on the trend and seasonal components. The initial pass of the outer loop is performed with all robustness weights equal to 1, followed by $n_{(0)}$ passes of the outer loop. The Kalman Smoother (KS) was used to provide minimum-variance, unbiased linear estimations of observations and to impute missing data to satisfy the STL (Reinsel, 1997; Durbin et al., 2012; Carslaw, 2015). Overall, statistical seasonal auto-regressive and moving averages with annual seasonal components were employed. Statistical analyses were carried out with SPSS 19.0.

In order to carry out seasonal analyses of data, seasons were defined according to temperature records in the NH, as described previously (Hernandez-Paniagua et al., 2015): winter (December-February), spring (March-May), summer (June-August) and autumn (September-November). Wind-sector analyses of data were performed by defining 8 wind sectors each of 45° starting from 0° ± 22.5°. The lower bound of each sector was established by adding 0.5° to avoid data duplicity. Data were assigned to a calm sector when wind speed was ≤ 0.36 km $h^{-1}$ (0.1 m $s^{-1}$). To assess regional transport, air mass back-trajectories (AMBT) were calculated using the HYSPLIT model v.4 (NOAA Air Resources Laboratory (ARL); Stein et al., 2015), with the Global NOAA-NCEP/NCAR reanalysis data files on a latitude-

longitude grid of 2.5°, downloaded from the NOAA ARL website (http://ready.arl.noaa.gov/HYSPLIT.php). HYSPLIT frequency plots of 96-h AMBT were constructed for every 6 h during the year 2014 with an arrival altitude of 100 m above ground level.

## 3. Results

### 3.1 Wind occurrence at the MMA

The MMA is highly influenced by anti-cyclonic easterly air masses that arrive from the Gulf of Mexico, especially during spring and summer (Fig. S5). Figure 2 shows the frequency count of 1-h averages of wind direction by site and season within the MMA during 1993-2014. At all sites, apart from OBI, the predominant wind direction is clearly E, which occurs between 35-58 % of the time depending on season. Easterly air masses are augmented by emissions from the industrial area E of the MMA, which are transported across the urban core and prevented from dispersing by the mountains located S-SW of the MMA. On average, the highest wind speeds are observed during summer at all sites. By contrast, calm winds of $\leq 0.36$ km h$^{-1}$ (0.1 m s$^{-1}$) occurred less than 2 % of the time at all sites, most frequently in winter, and least frequently in summer.

### 3.2 Time-series in $O_3$ and $O_X$ recorded within the MMA during 1993-2014

Within the MMA, the highest $O_3$ mixing ratios (1-h averages) are typically observed between April-September during the photochemical season, whereas the lowest values are usually recorded between December-January (winter) (Fig. S6). Table S1 summarises the minimum, maximum, average (mean) and median hourly $O_3$ mixing ratios recorded during 1993-2014. The highest $O_3$ mixing ratios recorded were 186 ppb at GPE in 1997, 146 ppb at SNN in 2004, and 224 ppb at SNB in 2001. At OBI and STA, the highest $O_3$ mixing ratios were both recorded on June 2, 1993: 182 ppb at 12:00 CDT at OBI, and 183 ppb at 13:00 CDT at STA, during the occurrence of E winds. Note that all times below are given in CDT. Annual $O_3$ averages varied from $14 \pm 14$ ppb at OBI in 2001 to $32 \pm 23$ ppb at SNB in 1993, whereas $O_3$ annual medians ranged from 10 ppb at OBI in 2001 to 28 ppb at SNN in 1993.

Reaction with $O_3$ rapidly converts NO to $NO_2$, and therefore mixing ratios of odd oxygen ($O_X = O_3 + NO_2$) were calculated to account for $O_3$ stored as $NO_2$ for each hour during 1993-2014 at the 5 sites within the MMA (Table S2; Fig. S7). Minimum values of $O_X$ ranged from 2 ppb, observed at all sites mostly during 1993-2014 to 13 ppb at OBI in 2007. Maximum values of $O_X$ ranged from 99 ppb at SNN in 2002, to 330 at OBI in 1993. $O_X$ annual averages varied from $23 \pm 17$ ppb at SNN in 2002 to $51 \pm 27$ ppb at OBI and at STA in 2001 and 2006, respectively, whereas $O_X$ annual medians ranged from 21 ppb at SNB and SNN, in 2001 and 2002, respectively, to 46 ppb at OBI and STA in 2001 and 2006, respectively. It is clear that the highest $O_3$ and $O_X$ mixing ratios were recorded when control of precursor emissions of VOCs and NOx were less stringent than subsequently.

## 3.2 Diurnal variations in $O_3$ and $O_X$ within the MMA

Diurnal variations in $O_3$ arise from the balance between its net production and destruction. Here, $O_3$ diurnal variations were used to assess changes in the net $O_3$ production. Figure 3 shows daily profiles by season of $O_3$, $O_X$, NO, $NO_2$, $NO_X$, and SR averaged over the 5 sites within the MMA. $O_3$ generally dips during the morning rush hour due to titration with NO and mirrors the increase in $NO_2$, which occurs around 07:00 in spring and summer, and around 08:00 in autumn and winter. The 1-h difference in the $O_3$ dip derives from the change to daylight saving time during spring and summer. $O_3$ generally peaks during the enhanced photochemical period, around 13:00 in spring, 12:00 in summer (co-incident with SR), and about 14:00 in autumn and winter. Similar profiles are observed for $O_3$ in all seasons, being negatively correlated with $NO_2$ (r=0.93 (winter) to r=0.97 (summer) ($p<0.05$)), due to the rapid photolysis of $NO_2$. Diurnal cycles of $O_X$ behave as $O_3$, with lowest values before the morning rush hour and the largest between midday (summer) and 15:00 (winter). During daytime, $O_X$ and $O_3$ diurnal cycles are strongly correlated in all seasons, ranging from r=0.97 in winter to r=0.99 in autumn ($p<0.05$), which suggests net $O_3$ production during daytime.

$O_3$ and $O_X$ levels depend strongly on the photochemical processing of $NO_X$ and VOCs emissions. To assess differences in the net $O_3$ production from site-to-site within the MMA, $O_3$ and $O_X$ amplitude values ($AV_d$) derived from normalised daily cycles were used as proxy. The normalised daily cycles were constructed by subtracting daily averages from hourly averages. Figure 4 shows normalised $O_3$ daily cycles and Fig. S8 normalised $O_X$ daily cycles. The lowest $AV_d$s both in $O_X$ and $O_3$ occur in winter consistent with reduced SR and low photolysis rates, while the largest ones are seen in summer. It is clear that during the whole year, the largest $AV_d$s are recorded at sites downwind of the industrial emission sources, in particular at STA, while the lowest $AV_d$s are observed at upwind sites. The larger $AV_d$s at downwind sites indicate higher net $O_3$ production, derived from photochemical processing of air masses from the E sector. The $AV_d$s seen at upwind sites indicate that these are less affected by emissions from the largest part of the MMA and from the industrial area.

## 3.3. Annual cycles of $O_3$ and $O_X$ within the MMA

Annual variations in $O_3$ and $O_x$ are correlated positively with the seasonality of temperature, RH and SR (Camalier et al., 2007; Zheng et al., 2007). Annual averages cycle for those meteorological variables, $O_3$ and $O_X$ were constructed by averaging monthly averages for the same month during the studied period. Figure 5a shows that $O_3$ exhibits the maxima during spring and minima in winter, with a downward peak in early autumn, behaviour characteristic of tropospheric $O_3$ in the NH. $O_X$ peaks in spring and dips in summer, although it is evident that $NO_X$ emissions lead to apparently similar $O_X$ levels in winter and spring despite the decrease in $O_3$ levels. A correlation analysis among monthly averages for both $O_3$ and $O_X$ with temperature, rainfall, RH and SR, revealed that the strongest relationship was between $O_3$ and SR (r= 0.72, $p<0.001$; Fig. 5a), with relationship evident with $O_X$.

Seasonal amplitude values ($AV_s$) provide insight into inter-annual variations in the net $O_3$ production in response to changes in precursor emissions and meteorology. The seasonal cycles in $O_3$ during 1993-2014 were determined by filtering monthly averages with the STL technique (Cleveland et al., 1990) (Fig. S9). $O_3$ $AV_s$s were calculated as the difference peak-to-trough (spring peak). An average $O_3$ $AV_s$ of 15.1 ± 2.97 (1$\sigma$) ppb was calculated from 1993 to 2014 within the MMA, with the lowest $O_3$ $AV_s$ of 10.3 ppb determined in 1998, and the largest $O_3$ $AV_s$ of 19.0 ppb observed in 2014. Figure 5b shows that $O_3$ $AV_s$ decreased significantly at all sites between 1993 and 1997-1998, at rates from 0.78 ppb $O_3$ $yr^{-1}$ at GPE to 2.28 ppb $O_3$ $yr^{-1}$ at SNN (Fig. 5c). $O_3$ $AV_s$s increased constantly ($p<0.05$) at all sites since 1998, ranging from 0.90 ppb $O_3$ $yr^{-1}$ at GPE to 0.75 ppb $O_3$ $yr^{-1}$ at SNN. $O_X$ $AV_s$s exhibited no discernible trends at all sites for the whole studied period, although, SNN show a significant ($p<0.05$) decline during 1993-2001 (1.5 ppb $yr^{-1}$) and at STA show an increase during 2004-2010 (1.3 ppb $yr^{-1}$). The trends in $O_X$ follow those observed for $NO_X$ at SNN and STA during 1993-2014, which indicates that nearby industrial emissions have a significant contribution on the observed $O_X$ levels within the MMA.

### 3.4. Long-term trends in $O_3$ and $O_X$ within the MMA during 1993-2014

Quantifying the absolute changes in ground-level $O_3$ in response to trends in its precursor emissions is crucial to evaluate the impacts of air quality control (Parrish et al., 2009; Simon et al., 2015). The growing economy within the MMA has increased $O_3$ precursor emissions from point and area sources, due to the limited emissions control programs (INEGI, 2015; SDS, 2015). Moreover, predominant E-SE winds throughout the year transports primary pollutants and their oxidised products downwind from the industrial area, which can offset reductions in emissions from other sources. Here, to characterise changes in net $O_3$ production during 1993-2014 within the MMA in response to changes in its precursor emissions, long-term trends for daytime (06:00-18:00 CDT) $O_3$ and $O_x$ measurements were derived by averaging data in seasonal periods. Seasonal averaging was used to minimise variability inherent in longer-term averages and the de-seasonalisation process avoids confounding overall trends, especially when seasons exhibit opposite trends. (Parrish et al., 2009).

Figure 6 shows seasonal trends in $O_3$ within the MMA, and Table 3 summarises the parameterisation of the trends. Significant increases ($p<0.1$) in $O_3$ are observed at all sites, apart from STA, in spring and summer, while in autumn, $O_3$ increases significantly only at SNN and SNB. The increases in $O_3$ range from 0.26 ppb $yr^{-1}$ in spring at OBI to 0.47 ppb $yr^{-1}$ in summer at SNN. Overall, the lowest $O_3$ growth rates are observed at the urban background GPE site, whereas the largest ones are at the industrial SNN site. It is worth nothing that only SNN and OBI exhibit significant increases in autumn, despite a decrease in the frequency of high wind speeds (>20 km $h^{-1}$). The existence of significant trends at all sites during spring-summer, except for OBI, is consistent with the downwind transport of industrial emissions and the high frequency of photochemical processed air masses with NE-S-SE origin, where the industrial area is located (Fig. S10).

Seasonal trends in $O_X$ are shown in Fig. 7, with the parameters of the trends listed in Table 3. Consistent
with the seasonal $O_3$ trends observed, significant increases ($p<0.1$) in $O_X$ within the MMA are determined
in spring at all sites except for STA, and range from 0.02 ppb $yr^{-1}$ at OBI to 0.67 ppb $yr^{-1}$ at SNB. It is
worth nothing that the industrial SNN and SNB sites show significant increases in $O_X$ in all seasons, with
the lowest growth rates in winter and the largest in summer and spring, respectively. Moreover, STA
exhibits the only significant decrease in $O_X$ of 0.63 ppb $yr^{-1}$ during winter. As for $O_3$, the $O_X$ increasing
trends are consistent with the transport of primary emissions during the high occurrence of NE-E-SE air
masses at WS >10 km $h^{-1}$, which is highlighted during the photochemical season (April-September).
Furthermore, the small shift in wind direction at STA to NW during winter coincides with the only observed
decrease in net $O_3$ production within the MMA, which confirms that $O_3$ precursors are emitted E of the
MMA. This also makes evident that increasing upwind industrial emissions have offset reductions in
emissions from on-road sources as revealed by the decline in $NO_X$ evident at OBI.

**3.5 Comparison of MMA $O_3$ and $O_X$ weekly profiles with those at MCMA and GMA**
$O_3$ production varies from city-to-city in response to local $NO_x$ and VOCs emissions. Assessment of
weekly profiles of $O_3$ and $O_X$ may provide insights of the geographic response in net $O_3$ production to
diurnal variations in precursor emissions. Hourly $O_3$ and $O_X$ averages were used to construct weekday
and weekend average profiles for the MCMA from 1993 to 2014, and for the GMA from 1996 to 2014.
Figure 8 compares weekly $O_3$ and $O_x$ profiles by season within the MMA with those for the MCMA and
GMA. In each case, and consistent with observations in other major urban areas of NA, the lowest $O_3$
mixing ratios occur during the morning rush hour due to $O_3$ tritation with NO emitted from on-road
sources, whereas peak values of $O_3$ are apparent after mid-day during periods of enhanced SR
(Stephens et al., 2008; Jaimes-Palomera et al., 2016). It should be noted that the peak value of $O_3$ for
the GMA in winter and spring occurs an hour or so earlier than for the MMA and MCMA, which is
consistent with higher VOC/$NO_x$ emissions ratios at the GMA (Kanda et al., 2016). As might be
anticipated, larger $AV_d$ of 76.9 ± 1.6 ppb $O_3$ are observed for the MCMA than for the GMA (46.1 ± 1.0
ppb $O_3$) and MMA (37.6 ± 0.4 ppb $O_3$), related to the levels of emissions of the $O_3$ precursors. The $O_x$
profiles show a trough during the morning rush hour and a peak between 12:00 and 14:00 at all urban
areas. Despite large variations between weekday and weekend $NO_x$ mixing ratios at the 3 urban areas
as shown in Fig. 8, no significant differences ($p>0.05$) in $O_3$ and $O_X$ are observed at any of the
metropolitan areas between $O_3$ and $O_x$ weekends and weekdays $AV_d$s.

Stephens et al. (2008) suggested that the most plausible explanation for the lack of weekend $O_3$ effect
at MCMA during 1987-2007, is that weekday $O_3$ production is limited by VOCs and inhibited by $NO_X$.
Therefore, the very similar levels $O_3$ observed during weekdays and weekends can be explained by
simultaneous decreases in $NO_X$ and VOCs emissions and the resulting effects on net $O_3$ production.
Similarly, a VOC-limited $O_3$ production regime was reported for the MMA by Sierra et al. (2013), whereas
Kanda et al. (2016) reported that at the GMA the $O_3$ production lies in the region between VOC- and
NO$_X$-sensitivity. Therefore, it can be hypothesised that simultaneous decreases in emissions of NO$_X$ and
VOCs during weekends at the GMA and MMA explain the similarity in behaviour in O$_3$ and O$_X$ as at the
MCMA. Indeed, Wolff et al. (2013) reported that at several urban areas of the US, similar or even higher
(±5 %) O$_3$ levels during weekdays than at weekends were due to lower O$_3$ precursor emissions over
weekends. Furthermore, the number of sites in the US that exhibited a weekend effect decreased from
ca. 35 % to less than 5 % from 1997-1999 to 2008-2010, which was attributed to an increase in the
VOC/NO$_X$ emission ratio derived from a greater decline in NO$_X$ than in VOCs emissions, mostly driven
by reductions from on-road sources. A change to a NO$_X$-limited O$_3$ production regime during weekends
at the three urban areas seems unlikely, since this would result in lower O$_3$ levels during weekends,
which is not observed at any of the studied urban areas (Torres-Jardon et al., 2009).

**3.6 Long-term trends at MCMA, GMA and MMA from 1993 to 2014**
The high mixing ratios of O$_3$ observed typically at the 3 largest urban areas in Mexico have motivated
the introduction of control strategies to decrease emissions of the O$_3$ precursors, NOx and VOCs. The
success of the control strategies implemented can be evaluated by assessing trends in O$_3$ and O$_X$. As
for the MMA, seasonal trends in O$_3$ and O$_X$ within the MCMA and GMA were calculated from daytime
measurements. Figure 9 shows a comparison of inter-annual trends in O$_3$ and O$_X$ at the 3 urban areas
in Mexico, and Table 4 lists the parameters of the trends. Overall, during 1993-2014, daytime O$_3$ at the
MCMA decreased significantly ($p$<0.05) by 1.15 ppb yr$^{-1}$ (2.04 % yr$^{-1}$), and increased at the MMA by 0.22
ppb yr$^{-1}$ (0.84 % yr$^{-1}$); at the GMA no discernible trend was observed during 1996-2014. For daytime O$_X$
at the MCMA and GMA during the same periods, significant decreases ($p$<0.05) of 1.87 and 1.46 ppb yr$^{-}$
$^{1}$ were determined, respectively, while the MMA does not exhibit a significant change. At the MCMA, the
overall trends in O$_3$ and O$_X$ are strongly driven by their wintertime decreases of 1.62 and 2.47 ppb yr$^{-1}$,
respectively; whereas at the MMA, the annual growth in O$_3$ is driven by increases in spring and summer
of 0.32 and 0.27 ppb yr$^{-1}$, respectively. Although, at the MMA, an increase in O$_X$ of 0.28 ppb yr$^{-1}$ is
observed only during summer, the overall O$_X$ trend is strongly affected by the non-significant trends in
the other seasons. It is worth nothing that at the GMA, the overall decrease in O$_X$ of 1.46 ppb yr$^{-1}$ is
similar for all seasons, which range between 1.40 ppb yr$^{-1}$ (autumn) and 1.89 ppb yr$^{-1}$ (spring).

The overall trends in net O$_3$ production during 1993-2014 at the MCMA and GMA are consistent with the
significant ($p$<0.05) annual decreases in NO$_X$ of 1.21 and 1.25 ppb yr$^{-1}$, respectively (Fig. 10). By contrast,
while average NO$_X$ levels have increased annually at the MMA at 0.33 ppb yr$^{-1}$ ($p$<0.05), the average net
O$_3$ production has remain steady. Either the non-linear response in O$_X$ to the changes in NO$_X$ in an
environment of high NO$_X$ mixing ratios (>60 ppb) displace the chemical equilibrium to favour NO as the
dominant component of NO$_X$ which does not account for the levels of O$_X$ (Clapp and Jenkin, 2001). Or
the O$_X$ trends derived from the combined data set for the MMA do not represent local observed trends,
because a compensating effect between O$_X$ reductions and increases.

**3.7 Compliance with the 1-h and 8-h Mexican Standards for $O_3$ within the MMA**

Between 1993 and 2014, there were two official standards for maximum permitted mixing ratios of $O_3$ in Mexico: i) a running 8-h average of 80 ppb, not to be exceeded more than 4 times per calendar year, and ii) a 1-h average of 110 ppb (NOM-020-SSA1-1993). Since 19 Oct 2014, the maximum permitted $O_3$ levels were lowered to a running 8-h average of 70 ppb and a 1-h average of 95 ppb, (NOM-020-SSA1-2014). However, because both standards are applicable for whole calendar years, the old permitted $O_3$ levels were used in this study to determine the number of annual exceedances to both $O_3$ standards. Figure 11 shows that within the MMA, the $O_3$ 1-h average and the running 8-h standards were frequently exceeded (INE, 2011; SEMARNAT, 2015). The largest number of exceedances occurs at STA, followed by SNB, GPE and OBI, whereas the fewest breaches are observed at SNN markedly since 2004. However, there have been 3 periods of clear decreased exceedances at all sites (except STA in 2014), during 1994-1995, 1999-2000, and 2012-2013, which are consistent with marked changes in the national GDP during economic recessions in Mexico (Fig. S11a). However, although, national GDP exhibits a notable decrease during the 2008-2009 global economic recession, only in 2009 do the $O_3$ annual exceedances within the MMA seem to follow (Fig. S11b).

Therefore, if $O_3$ levels continue to increase within the MMA, as determined in the long-term trend assessment, an increase also in peak $O_3$ mixing ratios is likely to occur. Hence, to analyse changes in peak $O_3$, daily maxima 1-h averages from 1993 to 2014 were used to determine seasonal trends in peak levels. Figure 12 shows trends in 1-h daily maxima and Table 5 list the parameters of the trends. Daily maxima $O_3$ 1-h averages have increased significantly ($p<0.05$) in spring and summer at all sites, except for STA, and also in autumn at the industrial sites SNN and SNB. The largest increases in the daily maxima are seen at SNN, where similar increases between 0.85 and 0.93 ppb $yr^{-1}$ are determined between spring and autumn. SNB exhibits slightly lower growth rates in spring and summer, but a large difference in autumn. We have shown that predominantly E-SE winds transport photochemically processed air mases to SNN and SNB during spring-summer leading to the observed exceedances. Moreover, the change in the wind occurrence in autumn at SNB leads to a lower growth rate than at SNN, where the calmest winds during the whole year drive the largest increase interpreted to be due to the photochemical processing of precursors emitted locally. The GPE and OBI sites exhibit increases only in spring and summer, with the lowest increases of all sites determined at OBI of 0.48 ppb $yr^{-1}$ in spring, which contrasts with the largest increase at OBI during the same season. However, such increases are consistent with an increase in the occurrence of NE and E air masses at high speeds (>10 km $h^{-1}$) during spring-summer. STA shows a significant decrease in the maxima daily $O_3$ 1-h averages of 0.35 ppb $yr^{-1}$ in winter, which is consistent with an increase in the occurrence of NW air masses at WS < 5 km $h^{-1}$, loaded with high $NO_x$ mixing ratios (50 ppb) that promote the $O_3$ titration.

**4. Discussion**

**4.1 Strategies for air quality control in Mexico**

The Mexican environmental authorities have focused largely on improving the air quality within the MCMA since 1986, by implementing numerous strategies to control primary emissions, but have paid less attention to other large metropolitan areas in Mexico (PICCA, 1990; ProAire-MCMA, 2011). Control measures have been designed based on NAEI and local emission inventories data, which possess significant uncertainties (Arriaga-Colina et al., 2004; Velasco et al., 2007; Kanda et al., 2016). However, despite these uncertainties, the emission control strategies have helped to reduce $O_3$ levels within the MCMA since 1991-1992 (ProAire-MCMA, 2001). Here, we describe the most effective measures introduced to control $O_3$ precursor emissions within the MCMA, and then discuss potential benefits of implementing such measures within the MMA.

From 1993 to 2014, $NO_X$ levels within the MCMA decreased at a rate of around 1.2 ppb $yr^{-1}$ (1.6 % $yr^{-1}$) as determined from ground-based measurements. This decline is remarkably consistent with the decrease during 2005-2014 in the $NO_2$ column over the MCMA of 1.6 % $yr^{-1}$ reported by Duncan et al. (2016). The decrease in $NO_X$ has been driven largely by reductions in emissions from on-road sources, in response to the introduction of mandatory 3-way catalytic converters in new vehicles since 1993 (NOM-042; SEMARNAT, 1993), and by the introduction of a no driving day and more stringent exhaust emissions inspection programs for private cars since 1989 (NOM-041; SEMARNAT, 1993). The $NO_X$ reduction measures also required public transport vehicles to switch from petrol to LP gas fuelled engines, new road corridors were designed for improving the intracity transport and the public transport fleet was renewed (ProAire-MCMA, 2001). For industrial sources, the switch from fuel oil to LP gas fuel, relocation of highly polluting industries away from the MCMA, and implementation of regular inspections programs of $NO_X$ emission for industrial and area sources were also implemented (ProAire-MCMA, 2001).

While the outlook for $NO_X$ levels within the MCMA is clear, studies of VOCs levels have reported no concluding trends. For instance, Arriaga-Colina et al. (2004) reported a decrease in VOCs of around 10 % from 1992 to 2001 over the N MCMA, while Garzón et al. (2015) reported that on average VOCs increased over most of the MCMA between 1992-2002 but decreased by 2.4 ppb $yr^{-1}$ between 2002-2012. However, the decrease in VOCs from 2002 to 2012 reported by Garzón et al. (2015) is consistent with a reduction in light alkanes and aromatics levels during the morning rush hour reported by Jaimes-Palomera et al. (2016). Continuous measurements of VOCs have been introduced recently by the MCMA government, which precludes an assessment of VOCs long-term trends. The measures implemented to control VOCs emissions from on-road sources have included the reformulation of petrol with the reduction of highly reactive VOCs and addition of oxygenated compounds, and fitting of 3-way catalytic converter in all new vehicles (NOM-042; SEMARNAT, 1993; ProAire-MCMA, 2001). For area sources, control measures include the introduction of vapour emissions control systems at petrol stations and introduction of a LP gas leak detection program for the distribution network (ProAire-MCMA, 2011). As

for NO$_X$, industrial VOCs emission sources have been subject to regular emissions inspections and
relocation of the most significant emitters (ProAire-MCMA, 2011).

Therefore, the moderate success on controlling O$_3$ levels within the MMA can be interpreted as the
implementation of effective controls measures on VOCs and NO$_X$ emissions. Thus, a comparison
between VOCs and NO$_X$ trends derived from the NAEI and local emissions inventories with those
determined from ground-levels measurements can provide insight into further improvements in
decreasing O$_3$ levels not only within the MCMA but also at other large metropolitan areas in Mexico.
Within the MCMA, the NAEI NO$_X$ emissions trends are consistent with the decrease determined from
ground-based measurements made by SIMAT, but the MCMA local inventory trends disagree with the
SIMAT trends (Fig. S1 and Fig. 10). For VOCs, the NAEI and the MCMA inventories oppose measured
trends in VOCs during 1993-2001 (Arriaga-Colina et al., 2004; Garzón et al., 2015). This can be
explained by underestimates of VOC emissions within the MCMA of a factor of 2-3 (Arriaga-Colina et al.,
2004; Velasco et al., 2007). Such discrepancies suggest that, significant improvements in NO$_X$ and VOCs
emissions inventories are still required to better inform O$_3$ control strategies.

**4.2 Ground-level O$_3$ and O$_X$ variations within the MMA**
The O$_3$ and O$_X$ diurnal variations result from the particular chemical environment and meteorological
conditions at each monitoring site within the MMA. Thus, the largest O$_3$ and O$_X$ mixing ratios, except for
OBI, are observed typically for air masses from the E and SE wind sectors, whereas at OBI, the largest
O$_3$ and O$_X$ values are recorded during the occurrence of NE and E air masses. It is clear that short-range
transport and large upwind emissions of O$_3$ precursors from the industrial area dominate the MMA
(SEMARNAT, 2006, 2011, 2014; SDS, 2015). This is underlined at OBI with the highest values of O$_X$
where the predominant wind direction is NE, consistent with the transport of emissions from the industrial
area located NE, and photochemical processing of air masses (Carrillo et al., 2017). The daily cycles of
O$_3$ determined within the MMA are consistent with those reported for Los Angeles (VanCuren, 2015),
and Toronto (Pugliese et al., 2014). At Toronto, the O$_3$ maxima were enhanced by the arrival of
photochemical processed air masses transported from polluted wind sectors, and decreased during clear
air masses. This behaviour is similar to that observed within the MCMA with enhanced O$_3$ maxima during
the occurrence of E-SE (polluted) and decreased levels when SW-W (relatively clean) air masses
occurred.

**4.3. Origin of the O$_3$ annual cycles within the MMA**
The O$_3$ annual cycles within the MCMA are consistent with the spring maxima and winter minima
characteristic of the US southeast regions (Strode et al., 2015), and follow the O$_3$ cyclic pattern at NH
mid-latitudes (Monks 2000; Vingarzan, 2004). However, they are different to O$_3$ annual cycles reported
for the US west coast regions, particularly in California, where the maxima in the cycle occurs between
June-August, driven the local influence of precursor emissions upon O$_3$ production and photochemical

conditions (Vingarzan, 2004; Strode et al., 2015). The recurrent downward spikes in the $O_3$ annual cycles within the MMA between July-August result from high wind speeds (>10 km h$^{-1}$ on average) that disperse $O_3$ precursors and increase the boundary layer height (ProAire-MMA, 2008). The peak in $O_3$ observed in September is characteristic of humid regions, and can be ascribed to an increase in OH radicals derived from the increment in RH during the rainy season (Lee et al., 2014). A marked increase in RH within the MMA during September is consistent with the increase in $O_3$ observed as reported by Lee et al. (2014). Over the mid-western and eastern US regions, that $O_3$ peak has become less noticeable since 2000 (Zheng et al., 2007).

The annual variability in $O_3$ within the MMA is strongly coupled to the economic conditions (GDP) in Mexico. For instance, the economic crisis of 1994-1996 caused a marked reduction in industrial emissions of VOCs and $NO_X$, which is confirmed by the significantly decrease in $O_3$ annual variations at all sites within the MMA (Tiwari et al., 2014; INEGI, 2016). During the global economic recession of 2008-2009, Castellanos and Boersma (2012) reported a reduction of 10-30 % in tropospheric $NO_2$ over large European urban areas, which is consistent with a faster decline of $8 \pm 5$ % yr$^{-1}$ in the $NO_2$ column density during the same period for US urban regions (Russell et al., 2012). Increases in the $NO_2$ column density over the MMA as reported by Duncan et al. (2016) are explained by the gradual recovery of the economy since 1997 in Mexico. Moreover, increases in $O_3$ precursor emissions and in annual variability observed within the MMA are consistent with such economic growth. This explains clearly the opposite trends in $O_3$ annual variations before and after the economic crisis within the MMA, with the lowest changes seen at the urban GPE site and the greatest ones detected for the SNN industrial site.

**4.4 Increasing $O_3$ and $O_X$ levels within the MMA**

Ground-based measurements made during 1993-2014 reveal significant ($p<0.05$) increases in $NO_X$ within the MMA at all sites, apart from OBI, which exhibits a significant decrease (Fig. 13). Overall, the $NO_X$ increase within the MMA of 1.24 % yr$^{-1}$ (0.33 ppb yr$^{-1}$) during 1993-2014 is larger than the increase in the $NO_2$ column density over the MMA of around 0.78 % yr$^{-1}$ during 2005-2014 reported by Duncan et al. (2016), although both indicate a significant increase in the $NO_X$ levels at least since 2005. The largest increases in $NO_X$ correspond to industrial sites, SNN (0.51 ppb yr$^{-1}$) and SNB (0.74 ppb yr$^{-1}$), which is interpreted as a response to growing industrial activity, in combination with flexible emission regulations within the MMA (INEGI, 2016). The influence of industrial emissions upon $O_3$ at the MMA becomes evident by the lowest $NO_X$ growth rate observed at GPE of 0.19 ppb yr$^{-1}$, since OBI has few occurrences of air masses transporting pollutants from the largely industrialised areas throughout the year (Fig. 2). By contrast, the $NO_X$ decrease at OBI of -0.40 ppb yr$^{-1}$ arises from decreases in emissions from on-road sources (SDS, 2015). The large growth rates in $O_3$ and $NO_X$ at SNN and SNB are explained by increasing emissions of $O_3$ precursors from a growing number of industries and the urban development E of the MMA. The most likely explanation for the $O_3$ increase at OBI is a reduced titration effect by decreasing

$NO_X$ levels in combination with the non-linear response in $O_3$ production to decreasing $NO_X$ emissions
under the VOC-sensitive MMA airshed (Sierra et al., 2013; Menchaca-Torre et al. 2015).

The $O_3$ increasing trends within the MMA are opposite to those reported by Sather and Cavender (2016)
at 4 South Central US urban areas, where $NO_X$ and VOCs decreased by 31-70 % and 43-72 % during
1983-2015, respectively, resulting in a reduction between 18-37 ppb $O_3$ in the 8-h averages. The $O_3$
response to $NO_X$ decreases at OBI is similar to that observed in central London during 1996-2008 (Bigi
and Harrison, (2010), and at four urban areas in Japan during 1990-2010, explained by the decrease of
the NO titration effect (Akimoto et al., 2015). This suggests that controlling VOCs emissions may lead to
a decrease in the net $O_3$ production, whereas decreases in $NO_X$ may not have significant effects on $O_3$
production or even increase the $O_3$ levels due existence of a VOC-limited environment within the MMA
(Sierra et al., 2013, Carrillo et al., 2017).

The $O_X$ long-term trends during 1993-2014 within the MMA were consistent with those for $O_3$ at all sites.
Decreases in $NO_X$ and $O_3$ observed between 1994-1996 were the response to the economic crisis during
the same period in Mexico, when the DGP decreased by 5.9 % providing additional evidence of the
dominant role of industries within the MMA. Consistent with economic indicators, annual averaged petrol
sales in the Nuevo Leon state in 1995 decreased by 2.4 % in relation to 1994, but increased linearly from
1996 to 2008 at an approximate rate of 98,800 $m^3$ petrol $yr^{-1}$ (r = 0.90) (Fig. S12) (SENER, 2015). As for
petrol sales, registered vehicles in Nuevo Leon show significant variations between 1993-1996, but
increase linearly since 1997 at a rate of around 100,000 vehicles $yr^{-1}$ (r=0.99). This confirms that despite
the annual growth in the vehicular fleet, the fitting of 3-way catalyst technology and reformulation of petrol
introduced in 1997 has controlled on-road primary emissions (ProAire-MCMA, 2001) The decreases in
$NO_X$ observed at OBI and at all sites during the occurrence of SW-W-NW air masses reflect that if
applied, stricter emissions controls such as those for on-road sources can lead to a significant abatement
in primary emissions. It is clear that the industrial sources must be subject to similar emission control
measures as those implemented within the MMA for effectively reducing the $O_3$ levels.

**4.5 The opposite $O_3$ trends at Mexican urban areas**
The comparison of $O_3$ and $O_X$ trends at MMA, GMA and MCMA reveals different emission trends at each
of the studied cities. The trends in $O_3$ reported in this study for the MCMA, agree with the reduction of
20 ppb $O_3$ during 1991-2011 for the MCMA (Jaimes et al., 2012), and with the reduction of 8 ppb $O_3$
during 2000-2011 for the MMA (Benítez-García et al., 2014). At the GMA, the no trend status in $O_3$
determined here is in contrast with the increase of 12 ppb $O_3$ during 2000-2011 (Benítez-García et al.,
2014), which is due to the different periods assessed in the latter. Decreases in $O_3$ in US urban areas
arise from effective control of $O_3$ precursor emissions (Strode et al., 2015), which has occurred at the
MCMA. By contrast, $O_3$ levels increased in urban areas of Japan by 0.22-0.37 ppb $yr^{-1}$ (Akimoto et al.,
2015), and in the Greater London by 0.5 ppb $yr^{-1}$ (Bigi and Harrison, 2010), due to faster declines in $NO_X$
than in VOCs and are slightly similar that observed in $O_3$ averages for the MMA (0.20 ppb $yr^{-1}$), where
average $NO_X$ levels have increased and also likely VOCs.

Trends in net $O_3$ production can be interpreted as the response to trends in its precursors, which also
respond to implemented policies to control their emissions and to economic factors. Figure 10 shows
that $NO_X$ decreased significantly within the MCMA (1.57 % $yr^{-1}$) and the GMA (1.83 % $yr^{-1}$) during 1993-
2014 and 1996-2014, respectively, but increased within the MMA (1.83 % $yr^{-1}$) during 1993-2014. Such
$NO_X$ trends are within the range of the trends in the $NO_2$ column density reported by Duncan et al. (2016)
in Table S9, which reveals an increase of 0.78 ± 1.12 % $yr^{-1}$ for the MMA, but decreases of 1.82 ± 0.84
% $yr^{-1}$ for the GMA and of 0.10 ± 1.67 % $yr^{-1}$ for the MCMA, all during 2005-2014. To date, long-term
trends in VOCs have only been reported only the MCMA with an average decrease of ca. 2.4 ppb $yr^{-1}$
since 2002, mostly in propane, ethanol and acetone (Garzón et al., 2016), while there are no studies of
long-term trends in VOCs within the MMA and the GMA.

It is clear that $O_3$ and $O_X$ decreases within the MCMA have been driven by reductions in $NO_X$ and VOCs
emissions, and that the implemented strategies described in Sect. 4.1 have proved to be effective in
controlling primary emissions. By contrast, growing industrial emissions within the MMA must be subject
to stringent controls to abate $O_3$ levels. In the GMA, where the industrial activity is lower that at the
MCMA and MMA (Kanda et al., 2016), the policies introduced at national scale for controlling on-road
sources emissions have resulted in the decrease of $NO_X$ emissions and in the stabilisation of $O_3$ levels.
Finally, the results presented here demonstrate the merits of the assessment and analysis of long-term
$O_3$ levels, which can be used by environmental authorities to revise and to redesign programs and
policies to improve air quality. Continuing with ground-based $O_3$ and $NO_X$ monitoring is strongly
recommended to better understand the response further changes in local and regional $O_3$ levels to
changes in primary emissions. Monitoring of VOCs at the GMA and MMA is also recommended to as
the VOCs emissions data reported in the NAEI possess significant uncertainties.

**5. Conclusions**
Diurnal and annual cycles, and long-term trends in $O_3$ and $O_X$ within the MMA, are interpreted as
response to changes in $NO_X$ and VOCs emissions, photochemistry and meteorology. Continuous high-
frequency and high-precision $O_3$ and $NO_X$ data recorded during 1993-2014 at 5 sites within the MMA
and at 29 sites within the MCMA, and during 1996-2014 at 10 sites within the GMA, were used to
calculate long-term trends. Within the MMA, the greatest mixing ratios in $O_3$ were recorded during E and
SE winds, at sites downwind of significant precursors from industrial sources. By contrast, the lowest $O_3$
mixing ratios were recorded at SNN, and for all sites were observed for the W and SW sectors, where
air masses travel from central Mexico over 100-300 km of semi-arid region sparsely populated. Maximum
daily 1-h values of $O_3$ and $O_X$ increased significantly at GPE, SNN and SNB, owing to increasing
emissions of precursors, while at OBI increasing $O_3$ and decreasing $O_X$ trends arise from the non-linear
response to decreasing $NO_X$ emissions from on-road sources.

Annual cycles in $O_3$ at all sites peak in spring and through in winter, with a downward spike during
summer caused by high winds that disperse $O_3$, and increase the boundary layer height. Decreases in
$O_3$ precursor emissions during the economic crisis experienced in Mexico between 1994-1996, caused
significant decline trends $O_3$ annual variations from 1993 to 1997 or 1998, depending on site, followed
by significant increases derived from the recovery of the economy. The dominant role of industrial
sources on $O_3$ precursor levels within the MMA was evident at the industrial site SNN during the 1994-
1996 economic crisis.

At all metropolitan areas studied, $O_3$ and $O_X$ levels showed no significant differences between weekdays
and weekend, although an earlier occurrence of the $O_3$ peak at the GMA was detected, ascribed to larger
VOCs/$NO_X$ emission ratio. The lack of the weekend effect was attributed to weekday $O_3$ production being
limited by VOCs, whereas increases in the VOC/$NO_x$ ratio during weekends in response to reduced
emissions from mobile sources resulted in similar $O_3$ mixing ratios that during weekdays. Larger $AV_d$s
during weekdays and weekends were seen at MCMA than at GMA and MMA related to the relative
emissions of the $O_3$ precursors.

Significant seasonal trends in $O_3$ and $O_X$ during spring were observed at all sites, apart from STA,
whereas industrial sites exhibited significant increases for $O_X$ in all seasons. The largest increases in $O_3$
and $O_X$ were observed during the occurrence of NE-E-SE air masses. The only significant decrease in
$O_X$ at STA was related to the NW wind occurrence during winter. $NO_X$ mixing ratios increased significantly
at all sites, except at OBI, due to the dominant role of industrial sources on $NO_X$ levels. The overall
significant increasing trend of 0.22 ppb $O_3$ yr$^{-1}$ within the MMA contrasts within a significant decreasing
trend of 1.15 ppb $O_3$ yr$^{-1}$ within the MCMA during 1993-2014, whereas a non-significant trend is evident
within the GMA during 1996-2014. At the MCMA and GMA, the overall $O_X$ trends reflect the trends in $O_3$
precursors. According to the long-term trends in $O_3$ for the MMA, the number of exceedances of the air
quality standards will very likely increase as result of increasing precursor emissions. The moderate
mitigation of $O_3$ levels within the MCMA, derived from measures implemented to controle missions from
on-road, industrial and area sources, emphasises the need for more stringent control of emissions mostly
from industrial sources within the MMA in order to improve air quality. Finally, comparison between
emission inventories estimates of NOx and VOCs with ground-based measurements, indicate that
significant reductions in uncertainties are required to better inform air quality policies.

**6. Acknowledgments**
This research was supported by Tecnologico de Monterrey through the Research Group for Energy and
Climate Change (Grant 0824A0104 and 002EICIR01). Grateful acknowledgements are made to the

Secretariat for Sustainable Development of the Nuevo Leon State, the Secretariat for the Environment of Mexico City and the Secretariat for the Environment and Territorial Development of the Jalisco State for the public domain records. We gratefully thank the NOAA Air Resources Laboratory (ARL) for access to the HYSPLIT model and READY website (http://www.ready.noaa.gov), and Dr. Sigfrido Iglesias for providing the imputed $O_3$ and $NO_x$ data for the MMA time-series. We are also grateful to Professor Paul Monks and Professor Richard Derwent for encouraging comments on an earlier version of the manuscript.

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

**Table 1.** Air quality limit values stated in Mexican legislation.

| Pollutant | Mexican Official Standard | Limit value* |
|---|---|---|
| $O_3$ (ppb) | NOM-020-SSA1-1993 | 110 (1-h), 80 (8-h)[a,b] |
| | NOM-020-SSA1-2014 | 95 (1-h) , 70 (8-h)[a,b] |
| $PM_{10}$ ($\mu g\ m^{-3}$) | NOM-025-SSA1-1993 | 75 (24-h), 40 (1-yr) |
| | NOM-025-SSA1-2014 | 50 (24h), 35 (1-yr) |
| $PM_{2.5}$ ($\mu g\ m^{-3}$) | NOM-025-SSA1-1993 | 45 (24-h), 12 (1-yr) |
| | NOM-025-SSA1-2014 | 30 (24-h), 10 (1-yr) |
| CO (ppm) | NOM-02-SSA1-1993 | 11 (8-h)[b] |
| $NO_2$ (ppm) | NOM-023-SSA1 -1993 | 0.21 (1-h) |

*Average period.
[a]Not to be exceeded more than 4 times in a calendar year.
[b]Running average.


**Table 2.** Site description, location and instrumentation used during 1993 to 2014 within the MMA.

| Site | Code | Location | Elevation (m a.s.l.) | Site description |
|---|---|---|---|---|
| Guadalupe | GPE | 25° 40.110' N, 100° 14.907' W | 492 | Urban background site in the La Pastora park, surrounded by a highly populated area, 450 m from Pablo Rivas Rd. |
| San Nicolas | SNN | 25° 44.727' N, 100° 15.301' W | 476 | Urban site surrounded by a large number of industries and residential areas, 450 m from Juan Diego Diaz de Beriagna Rd. |
| Obispado | OBI | 25° 40.561' N, 100° 20.314' W | 560 | Urban site near the city centre of MMA, 250 m from Jose Eleuterio González Rd. and 250 m from Antonio L. Rodríguez Rd. |
| San Bernabe | SNB | 25° 45.415' N, 100° 21.949' W | 571 | Urban site in a residential area downwind of an industrial area with high traffic volume, 140 m from Aztlan Rd. |
| Santa Catarina | STA | 25° 40.542' N, 100° 27.901' W | 679 | Urban site downwind of industrial sources, 200 m from Manuel Ordoñez Rd. |


**Table 3.** Results for $O_3$ and $O_X$ long-term trends expressed in ppb $yr^{-1}$ for 1993-2014 at the 5 sites within the MMA by season.

| Site | Period | Ozone ($O_3$) | | | Odd oxygen ($O_x = O_3 + NO_2$) | | |
|------|--------|---------------|---|---|--------------------------------|---|---|
| | | ppb $yr^{-1}$ | % $yr^{-1}$ | Significance | ppb $yr^{-1}$ | % $yr^{-1}$ | Significance |
| GPE | Annual | 0.21 | 0.78 | * | 0.31 | 0.80 | ** |
| | Spring | 0.24 | 0.73 | * | 0.32 | 0.69 | * |
| | Summer | 0.30 | 1.16 | * | 0.38 | 1.18 | * |
| | Autumn | 0.14 | 0.53 | | 0.25 | 0.62 | |
| | Winter | 0.12 | 0.53 | | 0.14 | 0.33 | * |
| SNN | Annual | 0.33 | 1.40 | *** | 0.45 | 1.25 | * |
| | Spring | 0.39 | 1.38 | * | 0.49 | 1.22 | * |
| | Summer | 0.47 | 2.24 | * | 0.58 | 1.87 | *** |
| | Autumn | 0.41 | 1.96 | * | 0.65 | 1.94 | * |
| | Winter | 0.14 | 0.68 | | 0.23 | 0.58 | + |
| OBI | Annual | 0.30 | 1.29 | * | -0.17 | -0.35 | |
| | Spring | 0.43 | 1.56 | * | 0.02 | 0.03 | * |
| | Summer | 0.26 | 0.98 | * | -0.04 | -0.09 | |
| | Autumn | 0.29 | 1.33 | + | -0.66 | -1.15 | |
| | Winter | 0.25 | 1.46 | | -0.28 | -0.53 | |
| SNB | Annual | 0.19 | 0.65 | + | 0.61 | 1.66 | ** |
| | Spring | 0.37 | 1.07 | + | 0.67 | 1.65 | + |
| | Summer | 0.31 | 1.06 | *** | 0.66 | 2.17 | *** |
| | Autumn | 0.19 | 0.64 | | 0.60 | 1.61 | + |
| | Winter | 0.02 | 0.07 | | 0.47 | 1.12 | + |
| STA | Annual | 0.01 | 0.01 | | -0.15 | -0.28 | |
| | Spring | -0.04 | -0.11 | | -0.01 | -0.02 | |
| | Summer | 0.09 | 0.28 | | 0.13 | 0.27 | |
| | Autumn | 0.00 | 0.00 | | -0.22 | -0.41 | |
| | Winter | -0.09 | -0.43 | | -0.63 | -1.15 | * |

+Level of significance $p < 0.1$.
*Level of significance $p < 0.05$.
**Level of significance $p < 0.001$.
***Level of significance $p < 0.001$.

**Table 4.** Results for $O_3$ and $O_X$ long-term trends by season expressed in ppb yr$^{-1}$ during 1993-2014 for the MCMA and MMA, and during 1996-2014 for the GMA.

| Urban area | Period | Ozone ($O_3$) | | | Odd oxygen ($O_3 + NO_2$) | | |
|---|---|---|---|---|---|---|---|
| | | ppb yr$^{-1}$ | % yr$^{-1}$ | Significance | ppb yr$^{-1}$ | % yr$^{-1}$ | Significance |
| MCMA | Annual | -1.15 | -2.04 | *** | -1.87 | -1.94 | *** |
| | Spring | -0.97 | -1.53 | *** | -1.77 | -1.71 | *** |
| | Summer | -0.97 | -1.88 | *** | -1.44 | -1.67 | *** |
| | Autumn | -1.12 | -2.20 | *** | -1.89 | -2.15 | *** |
| | Winter | -1.62 | -2.64 | *** | -2.47 | -2.27 | *** |
| GMA | Annual | -0.29 | -0.81 | | -1.46 | -1.85 | + |
| | Spring | -0.26 | -0.57 | | -1.89 | -2.07 | * |
| | Summer | -0.10 | -0.32 | | -1.43 | -1.89 | * |
| | Autumn | -0.09 | 0.33 | | -1.40 | -1.97 | * |
| | Winter | -0.34 | -1.01 | | -1.74 | -2.08 | *** |
| MMA | Annual | 0.22 | 0.84 | ** | 0.13 | 0.30 | |
| | Spring | 0.32 | 1.04 | ** | 0.29 | 0.63 | |
| | Summer | 0.27 | 0.99 | *** | 0.28 | 0.72 | *** |
| | Autumn | 0.25 | 1.03 | | 0.13 | 0.31 | |
| | Winter | 0.10 | 0.45 | | 0.01 | -0.01 | |

+Level of significance $p < 0.1$.
*Level of significance $p < 0.05$.
**Level of significance $p < 0.001$.
***Level of significance $p < 0.001$.

**Table 5.** Results for $O_3$ daily maxima long-term trends by season in ppb $yr^{-1}$ during 1993-2014 at the 5
sites within the MMA.

| Site | Period | Ozone ($O_3$) | | |
|------|--------|---------|--------|-------------|
| | | ppb $yr^{-1}$ | % $yr^{-1}$ | Significance |
| GPE | Annual | 0.45 | 1.02 | ** |
| | Spring | 0.48 | 0.94 | ** |
| | Summer | 0.64 | 1.50 | * |
| | Autumn | 0.35 | 0.74 | |
| | Winter | 0.26 | 0.63 | |
| SNN | Annual | 0.79 | 2.13 | *** |
| | Spring | 0.87 | 2.01 | *** |
| | Summer | 0.85 | 2.42 | *** |
| | Autumn | 0.93 | 2.73 | * |
| | Winter | 0.44 | 1.29 | |
| OBI | Annual | 0.65 | 1.51 | * |
| | Spring | 0.78 | 1.62 | ** |
| | Summer | 0.53 | 1.10 | * |
| | Autumn | 0.75 | 1.77 | |
| | Winter | 0.21 | 0.55 | |
| SNB | Annual | 0.40 | 0.80 | *** |
| | Spring | 0.85 | 1.58 | *** |
| | Summer | 0.67 | 1.36 | *** |
| | Autumn | 0.52 | 1.05 | * |
| | Winter | 0.05 | 0.10 | |
| STA | Annual | 0.01 | -0.01 | |
| | Spring | -0.05 | -0.09 | |
| | Summer | 0.22 | 0.35 | |
| | Autumn | -0.07 | -0.12 | |
| | Winter | -0.35 | -0.75 | + |

[+]Level of significance $p < 0.1$.
*Level of significance $p < 0.05$.
**Level of significance $p < 0.001$.
***Level of significance $p < 0.001$.


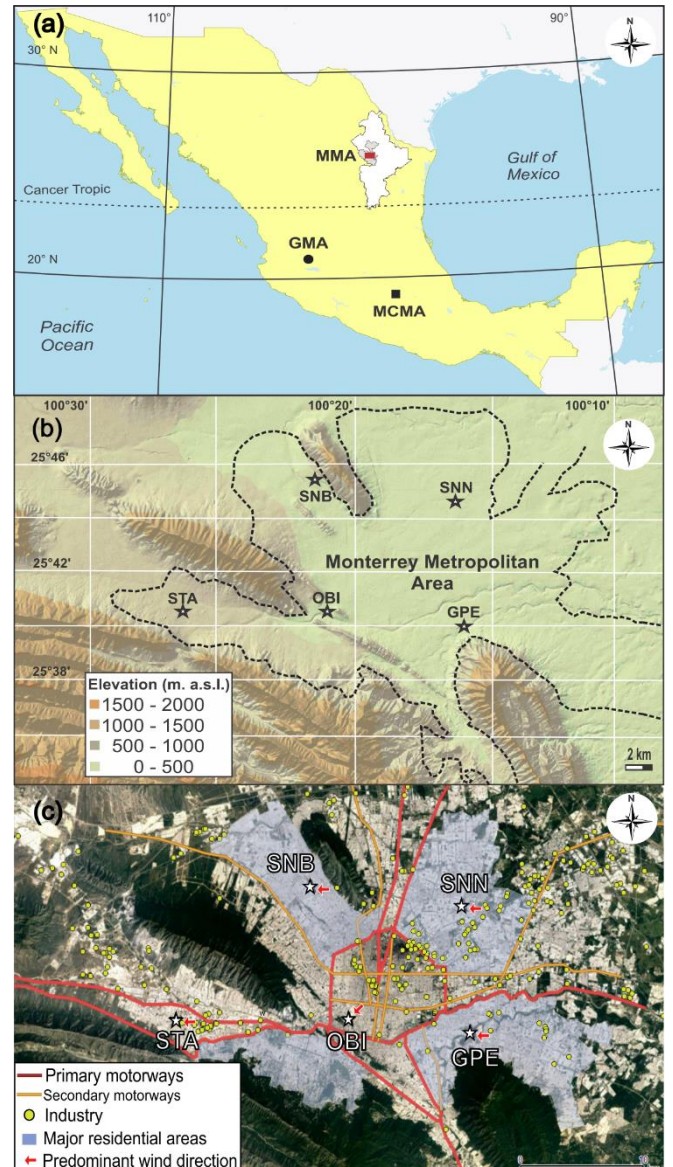


**Fig. 1(a).** The MMA, MCMA and GMA in the national context. **(b).** Topography of the MMA and
distribution of the 5 monitoring sites over the area. **(c).** The 5 monitoring sites in relation to primary and
secondary motorways, industries and major residential areas. The red arrows show the predominant
wind direction at each site during 1993 to 2014.



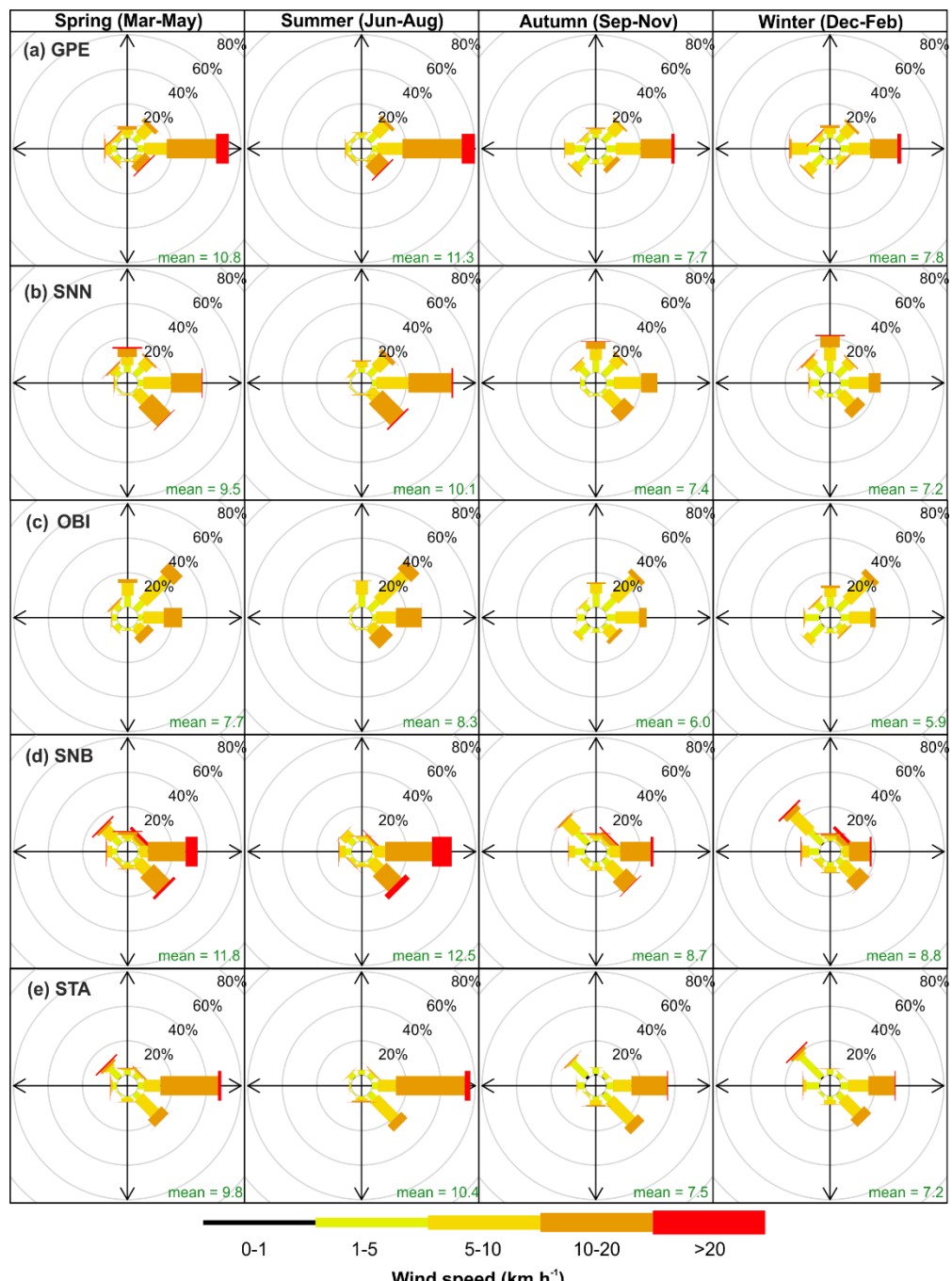

Fig. 2. Frequency of counts of measured wind direction occurrence by season and site within the MMA during 1993-2014.

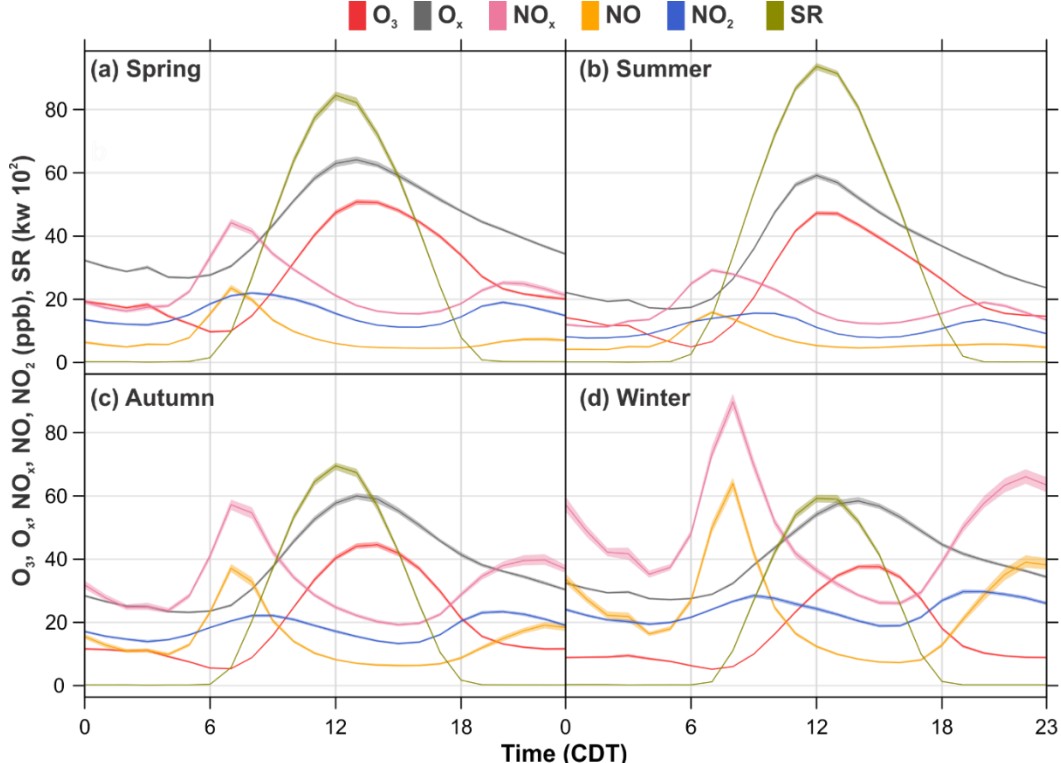


**Fig. 3.** Seasonal average daily profiles for $O_3$, $O_X$, $NO_X$, $NO$, $NO_2$ and SR within the MMA during 1993-2014. The shading shows the 95 % confidence intervals of the average.


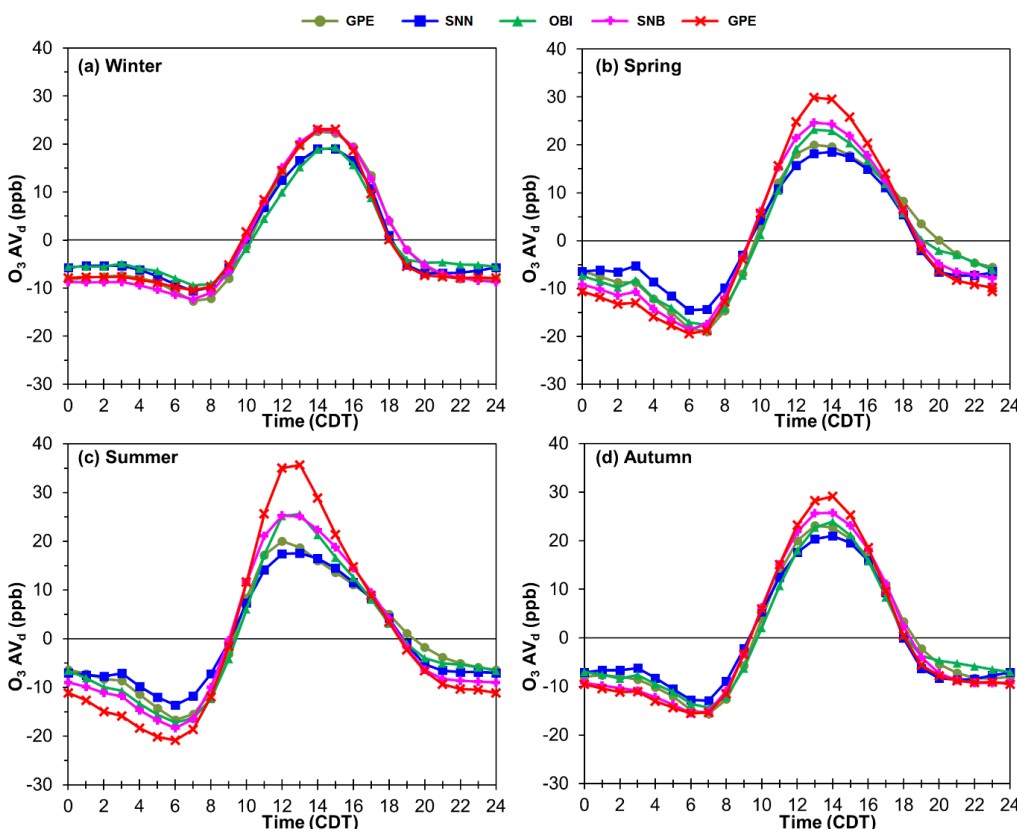


**Fig. 4.** Seasonal $O_3$ de-trended daily profiles within the MMA during 1993-2014. De-trended $O_3$ daily cycles were constructed by subtracting daily averages from hourly averages to remove the impact of long-term trends.


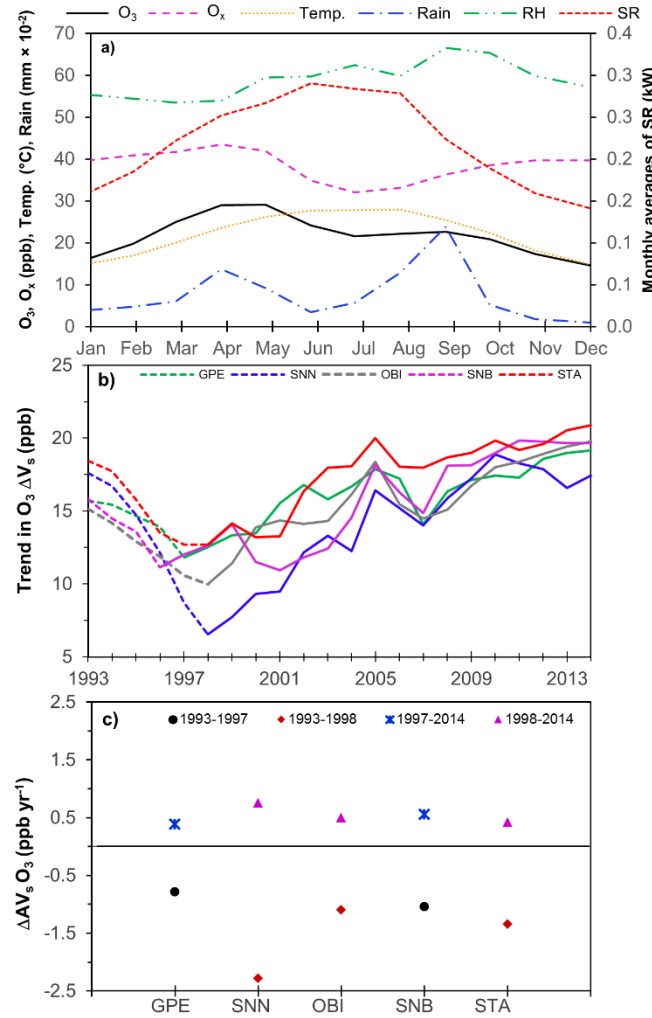


**Fig. 5a).** Annual cycles of $O_3$, temperature, rainfall, RH and SR constructed by averaging records from 1993 to 2014 for a 1-year period. **b).** Trends in $AV_s$ of $O_3$ recorded at the 5 monitoring sites within the MMA from 1993 to 2014. The decline in $AV_s$ observed is due to the economic crisis experienced in Mexico during 1994-1996, followed by persistent increases in $AV_s$ since 1998. **c).** Annual rates of change in $O_3$ $AV_s$ by site, before and after the 1994-1996 economic crisis.

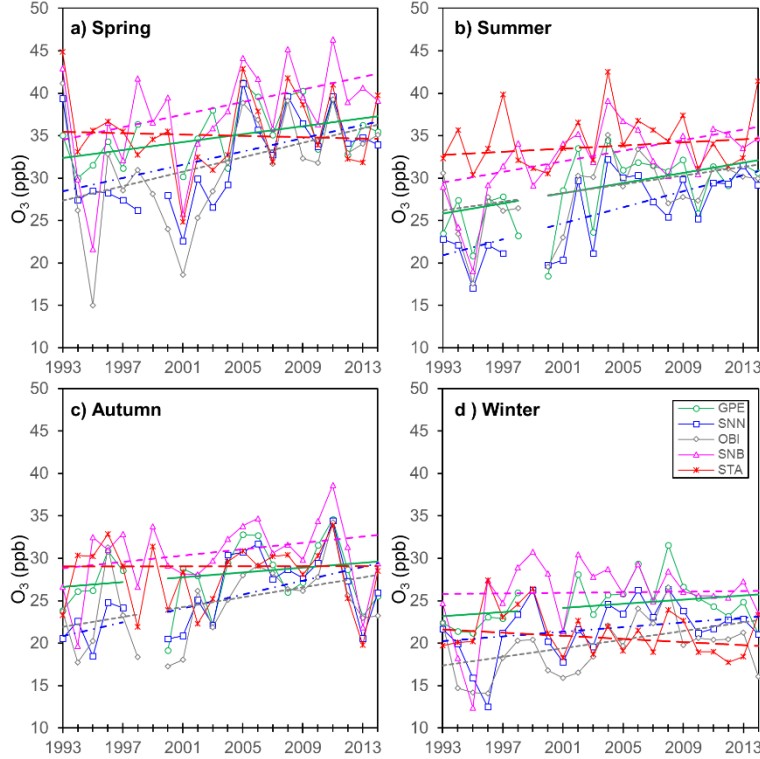


**Fig. 6.** Seasonal trends of $O_3$ within the MMA during 1993-2014. Each data point represents the average
of the 3-month period that defines the season. The continuous lines show the Sen trend.

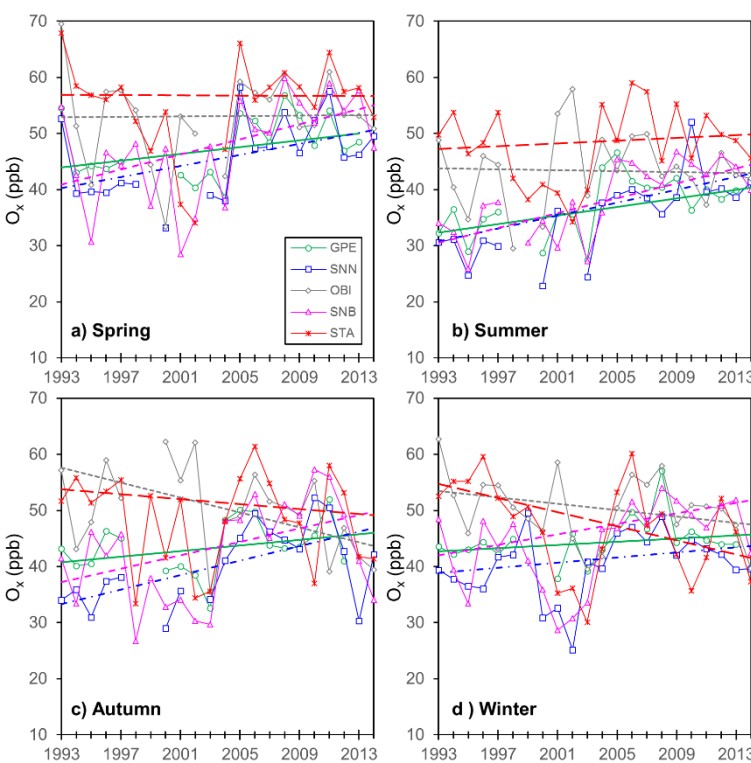


**Fig. 7.** Seasonal trends of $O_X$ within the MMA during 1993-2014. Each data point represents the average
of the 3-month period that defines the season. The continuous lines show the Sen trend.


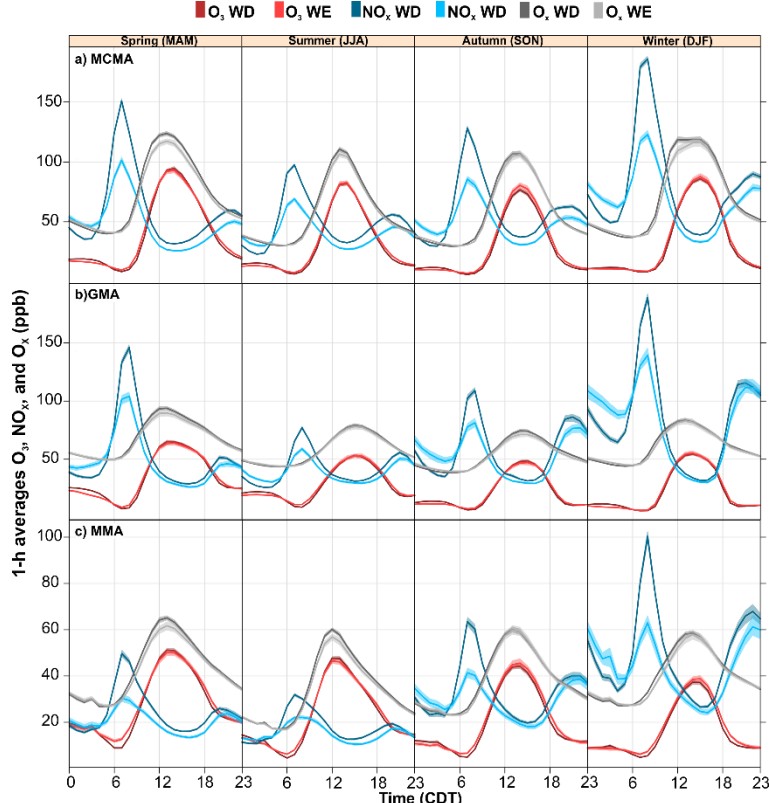

**Fig. 8.** Seasonal average diurnal cycles of $O_3$, $O_X$ and $NO_X$ during 1993-2014 for the MCMA and the MMA, and between 1996-2014 for the GMA. The shading shows the 95% confidence intervals of the average, calculated through bootstrap resampling (Carslaw, 2015).



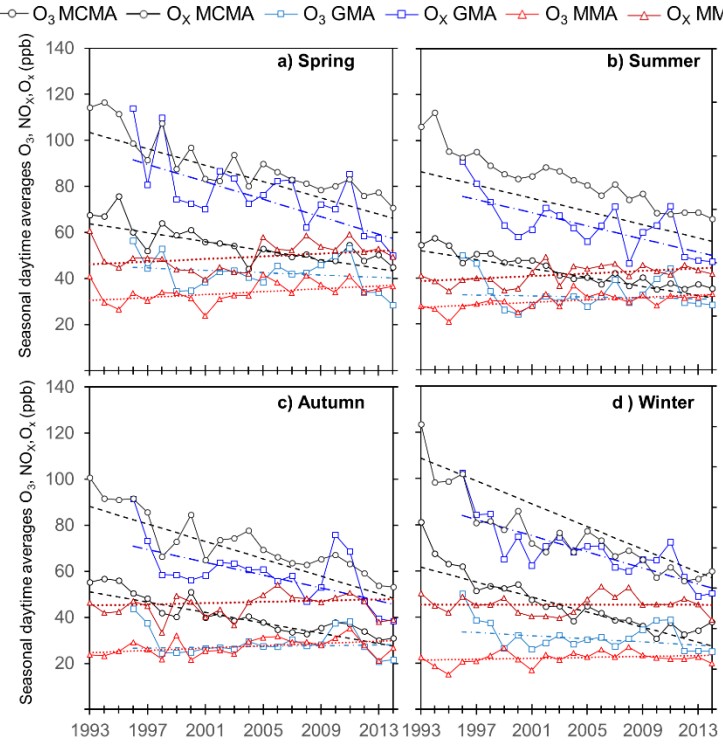


**Fig. 9.** Seasonal trends in $O_3$ and $O_X$ for the MCMA and MMA during 1993-2014, and for the GMA during 1996-2014. Each data point represents the average of the 3-month period that defines the season. The dashed lines show the Sen trend.


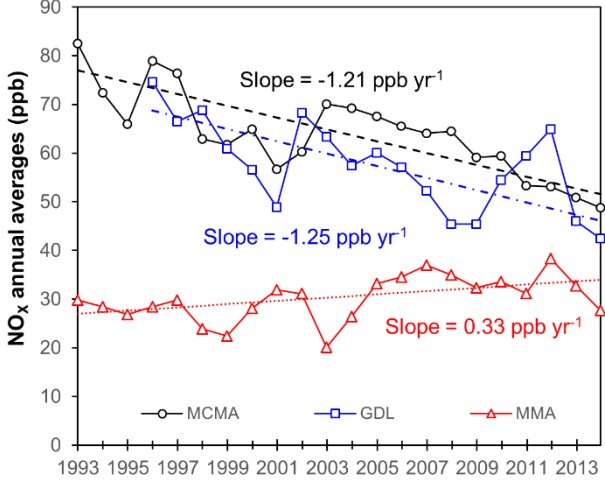

**Fig. 10.** Trends for $NO_X$ at the MCMA and MMA during 1993-2014, and at the GMA during 1996-2014.
The dashed lines represent the Sen slopes. All trends are statistically significant at $p < 0.05$.

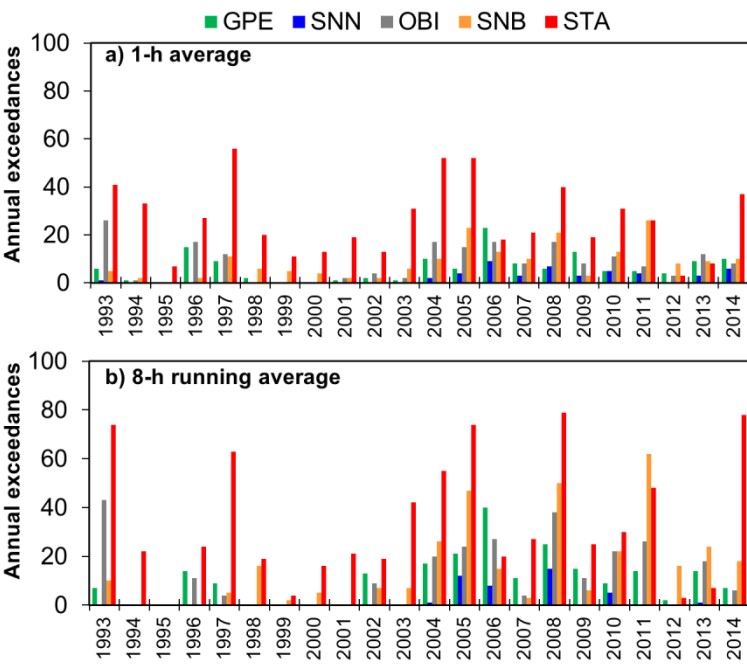

**Fig. 11.** Annual exceedances of the $O_3$ NOM for 1-h averages (110 ppb) and 8-h running averages (80
ppb) at the 5 monitoring sites within the MMA from 1993 to 2014.

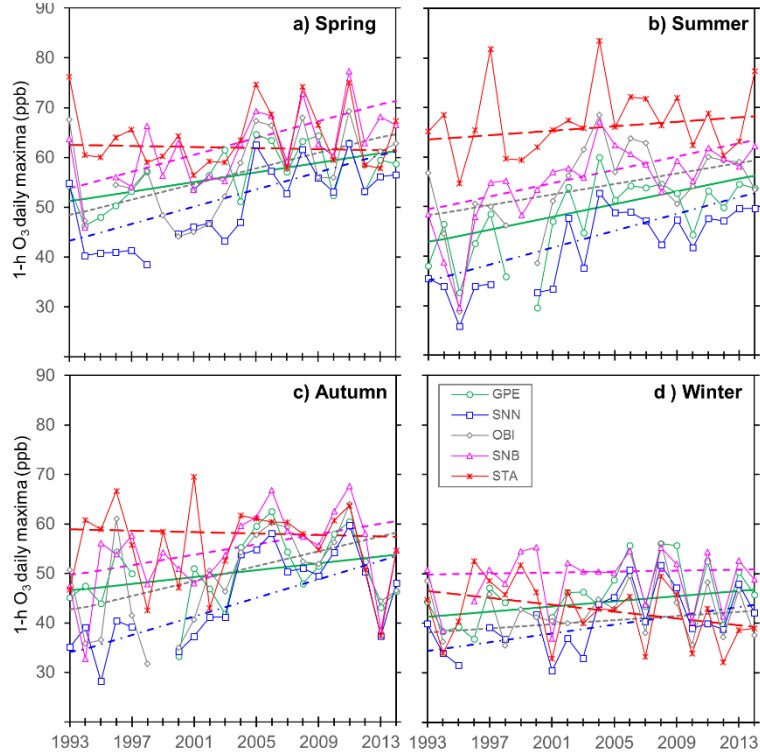

**Fig. 12.** Seasonal trends in 1-h $O_3$ daily maxima at the MMA during 1993-2014. Each data point
represents the average of the 3-month period that defines the season. The dashed lines show the Sen
trend.

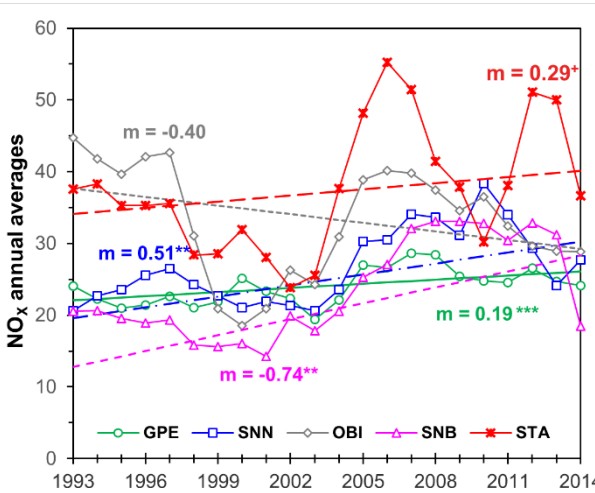

**Fig. 13.** Long-term trends for $NO_X$ at the 5 monitoring sites within the MMA during 1993-2014. The
dashed lines represent the Sen slopes. Annual $NO_X$ rates of change are described as m for slope and
expressed in units of ppb $yr^{-1}$. Levels of confidence are represented as [+] = $p<0.1$, [*] = $p<0.05$, [**] = $p<0.001$,
[***] = $p<0.001$.