# Peer review of "Observed trends in ground-level O3 in Monterrey, Mexico during 1993-2014: Comparison with"

_Atmospheric Chemistry and Physics, 2016_

## Referee Comment (RC1) · Anonymous Referee #1 · 14 Sep 2016

Hernandez et al present trends in ozone precursor emissions and measured ozone levels in three urban areas in Mexico: Monterrey, Mexico City, and Guadalajara. This is an important research topic because, while there has been a long history of ozone trends analysis in the EU and US, there has been relatively little published on trends in other parts of the world. The paper itself needs some revisions before it is suitable for publication in ACP. Please see comments below.

Overarching comments: Trends in emissions of ozone precursor: The authors need to more fully explore trends in ozone precursor emissions and discuss how the trends

shown were derived. They provide some citations but don't address how reliable these sources are and whether there have been methodological changes over time in the emissions estimates that might impact the calculated emissions trends. Since these trends are later used to explain resulting ozone trends, they are a fundamental basis of the paper and need more discussion and exploration. In addition, Duncan et al., 2016 analyzed NOx trends in these three metro areas based on satellite NO2 column measurements between 2005-2014. The NOx trends reported by Duncan et al do not match those reported by the authors in Fig 1a. For instance, Duncan et al (Table S9) found that NO2 had decreased in Guadalajara in this period while Fig 1a suggests that the increased. Additionally, Duncan found that NO2 in Monterrey increased 8x more than NO2 in Mexico City while Fig 1a shows them increasing at similar rates. The authors should compare their results with Duncan et al and use this to explore uncertainties and limitations in in the emissions trends shown in Fig 1a.

Incomplete coverage of past trends work: In the introduction and throughout the paper the authors have a haphazard presentation of past trends work. One of the largest long-term ozone monitoring networks is located in the United States and yet the authors fail to cite any of the numerous studies looking at trends of US ozone (a subset of US trends references are listed at the end of the review). Rather, the authors inexplicably try to understand Monterrey O3 trends by comparing them to studies from London, Tokyo and other far off places with little in common meteorologically or emissions-wise to Mexico. While it is worth discussing broadly the ozone trends across the Northern Hemisphere, the authors have a huge gap in this exploration because they don't include any work from the US. Additionally, when trying to explain/understand O3 phenomena in Mexico, the authors should try to make comparisons to locations that have similar meteorological or emissions change drivers. Instead, the comparisons and reported trends from the literature are discussed in a disjointed way and don't provide an overall picture or provide context for the Mexican trends work presented here.

Lack of transparency of O3 metrics discussed: In the introduction, the authors cite

numerous trends studies and say that ozone has changed by XX ppb but their description leaves out what metrics are being used. A 5 ppb change in annual average O3 would mean something completely different than a 5 ppb change in 5th percentile or 95th percentile O3. Additionally, O3 calculated using all hours versus O3 calculated using daily max (1-hr or 8-hr) will behave quite differently. In order for the reader to fully understand the literature that is being cited, the authors must provide information on which metrics the studies investigated. In addition, while the results in this paper do generally state the metric used, the authors switch between metrics (monthly avg – all hrs, annual avg – all hrs, 1-hr daily max values) without providing the reader with any information on why different metrics were used or how they might relate to each other. The authors need to provide more context in their own results about the meaning of each metric and what it reveals about O3 changes.

Specific comments: Line 43: add ", methane" between "CO" and "and volatile organic compounds".

Line 87-92: It would be helpful if the authors provided some basic background information on the relationship between emissions of NOx and VOC and O3. For instance explaining the conditions under which NOx increases versus decreases O3 concentrations.

Line 142-143: But didn't the authors state that previous trends work had been conducted for Mexico City and Guadalajara?

Lines 176-178 and 180-186: These appear to be results which are stuck in the middle of the methods section. I suggest moving these to the results section.

Lines 195-199: Were new and old instruments ever co-located to inter-compare the measurements? Just following QA procedures is probably not sufficient to control for changes in O3 data due solely to different measurement techniques.

Line 267: GPE had a higher max value than STA according to numbers reported in the

following paragraphs.

Line 276: It would be more accurate if this sentence read: "Reaction with O3 rapidly converts NO to NO2".

Line 287: Here the authors switch from data using all hours (and daily averages) to daily max 1-hr O3 values. They should note the switch and explain the importance of the different metrics.

Line 288: Is this significant? If so state p-value.

Line 289: Here you state that changes 0.79 ppb/yr are "large" but on line 38 you referred to a change of 0.76 ppb/yr as "gradual". Be consistent with characterization of these trends.

Line 290: the authors should state the magnitude and direction of the trend at STA is before discussing causes.

Line 315: What are the daily O3 profiles normalized to? It is not clear what calculations were performed here.

Lines 315-326: It would be interesting if the authors could discuss whether AVd has changed over time.

Lines 329-338: In contrast, the maximum O3 concentrations in the US usually occur in June-August. It would be good to note this difference.

Line 355: The authors state that AVs are similar to those recorded in the US but they have provided no information about the US with which to make this comparison.

Line 371: Are monthly averages calculated using all hours or just daytime max values?

Lines 381-389: Duncan et al can provide NO2 trends at many more locations than just Toronto. Also the US EPA publishes trends reports which include trends in emissions which could be used for comparison.

Lines 444-448: This explanation does not fit with the current literature. The most dramatic weekend/weekday effects have been observed in Southern California under VOC limited conditions, so VOC limitation would not explain the lack of a weekend/weekday effect.

Lines 4851-456: Past work (Simon et al, Cooper et al) has shown that O3 trends are much more pronounced at high percentiles than at average levels, so an annual average may not be a very good metric to use to see long-term trends.

Lines 458-464: Zheng et al and Camalier et al have analyzed the impact of inter-annual meteorological variation on O3 trends. These studies should be cited and discussed.

Lines 466-475: The explanation linking O3 trends to emissions trends does not follow logically and is in contrast to results presented by Duncan et al.

Tables 3 and 4: Are O3 statistics based on hourly O3 data or some other averaging period/daily max period. Please clarify in text and table headings.

Fig 1a: Text should describe how this figure was created from the data sources listed. Do different data sources/years use consistent methodologies?

Fig 3: The label for panel d is missing

Figure 8: How were 95% CIs constructed? Were they based on all daily values? Or on variation among sites in annually averaged profiles? In either case, these confidence intervals look VERY small, I think there is an error in the plotting. It is hard to believe that there would be so little day to day or site to site variability.

References: Camalier, L.; Cox, W.; Dolwick, P. (2007) The effects of meteorology on ozone in urban areas and their use in assessing ozone trends, Atmospheric Environment, 41, 7127-7137

Duncan, B. N., L. N. Lamsal, A. M. Thompson, Y. Yoshida, Z. Lu, D. G. Streets, M. M. Hurwitz, and K. E. Pickering (2016), A space-based, high-resolution view of notable

changes in urban NOx pollution around the world (2005–2014), J. Geophys. Res. Atmos., 121, 976–996, doi:10.1002/2015JD024121.

Zheng, J.; Swall, J.L.; Cox, W.M.; et al. (2007) Interannual variation in meteorologically adjusted ozone levels in the eastern United States: A comparison of two approaches, Atmospheric Environment, 41, 705-716

US O3 trends references (partial list): Berlin, S. R.; Lnagford, A. O.; Estes, M.; Dong, M.; Parrish, D. D. Magnitude, decadal changes, and impact of regional background ozont transported into the greater Houston, Texas, Area. Environ. Sci. Technol. 2014, 47, 13985−13992.

Butler, T. J.; Vermeylen, F. M.; Rury, M.; Likens, G. E.; Lee, B.; Bowker, G. E.; Mc-Cluney, L. Response of ozone and nitrate to stationary source NOx emission reductions in the eastern USA. Atmos. Environ. 2011, 45 (5), 1084−1094.

Chan, E., Regional ground-level ozone trends in the context of meteorological influences across Canada and the eastern United States from 1997 to 2006. J. Geophys. Res.-Atmos. 2009, 114.

Chan, E.; Vet, R. J. Baseline levels and trends of ground level ozone in Canada and the United States. Atmos. Chem. Phys. 2010, 10(18), 8629−8647.

Cooper, O. R.; Parrish, D. D.; Stohl, A.; Trainer, M.; Nedelec, P.; Thouret, V.; Cammas, J. P.; Oltmans, S. J.; Johnson, B. J.; Tarasick, D.; Leblanc, T.; McDermid, I. S.; Jaffe, D.; Gao, R.; Stith, J.; Ryerson, T.; Aikin, K.; Campos, T.; Weinheimer, A.; Avery, M. A. Increasing springtime ozone mixing ratios in the free troposphere over western North America. Nature 2010, 463 (7279), 344−348.

Cooper, O. R.; Gao, R. S.; Tarasick, D.; Leblanc, T.; Sweeney, C., Long-term ozone trends at rural ozone monitoring sites across the United States, 1990−2010. J. Geophys. Res.-Atmos. 2012, 117.

Hidy, G.M.; Blanchard, C.L. (2015) Precursor reductions and ground-level ozone in the

Continental United States, Journal of the Air & Waste Management Association, 65, 1261-1282.

Hogrefe, C.; Hao, W.; Zalewsky, E. E.; Ku, J. Y.; Lynn, B.; Rosenzweig, C.; Schultz, M. G.; Rast, S.; Newchurch, M. J.; Wang, L.; Kinney, P. L.; Sistla, G. An analysis of long-term regional-scale ozone simulations over the Northeastern United States: Variability and trends. Atmos. Chem. Phys. 2011, 11 (2), 567−582.

Lefohn, A. S.; Shadwick, D.; Oltmans, S. J. Characterizing changes in surface ozone levels in metropolitan and rural areas in the United States for 1980−2008 and 1994−2008. Atmos. Environ. 2010, 44 (39), 5199−5210.

Lin, C. Y. C.; Jacob, D. J.; Munger, J. W.; Fiore, A. M. Increasing background ozone in surface air over the United States. Geophys. Res. Lett. 2000, 27 (21), 3465−3468.

Lin, M., Horowitz, L.W., Cooper, O.R. et al (2015) Revisiting the evidence of increasing springtime ozone mixing ratios in the free troposphere over western North America, Geophysical Research Letter, 42, 8719-8728.

Parrish, D. D.; Law, K. S.; Staehelin, J.; Derwent, R.; Cooper, O. R.; Tanimoto, H.; Volz-Thomas, A.; Gilge, S.; Scheel, H. E.; Steinbacher, M.; Chan, E. Long-term changes in lower tropospheric baseline ozone concentrations at northern mid-latitudes. Atmos. Chem. Phys. 2012, 12 (23), 11485−11504.

Pusede, S.E.; Steiner, A.L.; Cohen, R.C. (2015) Temperature and Recent Trends in the Chemistry of Continental Surface Ozone, Chemical Reviews, 115, 3898-3918

Sather, M. E.; Cavender, K. Update of long-term trends analysis of ambient 8-h ozone and precursor monitoring data in the South Central U.S.; encouraging news. J. Environ. Monit. 2012, 14 (2), 666− 676.

Sather, M.E. and Cavender, K. (2016) Trends analyses of 30 years of ambient 8 hour ozone and precursor monitoring data in the South Central US: progress and challenges, Environmental Science-Processes & Impacts, 18, 819-831

[Figure]

Simon, H.; Reff, A.; Wells, B.; et al. (2015) Ozone Trends Across the United States over a Period of Decreasing NOx and VOC Emissions, Environmental Science & Technology, 49, 186-195

Souri, A.H.; Choi, Y.; Li, X.; et al. (2016) A 15-year climatology of wind pattern impacts on surface ozone in Houston, Texas, Atmospheric Research, 174, 124-134

Strode, S.A.; Rodriguez, J.M.; Logan, J.A.; et al. (2015) Trends and variability in surface ozone over the United States. JGR-Atmospheres, 120, 9020-9042.

Tanner, R. L.; Bairai, S. T.; Mueller, S. F. (2015) Trends in concentrations of atmospheric gaseous and particulate species in rural eastern Tennessee as related to primary emission reductions, Atmospheric Chemistry and Physics, 15, 9781-9797.

U.S. EPA, Our Nation's air: Status and trends through 2015. Office of Air Quality Planning and Standards: RTP, NC, 2015, https://gispub.epa.gov/air/trendsreport/2016/
* * *

---

## Referee Comment (RC2) · Anonymous Referee #2 · 4 Oct 2016

This paper by Hernandez Paniagua deals with a very important topic of ground level ozone (O3) and its trend in Monterrey, Mexico (MMA) from 1993-2014. It also presents comparison of O3 trend with other big metropolitan areas in Mexico namely Mexico City (MCMA) and Guadalajara (GMA). While the topic is very important for both air quality and human health, the paper needs some revisions before it is suitable for publication in ACP.

Major Comments:

The paper as it stands is not fully digested. This is one of the major weakness of the paper. It could be significantly concise and coherent without losing any of the messages. A lot of figures could be moved to the supplemental section. For example Figures 3, 4, 5, and to some extent 6 do not really add much to the paper. Similarly, sections describing measurements at MCMA and GMA could be moved to the supplemental section as these measurements are not really different from MMA. On the other hand, the paper will benefit by expanding the analysis section.

There are also comparisons with places around the world that is not very relevant for MMA. While it is important to compare the results with measurements made at other places around the world, these locations need to be very carefully selected. O3 levels depend not only on the emissions of precursors but also the availability of the sunlight. The locations selected in the paper are from everywhere in the world from Canada to Japan, Cyprus to Saudi Arabia. These places do not have similar climate and very likely different emission scenarios and as a result not suitable candidate sites for comparison. Surprisingly there were no comparison being made with any locations in United States which likely has places with similar climate and emission sources (at least in terms of vehicular fleet make up). I suggest the other make a more selective comparison.

Despite the paper claiming it as a study of "impact of emissions of VOCs and NOx on trends of ground level O3", there is very little analysis of emissions of VOCs and NOx in the paper. The only analysis presented is the trend in VOC and NOx emission inventory in MMA, GMA and MCMA (Fig. 1). And, Fig. 1 is mainly used in the introduction and not interpreting the observed trends. It is misleading to claim it as the study of "impact of emissions" as it stands. I suggest the authors, present analysis of NOx and VOCs (CO) measurements and attempt to make connections between VOCs, NOx and O3. A more suitable title for the current paper would be "Trends of ground level O3 in Monterrey, Mexico during 1993-2014: Comparison with Mexico City and Guadalajara". The lack of NOx and VOCs trend makes most of the current conclusion seem more like speculations.

[Figure]

There are too many different O3 metrics being used in the paper and the authors constantly switch between them. There is no rationale being presented for why a certain metric is being used. I suggest the authors minimize the number of metric being used if possible or justify the use of different metrics in terms of what it reveals about O3 in MMA.

Specific Comments:

Line 23: Why is larger AVd observed at polluted sites close to industrial areas? O3 being a secondary pollutant should show larger AVd at downwind sites? In fact, the largest AVd is observed at STA which is furthest away from the industrial areas.

Line 30: GPE is described as highly populated area downwind of an industrial area in Table 1. So, GPE qualifies as site with largest and smallest seasonal cycle (AVs).

Line 70: Introduction: The introduction switches between O3 trend in rural background and urban areas. I suggest the authors focus solely on the urban areas as this is the focus of this particular paper.

Line 133-135: Based on Figure 14, O3 has gone up in MMA by only around 20-25% and decreased in GMA.

Line 172-186: This section belongs to the results and discussion section. These could be used to explain some of the observations rather than keeping it in the methodology section. Further, I suggest figures 3 and 4 be moved to a supplemental section to make the paper more concise.

Line 188: What is the time resolution of these measurements?

Line 204-235: Section 2.2 and 2.3: I suggest moving these sections to the supplemental as well. Most of the analysis focuses on the MMA and these two sections only describe the locations and measurements at MCMA and GMA. These measurements are not different from MMA. A reference sentence at the end of MMA measurement section would be sufficient.

Line 237: Section 2.4: I suggest the author expand this section to describe all the methods used in the analysis of the data. Please add a brief description for (i) ope-nair package for R, (ii) MAKESENS 1.0 macro, (iii) Seasonal Trend Decomposition technique. Please include how do they work or what is being done in each of these program and what are the advantages of such an analysis to reveal changes in O3?

Line 238: I suggest that figure 5 be moved to the supplemental section.

Line 265: It would be better to show 5, 50 and 95th percentile line for the data in Figure 6 than all the data.

Line 270: Add "(see Figure 8)" behind "(winter)".

Line 276: Reaction of O3 with NO to form NO2 is only a part of the null Ox cycle. Please include the full Ox cycle.

Line 278: Ox cannot have a minimum value of 0. This would require both NO2 and O3 to be 0. It is very likely a measurement error or lack of measurements. I suggest the authors employ some kind of data filtering. There are also instances where Ox is lower than O3 which is again not possible.

Line 282-285: It would be very helpful to include trends in measured NOx and CO to interpret the observed trends in O3 and Ox. This would also be in line with the title of the paper.

Line 287: Please add description of how the data is de-seasonalised.

Line 289: A NOx trend would also help justify this statement regarding increased local-ized industrial emissions.

Line 290-292: Isn't this the further study to connect emissions with O3?

Line 298: It is surprising that new vehicles are only limited to the city center or the impact of new vehicles are only seen there. Is there some kind of restrictions on the age limit of vehicle that can enter the city center? Else you should see the benefit over

the whole MMA unless the industrial emissions are offsetting the vehicular emissions.

Line 305: Is there seasonality in rush hour or is the observed shift in O3 dip due to change in time i.e. day light saving?

Line 315: How is the normalization performed? Do you subtract the mean O3?

Line 318: All the sites show lowest AVd in winter not just SNN.

Line 318-320: Please justify this statement or add a reference. This statement seems to be misplaced. The lowest AVd during winter is likely due to availability of less sunlight and subsequent slower photochemistry as shown in Section 3.3 and not due to inflow of VOC and NOx laden air masses. These should instead enhance O3 production resulting in larger AVd.

Line 340: It is hard to see what is going on in Figure 10. Please consider adding a second panel focusing only on one of the years.

Line 341: How does reduced rainfall decrease O3 levels? In line 345-346, it is mentioned that frequent rain storms suppresses O3 levels in late summer and early autumn. These two statements are contradictory.

Line 359: What is the benefit of calculating AVs. It is currently not clear. One could do a similar trend analysis without calculating AVs.

Line 364: How did NOx and CO change during the economic crisis? A large decrease in NOx was observed in US and Europe during the last recession (Castellanos et al., 2012, Russell et al., 2012). Do you see similar decrease in NOx as well?

Line 376: Please see previous comments regarding NOx trend.

Line 385: Please see previous comments regarding new vehicle being limited to city centers.

Line 399: Why would accumulation or stagnation of air mass not result in an increasing

trend? If more O3 and precursors are coming in from nearby places, then O3 should go up?

Line 400: Figure 4 does not show stagnation at STA.

Line 423: SNB has growth rate of -0.06 ppb not SNN.

Line 437: Please add p value.

Line 431: Are there any sites upwind of the MMA industrial area? This would help interpretation of the data.

Line 442: Figure 1 shows largest emissions for GMA in recent years. This contradicts the statement being made about Figure 1 describing the magnitude of AVd s in three cities.

Line 445-448: The statement regarding weekend effect is not clear. Figure 13 does not differentiate between weekday and weekend. It is not clear whether NOx or VOCs decrease during the weekend. So, the statement regarding why no differences in O3 is observed between weekday and weekend is not appropriate.

Line 450: Section 3.7: I suggest the authors consolidate the trend in O3 in the three metropolitan areas to the observation based only on the trend line. A statement regarding why the trend line is more appropriate than randomly choosing the start and end data to get the reduction/increase in O3 is justified.

Line 462: Why is there such a large variance in the annual averages?

Line 471: Figure 1 shows both VOCs and NOx are going up for MCMA not going down. So, why is O3 going down with both the precursors going up?

Line 487: Which standard is used for Table 5, new or the old one? If it is a mixed of two then, the data is not directly comparable. I suggest using the new O3 standard for all years.

Line 493: It is hard to evaluate the statement without not knowing what is represented in table 5. 2012 and 2013 showed a significant reduction in number of days exceeding the standard. Then, there is a big jump in 2014. If the big jump in 2014 due to the change in the standard, then it is kind of misleading to say "recommended that more stringent emission controls are introduced in order to improve air quality within the MMA".

Table 3: I suggest moving it to the supplemental section.

Table 5: Which standard is being used to calculate these exceedances?

Figure 2: Please remove the wind rose plot from OBI and add predominant wind direction for each site as a single arrow for each site.

Figure 9: Legend is missing STA. There are two GPE s instead.

Figure 10: Please add a zoom into one of the years.

References:

Castellanos, P. and Boersma, K. F.: Reductions in nitrogen oxides over Europe driven by environmental policy and economic recession, Scientific Reports, 2, 265, 1–7, doi:10.1038/srep00265, 2012.

Russell, A. R., Valin, L. C., and Cohen, R. C.: Trends in OMI NO2 observations over the United States: effects of emission control technology and the economic recession, Atmos. Chem. Phys., 12, 12197-12209, doi:10.5194/acp-12-12197-2012, 2012.

---

## Author Comment (AC1) · 20 Jan 2017

We thank the reviewers for the thorough revisions and for providing constructive comments on our manuscript, "Trends of ground-level $O_3$ in Mexico during 1993-2014: Comparison of Monterrey with Mexico City and Guadalajara". We are pleased that the editor and reviewer's perspective on addressing $O_3$ long-term trends in Mexican urban areas is in agreement with our own views on the issue. We have addressed the concerns and recommendations received, and we believe that these helped to improve significantly the quality of our manuscript. Please find below our detailed response to the comments received, which are also highlighted in red in the revised version of the manuscript, submitted along with this response.

**Reviewer #1:**

Hernandez et al present trends in ozone precursor emissions and measured ozone levels in three urban areas in Mexico: Monterrey, Mexico City, and Guadalajara. This is an important research topic because, while there has been a long history of ozone trends analysis in the EU and US, there has been relatively little published on trends in other parts of the world. The paper itself needs some revisions before it is suitable for publication in ACP. Please see comments below.

**Overarching comments:**

*Trends in emissions of ozone precursor:* The authors need to more fully explore trends in ozone precursor emissions and discuss how the trends were derived. They provide some citations but don't address how reliable these sources are and whether there have been methodological changes over time in the emissions estimates that might impact the calculated emissions trends. Since these trends are later used to explain resulting ozone trends, they are a fundamental basis of the paper and need more discussion and exploration. In addition, Duncan et al., 2016 analyzed $NO_x$ trends in these three metro areas based on satellite $NO_2$ column measurements between 2005-2014. The $NO_x$ trends reported by Duncan et al do not match those reported by the authors in Fig 1a. For instance, Duncan et al (Table S9) found that $NO_2$ had decreased in Guadalajara in this period while Fig. 1a suggests that the increased. Additionally, Duncan found that $NO_2$ in Monterrey increased 8x more than $NO_2$ in Mexico City while Fig 1a shows them increasing at similar rates. The authors should compare their results with Duncan et al and use this to explore uncertainties and limitations in the emissions trends shown in Fig 1a.

***Response:*** The reviewer is right, no details regarding the methodology used to obtain the estimates of emissions and their uncertainty were included previously. The source of the emission estimates reported here and the methodologies used to obtain them were included in section 2.2 NEI data. See Lines: 181-205. We have also modified Fig. 1, now Fig. S1 to include more concise information, and discussed in the introduction section the uncertainties in the emission estimates reported in existing studies. See lines 108-122, 124-133.

**Text modified:**
**"2.2 NEI data**
Estimates of $NO_x$ and VOCs emissions have been made at the national scale for the 1999-, 2005- and 2008-base years and reported in the NEI, and were obtained from the SEMARNAT website (http://sinea.semarnat.gob.mx). The data set is provided by emission source (mobile, point, area and natural), air pollutant, and at national, state and municipality scales. The NEI emission estimates are developed in accordance with the Manual for the Emission Inventories Program of Mexico (Radian, 2000), which is based on the US EPA AP-42 emission factors categorisation (EPA, 1995). The emission factors are regionalised for each Mexican state, based upon on-site measurements and survey information. Updates to the emission factors have been conducted for each released NEI, although no

changes in the methodology were implemented between the 1999- and 2008-base years. Overall, the mobile emissions were estimated using the MOBILE6-Mexico model (EPA, 2003). The emissions from point sources were derived using the annual operation reports submitted to the Environment Ministry. The emissions from area sources were obtained using the categorisation of Mexican area sources and the regionalised AP-42 emission factors.

The MCMA emissions inventories have been developed with a 2-year frequency since 1996, and were obtained from the MCMA Environment Secretariat website (http://www.aire.cdmx.gob.mx/). The methodology used to construct the MCMA inventories estimates is consistent with that used in the NEI (SEDEMA, 2016a), which is based on the AP-42 EPA emission factors. However, more speciated emission factors have been developed in each released version, considering updates in the local industrial activity, survey information and field measurement campaigns. To date, the only significant change in the methodology is the replacement of the Mobile6-Mexico model with the MOVES model to obtain the 2014-base year mobile emissions (SEDEMA, 2016b). As for the MCMA inventories, more speciated emission factors than those contained in the NEI were developed to produce the MMA emissions inventory 2013-base year (SDS, 2015), although, mobile emissions estimates were obtained with the Mobile6-Mexico model (EPA, 2003).".

*Incomplete coverage of past trends work:* In the introduction and throughout the paper the authors have a haphazard presentation of past trends work. One of the largest long-term ozone monitoring networks is located in the United States and yet the authors fail to cite any of the numerous studies looking at trends of US ozone (a subset of US trends references are listed at the end of the review). Rather, the authors inexplicably try to understand Monterrey $O_3$ trends by comparing them to studies from London, Tokyo and other far off places with little in common meteorologically or emissions-wise to Mexico. While it is worth discussing broadly the ozone trends across the Northern Hemisphere, the authors have a huge gap in this exploration because they don't include any work from the US. Additionally, when trying to explain/understand $O_3$ phenomena in Mexico, the authors should try to make comparisons to locations that have similar meteorological or emissions change drivers. Instead, the comparisons and reported trends from the literature are discussed in a disjointed way and don't provide an overall picture or provide context for the Mexican trends work presented here.

**Response:** The reviewer is right, no data regarding $O_3$ trends in US urban areas were included previously. As requested, we re-wrote the introduction section to include relevant information of $O_3$ trends in the US. We thank the reviewer for the list of references provided. See lines: 81-92.

*Lack of transparency of $O_3$ metrics discussed:* In the introduction, the authors cite numerous trends studies and say that ozone has changed by XX ppb but their description leaves out what metrics are being used. A 5 ppb change in annual average $O_3$ would mean something completely different than a 5 ppb change in 5th percentile or 95th percentile $O_3$. Additionally, $O_3$ calculated using all hours versus $O_3$ calculated using daily max (1-hr or 8-hr) will behave quite differently. In order for the reader to fully understand the literature that is being cited, the authors must provide information on which metrics the studies investigated. In addition, while the results in this paper do generally state the metric used, the authors switch between metrics (monthly avg – all hrs, annual avg – all hrs, 1-hr daily max values) without providing the reader with any information on why different metrics were used or how they might relate to each other. The authors need to provide more context in their own results about the meaning of each metric and what it reveals about $O_3$ changes.

**Response:** The reviewer is right, there was no description of the metrics used to derive the cited $O_3$ long-term trends. As requested, the metrics used to assess the changes in $O_3$ reported in the introduction were included in the text. See lines: 65, 71, 72, 74, 75, 80, 83-84, 88 and 89. Regarding the relevance of the metrics described in the current study, a sentence describing this was included before the discussion of each metric addressed. See lines: 135, 307-309, 367-371 and 402-403.

**Specific comments:**

Line 43: add ", methane" between "CO" and "and volatile organic compounds". *Response:* ", methane" was added. See line: 43.

Line 87-92: It would be helpful if the authors provided some basic background information on the relationship between emissions of NOx and VOC and $O_3$. For instance explaining the conditions under which $NO_x$ increases versus decreases $O_3$ concentrations.
*Response:* A brief description of the $O_3$ production regimes has been added. Text modified: "The system of $O_3$ production is not linear, being VOC-limited whether it responds to the input of VOCs, or $NO_x$-limited, whether $O_3$ production increases in response to increasing $NO_x$ emissions (Monks et al., 2015; Pusede et al., 2015).". See lines: 45-47.

Line 142-143: But didn't the authors state that previous trends work had been conducted for Mexico City and Guadalajara?
*Response:* We have clarified in the text that the existing studies have focused mostly on long-term trends in $O_3$ within the MCMA. Additionally, we have stated that to date, only Benitez-Garcia et al. (2014) have considered changes in ground-levels of $O_3$ within the GMA and the MMA, however their results were obtained using the non-robust, simple regression analysis of annual averages, which could result in significant misestimations of the actual trends. See lines: 102-106.

Lines 176-178 and 180-186: These appear to be results which are stuck in the middle of the methods section. I suggest moving these to the results section.
*Response:* As suggested, lines 176-175 and 180-186 were moved to the results section, which are now part of section "**3.1 Wind occurrence at the MMA***". See lines: 272-279.

Lines 195-199: Were new and old instruments ever co-located to inter-compare the measurements? Just following QA procedures is probably not sufficient to control for changes in $O_3$ data due solely to different measurement techniques.
*Response:* Unfortunately, no simultaneous measurements of $O_3$ were performed using the 49 and 49C instruments; since the analysers model 49 reach their recommended operative life by early 2003, when were replaced with the analysers model 49C. However, to rule out the impact of different instrumentation and calibration methodologies studies have recommended the use of 3-yr $O_3$ averages, when no intercomparing measurement period was conducted. For instance, Akimoto et al. (2015) used 3-yr averages of $O_3$ when assessing long-term changes in ground-levels of $O_3$ at 4 large metropolitan areas of Japan. Similarly, Parrish et al. (2011) assessed the decreases in the $O_3$ 4[th] highest annual maximum mixing ratio within Los Angeles and the MCMA, using data calculated as 3-yr averages. This was noted in Lines: 171-173. Additionally, long-term trends in $O_3$ annual averages were compared with those derived using the methodology as above in Supplementary information S1.1 (Fig. S2).

Text modified:
**S1.1 Comparison of long-term trends in $O_3$ annual averages with 3-yr averaged data**
Linear trends were tested both for $O_3$ annual averages and 3-yr average $O_3$ data with the non-parametric Theil-Sen approach. Although, slight larger $O_3$ growth rates are determined for the smoothed data than for the annual averages as shown in Fig. S2, non-significant differences ($p>0.05$) were observed between both Sen slopes. Considering this, and that the smoothing of $O_3$ annual averages could lead to miss significant features in the current trends (Carslaw et al., 2007; Carslaw, 2015), in the current study, $O_3$ annual averages with no smoothing were used to determine the long-term trends reported at the 3 metropolitan areas.

[Figure]

**Fig. S2.** Comparison of long-term trends in for O3 annual averages (1993-2014), and 3-yr average O$_3$ data (1993-2012). The dashed lines represent the Sen slopes. Statistical significance is expressed as $p<0.1 =^+$, $p<0.05 = *$, $p<0.01 = **$ and $p<0.001 = ***$.

Carslaw, D. C., and Carslaw, N.: Detecting and characterising small changes in urban nitrogen dioxide concentrations, Atmos. Environ., 41, 4723-4733, doi:10.1016/j.atmosenv.2007.03.034, 2007.

Parrish, D. D., Singh, H. B., Molina, L., and Madronich, S.: Air quality progress in North American megacities: A review, Atmos. Environ., *45*, 7015-7025. doi:10.1016/j.atmosenv.2011.09.039, 2011.

Akimoto, H., Mori, Y., Sasaki, K., Nakanishi, H., Ohizumi, T., and Itano, Y.: Analysis of monitoring data of ground-level ozone in Japan for long-term trend during 1990-2010: Causes of temporal and spatial variation, Atmos. Environ., 102, 302-310, doi:10.1016/j.atmosenv.2014.12.001, 2015.

Carslaw, D. C.: The openair manual - open-source tools for analysing air pollution data, Manual for version 1.1-4, King's College London, 2015.

Line 267: GPE had a higher max value than STA according to numbers reported in the following paragraphs.
***Response:*** The reviewer is right. There was a mistake in the sentence, we have specified the site where the highest O$_3$ 1-h average mixing ratio was measured during the studied period. Text modified: "The highest O$_3$ 1-h average was observed at SNB,". See lines: 283-285.

Line 276: It would be more accurate if this sentence read: "Reaction with O3 rapidly converts NO to NO$_2$".
***Response:*** The sentence was modified as suggested. Text modified: "Reaction with O$_3$ rapidly converts NO to NO$_2$,". See line: 294.

Line 287: Here the authors switch from data using all hours (and daily averages) to daily max 1-hr $O_3$ values. They should note the switch and explain the importance of the different metrics.

***Response:*** As requested by the reviewer, the relevance of the long-term trend assessment for maximum $O_3$ 1-h averages was stated. Text added: " A study conducted among asthmatic children resident in the MCMA revealed an increase in coughing and wheezing rates, associated with cumulative exposure to high 1-h averages mixing ratios of $O_3$ and $NO_2$ (Escamilla-Nuñez et al., 2008). To assess changes in cumulative exposure to $O_3$ and $O_x$ within the MMA, long-term trends of de-seasonalised maximum daily 1-h averages in $O_3$, $O_x$ and $NO_x$ were calculated, using annual averages filtered with the STL technique (Fig. 4). ". See lines: 307-311.

Escamilla-Nuñez, M. -C., Barraza-Villarreal, A., Hernandez-Cadena, L., Moreno-Macias, H., Ramirez-Aguilar, M., Sienra-Monge, J. -J., Cortez-Lugo, M., Texcalac, J.-L., del Rio-Navarro, B., and Romieu, I.: Traffic-related air pollution and respiratory symptoms among asthmatic children, resident in Mexico City: The EVA cohort study, Respir. Res., 9, doi:10.1186/1465-9921-9-74, 2008.

Line 288: Is this significant? If so state *p*-value.

***Response:*** As requested by the reviewer, *p*-values were written along the text where required. See line: 312.

Line 289: Here you state that changes 0.79 ppb/yr are "large" but on line 38 you referred to a change of 0.76 ppb/yr as "gradual". Be consistent with characterization of these trends.

***Response:*** The reviewer is right. We have changed "gradual" in Line 38 to "large" in order to be consistent with the trends characterisation. See line: 38.

Line 290: the authors should state the magnitude and direction of the trend at STA is before discussing causes.

***Response:*** As requested by the reviewer, the magnitude of the trend and significance value were stated. Text added: " By contrast, the non-significant ($p>0.05$) trend of -0.01 ppb $O_3$ $yr^{-1}$ observed at STA is may be masked by local import of $O_3$…". See lines 315-318.

Line 315: What are the daily $O_3$ profiles normalized to? It is not clear what calculations were performed here.

***Response:*** The reviewer is right. We have stated how the normalised cycles were constructed. Text added: " To compare the $O_3$ diurnal cycles by season, normalised daily profiles were constructed by subtracting daily averages from hourly averages in order to remove the impact of the long-term trends (Fig. 6; Hernández-Paniagua et al., 2015), with daily amplitude values ($AV_d$; calculated by subtracting the lowest normalised values from the highest normalised values) used to assess diurnal variations in $O_3$ among seasons.". See lines: 345-349.

Lines 315-326: It would be interesting if the authors could discuss whether $AV_d$ has changed over time.

***Response:*** As requested by the reviewer, the long-term trends in $AV_d$s from 1993 to 2014 were determined for the 5 sites within the MMA. Figure 7 shows the long-term trends in $AV_d$s, which are discussed in the manuscript. See lines: 361-371.

[Figure]

**Fig. 7.** Long-term trends of $AV_d$ $O_3$ annual averages at the 5 sites within the MMA during 1993-2014. The dashed lines represent the Sen slopes. Statistical significance is expressed as $p<0.1 =^+$, $p<0.05 = *$, $p<0.01 = **$ and $p<0.001 = ***$.

Lines 329-338: In contrast, the maximum $O_3$ concentrations in the US usually occur in June-August. It would be good to note this difference.

***Response:*** As requested, $O_3$ seasonal cycles within the MMA are compared with those reported for several regions of the US, including the southeast. This as the MMA is also influenced by air masses from the Gulf of Mexico. Text modified: "Figure 8b shows the seasonal cycles of $O_3$, with spring-time maxima and winter minima, in strong correlation with SR (Lelieveld and Dentener, 2000).

This behaviour agrees well with the $O_3$ spring maxima and winter minima characteristic of the US southeast regions (Strode et al., 2015), and follows the NH mid-latitudes $O_3$ cyclic pattern (Monks 2000; Vingarzan, 2004). However, it differs with the $O_3$ seasonal cycles observed over the US west coast regions (particularly in California), where the maxima occur between June-August, in response to the local influence of precursor emissions upon $O_3$ production and photochemical conditions (Vingarzan, 2004; Strode et al., 2015). By contrast, downward spikes in the seasonal cycles of $O_3$ within the MMA are observed recurrently between July-August (Fig. 8b), which likely result from high wind speeds (>6 km h$^{-1}$ in average) that disperse $O_3$ precursors and increase the boundary layer height (ProAire-AMM, 2008), and high day-time temperatures (>40° C) that could suppress the $O_3$ formation. Steiner et al. (2010) reported that within VOC-limited areas, temperatures >38° C may lead to decreases in $O_3$ formation, in response to a decrease in the peroxyacetyl nitrate lifetime (NO$_x$ sink). The peak in $O_3$ observed in September is characteristic of humid regions, and can be ascribed to an increase in OH radicals derived from the increment in RH during the rainy season (Lee et al., 2014). Zheng et al. (2007) reported that this $O_3$ secondary peak became less noticeable since 2000 over the mid-western and

eastern US regions. Indeed, the $O_3$ secondary peak is characteristic of the Asian summer monsoon, which transports maritime clean air to land with constant rainfall, thereby increasing RH (Xu et al., 2008).". See lines: 381-400.

Line 355: The authors state that AVs are similar to those recorded in the US but they have provided no information about the US with which to make this comparison.
***Response:*** Data of the seasonal cycles over the US were included in order to discuss $AV_s$ with those observed within the MMA. Text modified: " $AV_s$ for the MMA are similar to those calculated using dynamic linear models by Zheng et al. (2007), over the mid-western US region between ca. 12 ppb $O_3$ in 2004 and 18 ppb $O_3$ in 1999, but lower than those between ca. 19 ppb $O_3$ in 2004 and 27 ppb $O_3$ in 1999 determined for the eastern region. When compared with European regions, the $AV_s$ determined within the MMA are slightly lower than those calculated at the North Kensington site in London, which ranged from ca. 7.0 ppb $O_3$ in 2000 to ~25.5 ppb $O_3$ in 2005 (Bigi and Harrison, 2010), presumably due to lower emissions of $NO_x$ and VOCs within the MMA (SDS, 2015). It is striking that the average $AV_s$ for the MMA agrees well with that of 10.5 ppb $O_3$ recorded during 2004-2005 at the Pico Mountain Observatory in Portugal, which is a receptor of exported NA air pollution (Kumar et al., 2013). Thus, despite trends of increasing $O_3$ precursor emissions within the MMA, $AV_s$ lie within the range of those recorded at sites in the mid-west US, but are slightly lower than those determined for more populated and urbanised sites in the east US and Western Europe.". See lines: 407-418.

Line 371: Are monthly averages calculated using all hours or just daytime max values?
***Response:*** The $O_3$ monthly averages were derived from daily averages of all 1-h data, as described in section 2.3, which was clarified in the manuscript. Text added: "The long-term trends were constructed from de-seasonalised annual data derived from monthly averages filtered with STL, which were calculated from daily data of all 1-h averages, as described in Methodology (Sect. 2.3). ". See lines: 440-442.

Lines 381-389: Duncan et al. can provide $NO_2$ trends at many more locations than just Toronto. Also the US EPA publishes trends reports which include trends in emissions which could be used for comparison.
***Response:*** As requested, we have discussed the $O_3$ trends observed in terms of response to changes in $NO_x$ determined within the MMA, with contrast with the $NO_2$ trends reported by Duncan et al. and economic indicators. See: Figure 10. Additionally, studies of relevance conducted within the MMA and from the list provided by the reviewer were used to discuss and explain the observed trends in $O_3$ within the MMA. See lines: 459-465, 469-479, 481-489, 491-503.

Lines 444-448: This explanation does not fit with the current literature. The most dramatic weekend/weekday effects have been observed in Southern California under VOC limited conditions, so VOC limitation would not explain the lack of a weekend/weekday effect.
***Response:*** As requested by the reviewer, the discussion of the $O_3$ weekly cycles section was re-written. We discussed the non-significant changes between weekdays and weekends reported in our study with those reported by Wolff et al. (2013) for urban areas across the US. Moreover, we provide a plausible explanation for the $O_3$ weekly patterns observed, based on i) the assessment of ambient levels of $O_3$ as reported by Torres-Jardon (2004) for the MCMA and, ii) the vanishing effect in urban areas of the Southern California reported by Wolff et al. (2013). Text modified: "No significant differences ($p$>0.05) were observed at any of the metropolitan areas between $O_3$ $AV_d$ during weekends and weekdays. This lack of a weekend effect in $O_3$ was reported previously at the MCMA for 1987-2007 by Stephens et al. (2008), who attributed it to weekday $O_3$ production being limited by VOCs and inhibited by $NO_x$; this was also observed by Song et al. (2010). By contrast, simultaneous decreases in emissions of VOCs and $NO_x$ mostly from vehicle sources during weekends could have counteracting effects on the $O_3$ production rates, leading to similar levels of $O_3$ during weekdays at the 3 metropolitan areas. This behaviour was

reported previously by Wolff et al. (2013) for US urban areas of the Northeast, Midwest and Coastal California regions, which exhibited similar or even higher (±5 %) $O_3$ levels during weekdays than at weekends, despite lower $O_3$ precursor emissions during weekends. Moreover, Wolff et al. reported that from 1997-1999 to 2008-2010 the sites studied exhibiting a weekend effect decreased from ca. 35 % to less than 5 %, which was attributed to an increase in the VOC/$NO_x$ emission ratio derived from a greater decline in $NO_x$ than in VOCs emissions (Pusede et al., 2014).

It is likely that the $O_3$ weekly patterns observed at the metropolitan areas arise from reduced traffic activity during weekends, leading to increases in ratios of VOCs/$NO_x$. Within the MMA, this would be confirmed by lower $NO_x$ mixing ratios (on average 5 %) during weekends, changing to a transition $O_3$ production between VOC- and $NO_x$-limited during weekends. Moreover, a change to a $NO_x$-limited $O_3$ production derived from the reduction in $NO_x$ seems unlikely since this would result in lower $O_3$ levels during weekends, not observed at any of the studied urban areas (Torres-Jardon et al., 2004).". See lines: 531-542 and 544-550.

Lines 451-456: Past work (Simon et al, Cooper et al) has shown that $O_3$ trends are much more pronounced at high percentiles than at average levels, so an annual average may not be a very good metric to use to see long-term trends. *Response:* As suggested by the reviewer, we have included for the 3 metropolitan areas, the analysis of long-term trends at the annual 5[th] and 95[th] percentiles, median and averages. The observed trends are discussed with those reported in the references provided. Text modified: "Long-term trends of the annual 5[th] and 95[th] percentiles (%ile), median and average of $O_3$ during 1993-2014 were calculated using the Mann-Kendall test and Sen's estimate for the 5 sites within the MMA (Salmi et al., 2002; Carslaw and Ropkins, 2012), and are shown in Fig. 9. The long-term trends were constructed from de-seasonalised annual data derived from monthly averages filtered with STL, which were calculated from daily data of all 1-h averages, as described in Methodology (Sect. 2.3). Overall, $O_3$ shows significant increasing trends ($p<0.05$) mostly in the annual averages ranging from 0.11 ppb $O_3$ yr$^{-1}$ at SNB to 0.31 ppb $O_3$ yr$^{-1}$ at OBI, and in the 95[th] %ile, which ranged from 0.39 ppb $O_3$ yr$^{-1}$ at OBI and SNB to 0.75 ppb $O_3$ yr$^{-1}$ at SNN. The 5[th] %ile increased significant only at OBI in 0.08 ppb yr$^{-1}$, while the median increased at SNN by 0.14 ppb $O_3$ yr$^{-1}$ and at OBI by 0.23 ppb $O_3$ yr$^{-1}$. Note that if trends are segmented and considered only after the decline in 1994-1995, the only significant change is that the $O_3$ growth rate at SNN would increase to 0.31 ppb $O_3$ yr$^{-1}$ and GPE would decrease to 0.14 ppb $O_3$ yr$^{-1}$, while in the 95[th] %ile the trends would decline slightly at GPE and SNB to 0.27 ppb $O_3$ yr$^{-1}$, and at OBI to 0.42 ppb $O_3$ yr$^{-1}$. Despite exhibiting the highest $O_3$ mixing ratios within the MMA, STA did not exhibited significant trends in any of the tested metrics." See lines: 438-451.

"Long-term trends of de-seasonalised $O_3$ annual median, 5[th] and 95[th] percentiles at the 3 urban areas were determined following the same methodology as for annual averages (Fig. S10). Overall, the linear trends observed in $O_3$ annual averages for the MMA and MCMA are also seen in the other tested metrics, with significant ($p<0.05$) increases at MMA ranging from 0.05 ppb $O_3$ yr$^{-1}$ (5[th] percentile) to 0.41 ppb $O_3$ yr$^{-1}$ (95[th] percentile), and decreases at MCMA between 0.37 ppb $O_3$ yr$^{-1}$ (5[th] percentile) and 2.32 ppb $O_3$ yr$^{-1}$ (95[th] percentile). As for the $O_3$ annual averages, the GMA shows non-significant ($p>0.05$) trends in the other tested metrics. Notably, only the tropospheric CO decreased significantly ($p<0.05$) at the 3 urban areas studied, with the largest decrease rate of 0.12 ppm CO yr$^{-1}$ detected at the MCMA and the lowest one of 0.02 ppm CO yr$^{-1}$ calculated at the MMA. Thus, whereas $O_3$ precursors have decreased linearly within the MCMA and the GMA during the studied period, within the MMA those have increased during the same period despite the introduction of emission control policies (SDS, 2015)." See lines: 566-576.

**Lines 458-464:** Zheng et al and Camalier et al have analyzed the impact of inter-annual meteorological variation on $O_3$ trends. These studies should be cited and discussed.

***Response:*** As requested by the reviewer, the references provided were cited and discussed in the seasonal cycles analysis section. Briefly, since $O_3$ time-series contain a significant seasonal component as reported in the literature, several methodologies have been developed to remove it and filter the influence of meteorology when determining long-term trends. In the present study, the STL technique (Cleveland et al., 1990) was used to filter out the seasonal component from the $O_3$ data, as the seasonality accounts for the year-to-year variation caused by changes in SR, RH, temp. As described along the manuscript, all annual data used to determine long-term trends for all pollutants analysed were derived from de-seasonalised data. Therefore, it is expected that the reported trends have no significant influence of the year-to-year variations in meteorology. See lines: 374-383, 393-400 and 582-585.

**Lines 466-475:** The explanation linking $O_3$ trends to emissions trends does not follow logically and is in contrast to results presented by Duncan et al.

***Response:*** The results presented in our manuscript were revised and contrasted with the trends reported by Duncan et al. (2016). Additionally, we conducted an exhaustive revision of the data reported in the NEI and local emission inventories to verify consistency in methodologies used, which is described in section 2.2. See lines: 181-193, 195-205, 459-465, 474-479 and 572-576.

**Tables 3 and 4:** Are $O_3$ statistics based on hourly $O_3$ data or some other averaging period/daily max period. Please clarify in text and table headings.

***Response:*** As requested by the reviewer, the resolution of $O_3$ data reported in Table 3 and 4 (now Table S1 and S2) was added. See: Table S1 and Table S2. See line: 287.

**Fig 1a:** Text should describe how this figure was created from the data sources listed. Do different data sources/years use consistent methodologies?

***Response:*** The methodologies used to obtain the emission estimates are included in section 2.2. Fig. 1 was moved to Supplementary information (now Fig. S1). Fig. S1 only shows NEI emission data of VOCs and $NO_x$ as described in the caption.

**Fig 3:** The label for panel d is missing.

***Response:*** Label (d) was included in the Fig. S5.

**Figure 8:** How were 95% CIs constructed? Were they based on all daily values? Or on variation among sites in annually averaged profiles? In either case, these confidence intervals look VERY small, I think there is an error in the plotting. It is hard to believe that there would be so little day to day or site to site variability.

***Response:*** The 95 % confidence intervals shown in Fig. 12 were calculated through bootstrap re-sampling (Carslaw et al. 2015), since it provides a better estimation compared with calculations based on normal data distributions. This explains the small confidence intervals compared with those constructed using a parametric test.

References provided by the reviewer:

Camalier, L.; Cox, W.; Dolwick, P. (2007) The effects of meteorology on ozone in urban areas and their use in assessing ozone trends, Atmospheric Environment, 41, 7127-7137.

Duncan, B. N., L. N. Lamsal, A. M. Thompson, Y. Yoshida, Z. Lu, D. G. Streets, M. M. Hurwitz, and K. E. Pickering (2016), A space-based, high-resolution view of notable changes in urban NOx pollution around the world (2005–2014), J. Geophys. Res. Atmos., 121, 976–996, doi:10.1002/2015JD024121.

Zheng, J.; Swall, J.L.; Cox, W.M.; et al. (2007) Interannual variation in meteorologically adjusted ozone levels in the eastern United States: A comparison of two approaches, Atmospheric Environment, 41, 705-716.

*US O3 trends references (partial list):

Berlin, S. R.; Langford, A. O.; Estes, M.; Dong, M.; Parrish, D. D. Magnitude, decadal changes, and impact of regional background ozone transported into the greater Houston, Texas, Area. Environ. Sci. Technol. 2014, 47, 13985−13992.

Butler, T. J.; Vermeylen, F. M.; Rury, M.; Likens, G. E.; Lee, B.; Bowker, G. E.; McCluney, L. Response of ozone and nitrate to stationary source NOx emission reductions in the eastern USA. Atmos. Environ. 2011, 45 (5), 1084−1094.

Chan, E., Regional ground-level ozone trends in the context of meteorological influences across Canada and the eastern United States from 1997 to 2006. J. Geophys.Res.-Atmos. 2009, 114.

Chan, E.; Vet, R. J. Baseline levels and trends of ground level ozone in Canada and the United States. Atmos. Chem. Phys. 2010, 10(18), 8629−8647.

Cooper, O. R.; Parrish, D. D.; Stohl, A.; Trainer, M.; Nedelec, P.; Thouret, V.; Cammas, J. P.; Oltmans, S. J.; Johnson, B. J.; Tarasick, D.; Leblanc, T.; McDermid, I. S.; Jaffe,D.; Gao, R.; Stith, J.; Ryerson, T.; Aikin, K.; Campos, T.; Weinheimer, A.; Avery, M. A. Increasing springtime ozone mixing ratios in the free troposphere over western North America. Nature 2010, 463 (7279), 344−348.

Cooper, O. R.; Gao, R. S.; Tarasick, D.; Leblanc, T.; Sweeney, C., Long-term ozone trends at rural ozone monitoring sites across the United States, 1990−2010. J. Geophys. Res.-Atmos. 2012, 117.

Hidy, G.M.; Blanchard, C.L. (2015) Precursor reductions and ground-level ozone in the Continental United States, Journal of the Air & Waste Management Association, 65, 1261-1282.

Hogrefe, C.; Hao, W.; Zalewsky, E. E.; Ku, J. Y.; Lynn, B.; Rosenzweig, C.; Schultz, M. G.; Rast, S.; Newchurch, M. J.; Wang, L.; Kinney, P. L.; Sistla, G. An analysis of long-term regional-scale ozone simulations over the Northeastern United States: Variability and trends. Atmos. Chem. Phys. 2011, 11 (2), 567−582.

Lefohn, A. S.; Shadwick, D.; Oltmans, S. J. Characterizing changes in surface ozone levels in metropolitan and rural areas in the United States for 1980−2008 and 1994−2008. Atmos. Environ. 2010, 44 (39), 5199−5210.

Lin, C. Y. C.; Jacob, D. J.; Munger, J. W.; Fiore, A. M. Increasing background ozone in surface air over the United States. Geophys. Res. Lett. 2000, 27 (21), 3465−3468.

Lin, M., Horowitz, L.W., Cooper, O.R. et al (2015) Revisiting the evidence of increasing springtime ozone mixing ratios in the free troposphere over western North America, Geophysical Research Letter, 42, 8719-8728.

Parrish, D. D.; Law, K. S.; Staehelin, J.; Derwent, R.; Cooper, O. R.; Tanimoto, H.; Volz Thomas, A.; Gilge, S.; Scheel, H. E.; Steinbacher, M.; Chan, E. Long-term changes in lower tropospheric baseline ozone concentrations at northern mid-latitudes. Atmos. Chem. Phys. 2012, 12 (23), 11485−11504.

Pusede, S.E.; Steiner, A.L.; Cohen, R.C. (2015) Temperature and Recent Trends in the Chemistry of Continental Surface Ozone, Chemical Reviews, 115, 3898-3918.

Sather, M. E.; Cavender, K. Update of long-term trends analysis of ambient 8-h ozone and precursor monitoring data in the South Central U.S.; encouraging news. J. Environ. Monit. 2012, 14 (2), 666– 676.

Sather, M.E. and Cavender, K. (2016) Trends analyses of 30 years of ambient 8 hour ozone and precursor monitoring data in the South Central US: progress and challenges, Environmental Science-Processes & Impacts, 18, 819-831.

Simon, H.; Reff, A.; Wells, B.; et al. (2015) Ozone Trends Across the United States over a Period of Decreasing NOx and VOC Emissions, Environmental Science & Technology, 49, 186-195.

Souri, A.H.; Choi, Y.; Li, X.; et al. (2016) A 15-year climatology of wind pattern impacts on surface ozone in Houston, Texas, Atmospheric Research, 174, 124-134.

Strode, S.A.; Rodriguez, J.M.; Logan, J.A.; et al. (2015) Trends and variability in surface ozone over the United States. JGR-Atmospheres, 120, 9020-9042.

Tanner, R. L.; Bairai, S. T.; Mueller, S. F. (2015) Trends in concentrations of atmospheric gaseous and particulate species in rural eastern Tennessee as related to primary emission reductions, Atmospheric Chemistry and Physics, 15, 9781-9797.

U.S. EPA, Our Nation's air: Status and trends through 2015. Office of Air Quality Planning and Standards: RTP, NC, 2015, https://gispub.epa.gov/air/trendsreport/2016/

---

## Author Comment (AC2) · 20 Jan 2017

We thank the reviewers for the thorough revisions and for providing constructive comments on our manuscript, "Trends of ground-level $O_3$ in Mexico during 1993-2014: Comparison of Monterrey with Mexico City and Guadalajara". We are pleased that the editor and reviewer's perspective on addressing $O_3$ long-term trends in Mexican urban areas is in agreement with our own views on the issue. We have addressed the concerns and recommendations received, and we believe that these helped to improve significantly the quality of our manuscript. Please find below our detailed response to the comments received, which are also highlighted in red in the revised version of the manuscript, submitted along with this response.

**Reviewer #2:**

This paper by Hernandez Paniagua deals with a very important topic of ground level ozone ($O_3$) and its trend in Monterrey, Mexico (MMA) from 1993-2014. It also presents comparison of $O_3$ trend with other big metropolitan areas in Mexico namely Mexico City (MCMA) and Guadalajara (GMA). While the topic is very important for both air quality and human health, the paper needs some revisions before it is suitable for publication in ACP.

**Major Comments:**

The paper as it stands is not fully digested. This is one of the major weakness of the paper. It could be significantly concise and coherent without losing any of the messages. A lot of figures could be moved to the supplemental section. For example Figures 3, 4, 5, and to some extent 6 do not really add much to the paper. Similarly, sections describing measurements at MCMA and GMA could be moved to the supplemental section as these measurements are not really different from MMA. On the other hand, the paper will benefit by expanding the analysis section.
*Response:* As requested by the reviewer, Figures 3 and 5 were moved to Supplementary information, now Fig. S3 and S5, respectively. We consider that Figure 4 must be included in the main body of the manuscript, because it shows the wind direction occurrence discussed along the whole manuscript. Similarly, Figure 3 shows the continuous measurements of $O_3$ discussed in section 3.2, instead of moving it to Supplementary information, lines showing the 5th, median and 95th percentile as requested below, were included for better interpretation, and the trends of such metrics are discussed in Section 3.4. The Sections 2.2 and 2.3 concerning the monitoring of $O_3$ at the MCMA and the GMA were moved to Supplementary information as requested. See sections: S1.2 and S1.3.

There are also comparisons with places around the world that is not very relevant for MMA. While it is important to compare the results with measurements made at other places around the world, these locations need to be very carefully selected. $O_3$ levels depend not only on the emissions of precursors but also the availability of the sunlight. The locations selected in the paper are from everywhere in the world from Canada to Japan, Cyprus to Saudi Arabia. These places do not have similar climate and very likely different emission scenarios and as a result not suitable candidate sites for comparison. Surprisingly there were no comparison being made with any locations in United States which likely has places with similar climate and emission sources (at least in terms of vehicular fleet make up). I suggest the other make a more selective comparison.
*Response:* The reviewer makes a good point. We have modified the introduction to present only relevant studies carried in North America and Europe. See lines: 58-75, 77-92. Additionally, reports of trends in $O_3$ at US urban areas are cited and discussed along the results section to explain the $O_3$ trends reported in the current study. See lines: 114-119, 387-400, 496-502, 544-550 and 590-597.

Despite the paper claiming it as a study of "impact of emissions of VOCs and NO$_x$ on trends of ground level O$_3$", there is very little analysis of emissions of VOCs and NO$_x$ in the paper. The only analysis presented is the trend in VOC and NOx emission inventory in MMA, GMA and MCMA (Fig. 1). And, Fig. 1 is mainly used in the introduction and not interpreting the observed trends. It is misleading to claim it as the study of "impact of emissions" as it stands. I suggest the authors, present analysis of NO$_x$ and VOCs (CO) measurements and attempt to make connections between VOCs, NO$_x$ and O$_3$. A more suitable title for the current paper would be "Trends of ground level O$_3$ in Monterrey, Mexico during 1993-2014: Comparison with Mexico City and Guadalajara". The lack of NO$_x$ and VOCs trend makes most of the current conclusion seem more like speculations.

**Response:** We agree with the reviewer, data of O$_3$ precursors emissions is now presented in the introduction section. See lines: 109-122, 124-129. Additionally, data of NO$_x$ and CO recorded within the MMA were used to explain the observed trends in O$_3$. See lines: 311-319, 410-416, 457-465 and 566-574. We have also modified the title of our manuscript as suggested to: "Trends of ground-level O$_3$ in Monterrey, Mexico during 1993-2014: Comparison with Mexico City and Guadalajara".

There are too many different O$_3$ metrics being used in the paper and the authors constantly switch between them. There is no rationale being presented for why a certain metric is being used. I suggest the authors minimize the number of metric being used if possible or justify the use of different metrics in terms of what it reveals about O$_3$ in MMA.
**Response:** The reviewer is right, the importance of each metric discussed was included. See lines: 135, 307-309, 367-369 and 402-403.

**Specific Comments:**

Line 23: Why is larger AV$_d$ observed at polluted sites close to industrial areas? O$_3$ being a secondary pollutant should show larger AV$_d$ at downwind sites? In fact, the largest AV$_d$ is observed at STA which is furthest away from the industrial areas.
**Response:** The reviewer is right, there was a mistake in the description of the AV$_d$ in the abstract. Indeed, Fig. 6 clearly shows that the largest AV$_d$ occur at downwind sites, i.e. SNB and STA. Text modified: "… the largest AV$_d$ are observed at sites downwind of industrial areas.". See lines: 23-24.

Line 30: GPE is described as highly populated area downwind of an industrial area in Table 2. So, GPE qualifies as site with largest and smallest seasonal cycle (AV$_s$).
**Response:** The reviewer is right. There was mistake in the GPE site description in Table 2. Indeed, GPE is located within a densely-populated area, with few industries nearby, as shown in Fig. 1. Text modified in Table 2: "Urban background site in the La Pastora park, surrounded by a highly populated area, 450 m from Pablo Rivas Rd". The sentence concerning the AV$_s$ description in the abstract was also corrected, in accordance with the results presented in Fig. 6. Text modified: " The largest amplitudes of the seasonal cycles (AV$_s$) are typically recorded downwind of urban areas, whereas the lowest values are recorded in highly populated areas and close to industrial areas. ". See lines: 28-30.

Line 70: Introduction: The introduction switches between O$_3$ trend in rural background and urban areas. I suggest the authors focus solely on the urban areas as this is the focus of this particular paper.
**Response:** As suggested by reviewer, the introduction was modified to focus on studies of O$_3$ trends in urban areas. See lines: 70-75, 78-92, 96-102.

Line 133-135: Based on Figure 14, O$_3$ has gone up in MMA by only around 20-25% and decreased in GMA.

*Response:* We have rephrased the sentence to clarify that the exceedances mentioned are punctual. Text modified: "For instance, official reports indicate that since 2000, ground-level $O_3$ at the Guadalajara metropolitan area (GMA, the second most populated city) and the Monterrey metropolitan area (MMA, the third most populated city), has breached the 1-h average standard of 110 ppb $O_3$ by up to 80 %, and the 8-h running average standard of 80 ppb $O_3$ by up to 50 % (INE, 2011; SEMARNAT, 2015).". See lines: 98-102.

Line 172-186: This section belongs to the results and discussion section. These could be used to explain some of the observations rather than keeping it in the methodology section. Further, I suggest figures 3 and 4 be moved to a supplemental section to make the paper more concise.

*Response:* As suggested by the reviewer, the description of Air mass back-trajectories calculation was moved to the results section 3.1. Text modified:

"**3.1 Wind occurrence at the MMA**

The MMA is highly influenced by anti-cyclonic, easterly air masses that arrive from the Gulf of Mexico, especially during summer (Fig. S5). Figure 2 shows the frequency count of 1-h averages of wind direction by site and season within the MMA during 1993-2014. At all sites, apart from OBI, the predominant wind direction is clearly E, which occurs between 35-58 % of the time depending on season. These air masses are augmented by emissions from the industrial area E of the MMA, which are transported across the urban core and prevented from dispersing by the mountains located S-SW of the MMA. On average, the highest wind speeds are observed during summer. By contrast, calm winds of $\leq 0.36$ km h$^{-1}$ (0.1 m s$^{-1}$) occurred less than 2 % of the time at all sites, most frequently in winter, and least frequently in summer.".

Figure 3 was moved to Supplementary information, now Fig. S5. However, we decided to maintain Fig. 4 (now Fig. 2) within the main body of the manuscript, in order to provide information of the wind occurrence at each monitoring site, which is useful for interpreting the results presented along the manuscript.

Line 188: What is the time resolution of these measurements?

*Response:* The resolution of the $O_3$ measurements was added. Text modified: " Tropospheric $O_3$, 6 additional air pollutants (CO, NO, $NO_2$, $SO_2$, $PM_{10}$, and $PM_{2.5}$) and 7 meteorological parameters (wind speed (WS), wind direction (WD), temperature (Temp), rainfall, solar radiation (SR), relative humidity (RH) and pressure) have been monitored continuously, with data summarised as hourly averages, since November 1992 at 5 stations that form part of the Integral Environmental Monitoring System (SIMA) of the Nuevo Leon State Government (Table 2; SDS, 2016). ". See lines: 159-163.

Line 204-235: Section 2.2 and 2.3: I suggest moving these sections to the supplemental as well. Most of the analysis focuses on the MMA and these two sections only describe the locations and measurements at MCMA and GMA. These measurements are not different from MMA. A reference sentence at the end of MMA measurement section would be sufficient.

*Response:* As suggested by the reviewer, sections 2.2 and 2.3 were moved to Supplementary information S1.2-3. A reference of $O_3$ monitoring within the MCMA and the GMA was placed at the end of the paragraph. Text added: "The monitoring of $O_3$ and other air pollutants at the MCMA and the GMA is detailed in the Supplementary Information S1.2-3.". See lines: 177-178.

Line 237: Section 2.4: I suggest the author expand this section to describe all the methods used in the analysis of the data. Please add a brief description for (i) openair package for R, (ii) MAKESENS 1.0 macro, (iii) Seasonal Trend Decomposition technique. Please include how do they work or what is being done in each of these program and what are the advantages of such an analysis to reveal changes in $O_3$?

*Response:* As requested, brief descriptions of the *openair* software tools, the MAKESENS 1.0 macro and the STL technique were included in section 2.3. Analysis of data. Additionally, the pertinence of each function and test used in the current study was stated when required along the results section. See lines: 217-230, 232-242, 244-256.

Line 238: I suggest that figure 5 be moved to the supplemental section.
*Response:* As suggested, Figure 5 was moved to Supplementary information. See Fig. S6.

Line 265: It would be better to show 5, 50 and 95th percentile line for the data in Figure 6 than all the data.
*Response:* As requested, lines showing the 5th, 50th and 95th percentiles were included in Fig. 3 (before Fig. 6). However, we maintained the points representing the 1-h $O_3$ averages in order to show the variations in the magnitude of $O_3$ peaks during the studied period.

Line 270: Add "(see Figure 3)" behind "(winter)".
*Response:* The text was added: "The highest $O_3$ mixing ratios (1-h averages) are typically observed in April (spring), with lowest values usually recorded in December and January (winter) (Fig. 3).". See line: 287.

Line 276: Reaction of $O_3$ with NO to form $NO_2$ is only a part of the full Ox cycle. Please include the full Ox cycle.
*Response:* We modified the sentence as requested. See lines: 294-295.

Line 278: Ox cannot have a minimum value of 0. This would require both $NO_2$ and $O_3$ to be 0. It is very likely a measurement error or lack of measurements. I suggest the authors employ some kind of data filtering. There are also instances where Ox is lower than O3 which is again not possible.
*Response:* The reviewer is right, some $O_x$ values were calculated when $O_3$ or $NO_x$ 1-h averages were missing. Table S2 was corrected.

Line 282-285: It would be very helpful to include trends in measured NOx and CO to interpret the observed trends in O3 and Ox. This would also be in line with the title of the paper.
*Response:* As requested by the reviewer, the long-term trends in maximum daily 1-h averages for $NO_x$ were included in Figure 4. These were used to discuss the reported results when required. Text modified: "The largest annual increase observed at SNN is likely influenced by the significant ($p<0.05$) annual growth of 1.90 ppb y$^{-1}$ in $NO_x$ in levels as shown in Fig. 4, which can be ascribed to localised industrial emissions and constant urban growth W of the MMA (ProAire-AMM, 2008; SDS, 2015). By contrast, the non-significant ($p>0.05$) trend of -0.01 ppb $O_3$ yr$^{-1}$ observed at STA is may be masked by local import of $O_3$, combined with air masses stagnation, since $NO_x$ does exhibit a significant ($p<0.05$) annual increase of 1.59 ppb yr$^{-1}$. However, long-term monitoring of VOCs trends and sources is needed to determine the origin of the no trend current status at STA.". See lines: 313-319.

Line 287: Please add description of how the data is de-seasonalised.
*Response:* As requested by the reviewer, the STL filtering procedure is described in section 2.3 Analysis of data. the seasonal component. Text added: "The $O_3$ and other air pollutants time-series were decomposed into trend, seasonal and residual components using the Seasonal-Trend Decomposition technique (STL; Cleveland et al., 1990). STL consists of two recursive procedures: an inner loop nested inside an outer loop, assuming measurements of $x_i$ (independent) and $y_i$ (dependent) for i = 1 to *n*. The seasonal and trend components are updated once in each pass through the inner loop; each complete run of the inner loop consists of $n_{(i)}$ such passes. Each pass of the outer loop consists of the inner loop followed by a computation of the robustness weights, which are used in the following run of the inner

loop to minimise the influence of transient and aberrant behaviour on the trend and seasonal components. The initial pass of the outer loop is performed with all robustness weights equal to 1, followed by $n_{(0)}$ passes of the outer loop.".

Additionally, it was noted that STL was used to filter the $O_3$ data when required. See lines: 309-311, 361-362, 379-383, 403-405.

Line 289: A $NO_x$ trend would also help justify this statement regarding increased localized industrial emissions.

*Response:* As requested, we have included in Fig. 4 the long-term trend for maximum daily 1-h $NO_x$ averages for all monitoring sites, and the trends determined have been used to explain the increases in $O_3$ levels. Text modified: "The largest annual increase observed at SNN is likely influenced by the significant ($p<0.05$) annual growth of 1.90 ppb $y^{-1}$ in $NO_x$ in levels as shown in Fig. 4, which can be ascribed to localised industrial emissions and constant urban growth W of the MMA (ProAire-AMM, 2008; SDS, 2015). By contrast, the non-significant ($p>0.05$) trend of -0.01 ppb $O_3$ $yr^{-1}$ observed at STA is may be masked by local import of $O_3$, combined with air masses stagnation, since $NO_x$ does exhibit a significant ($p<0.05$) annual increase of 1.59 ppb $yr^{-1}$.". See lines: 313-318.

Line 290-292: Isn't this the further study to connect emissions with O3?

*Response:* We discussed in the manuscript that the changes in $NO_x$ levels are likely influencing the changes observed in $O_3$, however, we also stated that a further study of VOCs long-term trends would help to clarify the origin of the no trend status at STA. See lines: 313-318.

Line 298: It is surprising that new vehicles are only limited to the city center or the impact of new vehicles are only seen there. Is there some kind of restrictions on the age limit of vehicle that can enter the city center? Else you should see the benefit over the whole MMA unless the industrial emissions are offsetting the vehicular emissions.

*Response:* There is any restriction of automobile age circulation within the city centre of the MMA. We have clarified that because OBI is representative of mobile $NO_x$ sources, the growth reported in the vehicular fleet corresponds mostly to new vehicles equipped with efficient exhaust catalyst technology, and hence the decline in $NO_x$ seen at OBI (Fig. 4). Additionally, we show that the largest increase in maximum daily 1-h $NO_x$ averages is observed at the SNN industrial site. Finally, we also discussed in the manuscript that the increases in $NO_x$ at other sites are related with the constant growth in urbanisation, as can be seen in Fig. 4a for OBI. See lines: 321-330.

[Figure]

**Fig. 4.** Long-term trends of daily maximum 1-h values for $NO_x$, $O_3$ and $O_x$ observed at the 5 monitoring sites during 1993- 2014 within the MMA. The slopes show annual rates of change expressed in units of ppb yr$^{-1}$. The dashed lines represent the Sen slopes. Statistical significance is expressed as $p<0.1$ =$^+$, $p<0.05$ = *, $p<0.01$ = ** and $p<0.001$ = ***.

Line 305: Is there seasonality in rush hour or is the observed shift in O3 dip due to change in time i.e. day light saving?

***Response:*** The 1-h variation in the $O_3$ daily cycle arises from the change to daylight saving time during spring and summer, which was clarified in the text. Text modified: " Figure 5 shows daily profiles of $O_3$, $O_x$, NO, $NO_2$, $NO_x$, and SR averaged over the 5 sites within the MMA. $O_3$ generally dips during rush hour by reaction with NO, which occurs around 07:00 in spring and summer and 08:00 in autumn and winter;

the 1-h difference in the dip derives from the change to daylight saving time during spring and summer. ".  See lines: 334-336.

*Response:* We have stated how the normalisation was made. Additionally, this procedure was included also in the Figure 6 caption. Text modified: "To compare the $O_3$ diurnal cycles by season, normalised daily profiles were constructed by subtracting daily averages from hourly averages in order to remove the impact of the long-term trends (Fig. 6; Hernández-Paniagua et al., 2015), with daily amplitude values ($AV_d$; calculated by subtracting the lowest normalised values from the highest normalised values) used to assess diurnal variations in $O_3$ among seasons.". See lines: 345-349.

*Response:* The reviewer is right. The lowest $AV_d$ are observed at all sites during winter, whereas the lowest ones occur in summer. Text modified: "The lowest $AV_d$ values occur in winter at all sites in response to reduced SR, whereas the largest values observed during summer result from enhanced photochemistry under high SR. ". See lines: 349-350.

*Response:* We have clarified that the lowest annual average $AV_d$s at SNN result from the arrival of NE and E air masses laden with fresh emissions of $O_3$ precursors, which contrasts with the $AV_d$s determined at downwind sites receptors of photochemically processed air masses, particularly STA, where the largest average $AV_d$s were observed. Text modified:  "The lowest $AV_d$ values occur in winter at all sites in response to reduced SR, whereas the largest values observed during summer result from enhanced photochemistry under high SR. The lowest $AV_d$ observed at SNN is associated with the inflow of NE and E air masses laden with fresh emissions of $O_3$ precursors, which are transported to downwind sites (SNB and STA), and become stagnated by the surrounding mountains. This would explain that the largest $AV_d$s within the MMA are observed at sites receptor of photochemically processed air masses, particularly STA (Fig. 6).". See lines: 349-354.

*Response:* As requested, Fig. 8 (before Fig. 10) was modified. The annual cycle of meteorological variables reported in the literature as drivers of the $O_3$ cycles, was summarised in Fig. 8a. The statistical analysis of $O_3$ mixing ratios dependence to Temp., rain RH and SR, showed the strongest correlation for $O_3$ and SR (r=0.76, $p<0.05$). Figure 8b shows seasonal cycles for $O_3$ and SR derived from monthly averages filtered with the STL technique. Overall, the relationship between both variables is clear, with peaks for $O_3$ in spring and early autumn, and for SR in early summer. Figure 8c shows the trends in the $O_3$ $AV_s$ for all sites within the MMA, with a clear decline in $O_3$ $AV_s$ before and during the economic crisis between 1994-1996 as result of decreased emissions of $O_3$ precursors, and an increasing trend since 1998 in response to the recovery of the economy. Finally, Figure 8d shows the annual rates of change in $AV_s$ for the 5 monitoring sites within the MMA from 1993 to 2014.

[Figure]

**Fig. 8a).** Annual cycles of $O_3$, temp., rain, RH and SR constructed by averaging records from 1993 to 2014 for a 1-year period. **b).** Average seasonal cycles in $O_3$ and SR within the MMA, constructed from monthly averages filtered with the STL technique developed by Cleveland et al. (1990). **c).** Trends in $AV_s$ of $O_3$ recorded at the 5 monitoring sites within the MMA from 1993 to 2014. The decline in $AV_s$ observed is due to the economic crisis experienced in Mexico during 1994-1996, followed by persistent increases in $AV_s$ since 1998. **d).** Annual rates of change in $O_3$ $AV_s$ by site, before and after the 1994-1996 economic crisis.

Line 341: How does reduced rainfall decrease O3 levels? In line 345-346, it is mentioned that frequent rain storms suppresses O3 levels in late summer and early autumn. These two statements are contradictory.

*Response:* The reviewer is right, there was a mistake in the sentence. Lee et al. (2014) reported increases in $O_3$ levels during the rainy season due to increased concentrations of the hydroxyl radical. We have corrected the sentence and included such study. Text modified: "By contrast, downward spikes in the seasonal cycles of $O_3$ within the MMA are observed recurrently between July-August (Fig. 8b), which likely result from high wind speeds (>6 km h$^{-1}$ in average) that disperse $O_3$ precursors and increase the boundary layer height (ProAire-AMM, 2008), and high day-time temperatures (>40° C) that could suppress the $O_3$ formation. Steiner et al. (2010) reported that within VOC-limited areas, temperatures >38° C may lead to decreases in $O_3$ formation, in response to a decrease in the peroxyacetyl nitrate lifetime ($NO_x$ sink). The peak in $O_3$ observed in September is characteristic of humid regions, and can be ascribed to an increase in OH radicals derived from the increment in RH during the rainy season (Lee et al., 2014). Zheng et al. (2007) reported that this $O_3$ secondary peak became less noticeable since 2000 over the mid-western and eastern US regions. Indeed, the $O_3$ secondary peak is characteristic of the Asian summer monsoon, which transports maritime clean air to land with constant rainfall, thereby increasing RH (Xu et al., 2008).". See lines 390-400.

Lee, Y. C., Shindell, D. T., Faluvegi, G., Wenig, M., Lam, Y. F., Ning, Z., Hao, S., and Lai, C. S.: Increase of ozone concentrations, its temperature sensitivity and the precursor factor in South China, Tellus B. Chem. Phys. Meteorol., 66, doi:10.3402/tellusb.v66.23455, 2014.

Steiner, A. L., Davis, A. J., Sillman, S., Owen, R. C., Michalak, A. M., and Fiore, A. M: Observed suppression of ozone formation at extremely high temperatures due to chemical and biophysical feedbacks, Proc. Natl. Acad. Sci. U.S.A., 107, 19685-19690, doi:10.1073/pnas.1008336107, 2010.

Line 359: What is the benefit of calculating AVs. It is currently not clear. One could do a similar trend analysis without calculating AVs.

*Response:* We have stated the relevance of analysing $AV_s$. Since the $O_3$ seasonal cycle is related with the periodic component of the $O_3$ time series (Cleveland et al., 1990), an increase in the $AV_s$ may imply an increase in the mixing ratios related to seasonality. Moreover, the changes in $AV_s$ determined for the MMA, may reflect the changes in ambient levels of $O_3$ precursors between 1993-2014. The analysis of $AV_s$ presented in the current study confirms the dominant role of emissions from the industrial region over tropospheric $O_3$, since the greater changes were observed at the industrial site SNN compared to the urban site GPE. Text modified: "The seasonal amplitude value ($AV_s$) may provide insights regarding the response in $O_3$ production to year-to-year variations in the emissions of $O_3$ precursors and climate.". See lines: 402-405.

Line 364: How did NOx and CO change during the economic crisis? A large decrease in NOx was observed in US and Europe during the last recession (Castellanos et al., 2012, Russell et al., 2012). Do you see similar decrease in NOx as well?

*Response:* As suggested by the reviewer, we conducted an analysis of $NO_x$ and CO long-term trends of data recorded within the MMA and cited and discussed the provided references. Overall, significant decreases are observed for GPE, SNN and OBI of ca. 5-20 % between 1994-1996, although such decreases are discussed in detail in section 3.8. Additionally, we included Fig. S8, which shows changes in the national Gross Domestic Product (GDP) from 1993 to 2014. Overall, the GDP decreased significantly in 1995 by 7.8 % relative to 1994, during the 1994-1996 economic crisis in Mexico, and in 2009, by 6.4 % in relation to 2008 in response to the worldwide economic recession. This is in agreement with the decreases observed in the US and in some European urban areas during 2007-2009. Text added: "It is very likely that the observed decline in $O_3$ $AV_s$ is ascribed to the economic crisis experienced in Mexico during 1994-1996 (Tiwari et al., 2014; INEGI, 2016), which caused a reduction in VOCs and $NO_x$ emissions from the industrial activity as reflected in the gross domestic product in 1995 (Fig. S8). Moreover, the reported recovery of the economy since 1997 may have driven the increases in precursor

emissions leading to the observed increases in $O_3$ AV$_s$. During the global economic recession of 2008-2009, Castellanos and Boersma (2012) observed a reduction of 10-30 % in the tropospheric levels of $NO_2$ over large European urban areas, which is consistent with a faster decline of 8 ±5 % yr$^{-1}$ in the $NO_2$ column density during the same period detected by Russell et al. (2012) at US urban regions.". See lines: 425-435.

Line 376: Please see previous comments regarding NOx trend.
***Response:*** As requested, long-term trends of $NO_x$ daily maximum 1-h averages and annual averages were included and discussed in the corresponding sections, as evidence of the changes in emissions with the MMA from 1993-2014.

Line 385: Please see previous comments regarding new vehicle being limited to city centers.
***Response:*** The statement was re-phrased, clarifying that the observed decrease in $NO_x$ observed at OBI is likely due to decreased $NO_x$ emissions from mobile sources, which is offset at other sites by industrial emissions and urban growth. See lines 454-465.

Line 399: Why would accumulation or stagnation of air mass not result in an increasing trend? If more O3 and precursors are coming in from nearby places, then O3 should go up?
***Response:*** We have stated that the lack of a trend at STA is likely due to the occurrence and stagnation of photo-chemically processed air masses with high loading of $O_3$ and $NO_x$, and VOCs depleted, which is line with the increasing $NO_x$ trend observed at STA. Text modified: "The large growth rates both in $O_3$ and $NO_x$ identified at SNN are likely the result of increased emissions from a growing number of industries and sub-urban development E of the MMA. However, at OBI, the increasing positive trend in $O_3$ contrasts with the $NO_x$ decreasing trend of 0.40 ppb $NO_x$ yr$^{-1}$, which may arise from the $O_3$ production non-linear response in the VOC-sensitive MMA airshed, to increasing emissions of VOCs and decreasing $NO_x$ emissions (Sierra et al., 2013; Menchaca-Torre et al. 2015).". See lines: 457-462.

Line 400: Figure 4 does not show stagnation at STA.
***Response:*** We do not agree with this comment. Fig. 4 shows that STA is dominated by the arrival of E and SE air masses in all seasons. Moreover, with the exception of a significant increase of NW air masses occurrence in winter, low occurrence is observed from other wind directions, which is due to the influence of mountains surrounding STA, which act as natural barriers to the N and S-SW-W. Hence, this would confirm that air masses towards the S-SW-W and N would likely stagnate over STA.

Line 423: SNB has growth rate of -0.06 ppb not SNN.
***Response:*** The reviewer is right, there was a mistake. The sentence was corrected. Text modified: " Table 3 shows that significant ($p<0.05$) annual $O_3$ growth ranged from -0.05 ppb $O_3$ yr$^{-1}$ for STA and W, to 0.66 ppb $O_3$ yr$^{-1}$ for OBI and SE.". See lines: 508-509.

Line 427: Please add p value.
***Response:*** The $p$-value was added. Text modified: " By contrast, significant ($p<0.05$) decreasing trends of 0.48 ppb $O_x$ yr$^{-1}$ and 1.52 ppb $NO_x$ yr$^{-1}$ were calculated for the SW sector at OBI, whereas non-significant ($p>0.05$) trends were apparent at STA.". See lines: 513-514.

Line 431: Are there any sites upwind of the MMA industrial area? This would help interpretation of the data.
***Response:*** Although, there is a monitoring site E of SNN, it was set in 2011 and has experienced instrumentation problems since then. Therefore, the data recorded there were not used in the current study.

Line 442: Figure 1 shows largest emissions for GMA in recent years. This contradicts the statement being made about Figure 1 describing the magnitude of AVds in three cities.

**Response:** Since the main comment was the consistency of the methodology used to obtain the emission estimates in each NEI release, we included these in section 2.2 and modified Fig. 1 accordingly (now Fig. S1). Additionally, we discussed in the introduction the uncertainties reported for the NEI data, although, Fig. 13 clearly shows that largest mixing ratios of $O_3$ are observed at the MCMA and the lowest ones at the MMA, which is in agreement with the $AV_d$ reported. See section 2.2 and lines 534-542.

Line 445-448: The statement regarding weekend effect is not clear. Figure 13 does not differentiate between weekday and weekend. It is not clear whether NOx or VOCs decrease during the weekend. So, the statement regarding why no differences in O3 is observed between weekday and weekend is not appropriate.

**Response:** As requested, the discussion of the weekly cycles was re-written. Text modified: "No significant differences ($p > 0.05$) were observed at any of the metropolitan areas between $O_3$ $AV_d$ during weekends and weekdays. This lack of a weekend effect in $O_3$ was reported previously at the MCMA for 1987-2007 by Stephens et al. (2008), who attributed it to weekday $O_3$ production being limited by VOCs and inhibited by $NO_x$; this was also observed by Song et al. (2010). By contrast, simultaneous decreases in emissions of VOCs and $NO_x$ mostly from vehicle sources during weekends could have counteracting effects on the $O_3$ production rates, leading to similar levels of $O_3$ during weekdays at the 3 metropolitan areas. This behaviour was reported previously by Wolff et al. (2013) for US urban areas of the Northeast, Midwest and Coastal California regions, which exhibited similar or even higher (±5 %) $O_3$ levels during weekdays than at weekends, despite lower $O_3$ precursor emissions during weekends. Moreover, Wolff et al. reported that from 1997-1999 to 2008-2010 the sites studied exhibiting a weekend effect decreased from ca. 35 % to less than 5 %, which was attributed to an increase in the VOC/$NO_x$ emission ratio derived from a greater decline in $NO_x$ than in VOCs emissions (Pusede et al., 2014).

It is likely that the $O_3$ weekly patterns observed at the metropolitan areas arise from reduced traffic activity during weekends, leading to increases in ratios of VOCs/$NO_x$. Within the MMA, this would be confirmed by lower $NO_x$ mixing ratios (on average 5 %) during weekends, changing to a transition $O_3$ production between VOC- and $NO_x$-limited during weekends. Moreover, a change to a $NO_x$-limited $O_3$ production derived from the reduction in $NO_x$ seems unlikely since this would result in lower $O_3$ levels during weekends, not observed at any of the studied urban areas (Torres-Jardon et al., 2004).". See lines 531-549.

Line 450: Section 3.7: I suggest the authors consolidate the trend in O3 in the three metropolitan areas to the observation based only on the trend line. A statement regarding why the trend line is more appropriate than randomly choosing the start and end data to get the reduction/increase in O3 is justified.

**Response:** As suggested by the reviewer, section 3.7 was modified. Text modified: "Figure 13 shows long-term trends for these pollutants determined with the Mann-Kendall and Sen's estimate. Within the MMA, a significant ($p < 0.05$) increasing trend of 0.20 ppb $O_3$ $yr^{-1}$ is observed during 1993-2014, within the MCMA a significant ($p < 0.05$) decreasing trend of 0.71 ppb $O_3$ $yr^{-1}$ occurred during the same period, while within the GMA, a non-significant ($p > 0.05$) trend of -0.09 ppb $O_3$ $yr^{-1}$ is evident during 1996-2014. The observed trends in $O_3$ during the studied period, reflect the response to decreasing $NO_x$ (1.24 ppb $yr^{-1}$; $p < 0.05$) within the MCMA (Fig. 13a), and increasing $NO_x$ (0.28 ppb $yr^{-1}$; $p < 0.05$) within the MMA (Fig. 13c). Such changes in tropospheric $NO_x$ of 1.0 % $yr^{-1}$ within the MMA and of -1.24 % $yr^{-1}$ within the MCMA, agree with those reported by Duncan et al. (2016), in the $NO_2$ column during 2005-2014 over the MMA (0.8 % $yr^{-1}$) and MCMA (-0.1 % $yr^{-1}$). The status of no trend in $O_3$ within the GMA contrasts with the significant decrease in $NO_x$ levels (1.47 ppb $yr^{-1}$; $p < 0.05$) observed both at ground-level (-2.0 % $yr^{-1}$) and in the $NO_2$ column (-0.2 % $yr^{-1}$).". See lines: 554-564.

**Line 462: Why is there such a large variance in the annual averages?**

*Response:* We have discussed along the manuscript that changes in $O_3$ precursor emissions during the economic crisis in Mexico between 1994-1996, and the global recession in 2008-2009 may have led to decreases in $O_3$. See lines: 472-483, 617-625.

**Line 471: Figure 1 shows both VOCs and NOx are going up for MCMA not going down. So, why is $O_3$ going down with both the precursors going up?**

*Response:* Fig. 1 (now Fig. S1) was modified, and now shows that $NO_x$ emissions decreased during 1999-2008 and VOCs emissions remained constant during 2005-2008. This is in agreement with the observed trends in $O_3$, despite the uncertainties reported. Additionally, data of $NO_x$ measurements were included and discussed, which show a decreasing trend during 1993-2014. Text modified: "Within the MMA, a significant ($p<0.05$) increasing trend of 0.20 ppb $O_3$ $yr^{-1}$ is observed during 1993-2014, within the MCMA a significant ($p<0.05$) decreasing trend of 0.71 ppb $O_3$ $yr^{-1}$ occurred during the same period, while within the GMA, a non-significant ($p>0.05$) trend of -0.09 ppb $O_3$ $yr^{-1}$ is evident during 1996-2014. The observed trends in $O_3$ during the studied period, reflect the response to decreasing $NO_x$ (1.24 ppb $yr^{-1}$; $p<0.05$) within the MCMA (Fig. 13a), and increasing $NO_x$ (0.28 ppb $yr^{-1}$; $p<0.05$) within the MMA (Fig. 13c). Such changes in tropospheric $NO_x$ of 1.0 % $yr^{-1}$ within the MMA and of -1.24 % $yr^{-1}$ within the MCMA, agree with those reported by Duncan et al. (2016), in the $NO_2$ column during 2005-2014 over the MMA (0.8 % $yr^{-1}$) and MCMA (-0.1 % $yr^{-1}$). The status of no trend in $O_3$ within the GMA contrasts with the significant decrease in $NO_x$ levels (1.47 ppb $yr^{-1}$; $p<0.05$) observed both at ground-level (-2.0 % $yr^{-1}$) and in the $NO_2$ column (-0.2 % $yr^{-1}$)." See lines: 555-564, 566-576.

**Line 487: Which standard is used for Table 5, new or the old one? If it is a mixed of two then, the data is not directly comparable. I suggest using the new O3 standard for all years.**

*Response:* We clarified in the text how the number of annual exceedances was calculated. Text modified: " Such standards are applicable for whole calendar years and were not used in this study to determine the annual exceedances.". See lines 607-608.

**Line 493: It is hard to evaluate the statement without not knowing what is represented in table 5. 2012 and 2013 showed a significant reduction in number of days exceeding the standard. Then, there is a big jump in 2014. If the big jump in 2014 due to the change in the standard, then it is kind of misleading to say "recommended that more stringent emission controls are introduced in order to improve air quality within the MMA".**

*Response:* We clarified in the text that the decrease in the number of annual exceedances between 2012-2013 could be due to decreases in $NO_x$ emissions observed particularly at SNN, and possibly ascribed to decreased primary emissions from industries upwind the MMA. Therefore, a decrease of industrial emissions upwind would have a positive impact as observed in the MMA airshed. Text modified: "Between 2012-2013, the number of annual exceedances decreased at all sites, possibly ascribed to an acute deacceleration of the Mexican economy reflected in declines in ground-level $NO_x$, which is observed particularly at SNN (Fig. 10). Such decrease in primary emissions from the industries upwind the urban area may impact positively the MMA airshed, leading to the observed decreases in annual exceedances.". See lines: 617-625.

**Table 3: I suggest moving it to the supplemental section.**

*Response:* As suggested by the reviewer, Table 3 was moved to Supplementary information, now Table S1.

**Table 5: Which standard is being used to calculate these exceedances?**

*Response:* The annual exceedances of the $O_3$ 1-h and 8-h running averages were calculated using the old NOM-020-SSA1-1993, which set the maximum permitted $O_3$ levels in 110 ppb (1-h), and 80 ppb (8-

h). Additionally, in order to permit a better interpretation of the annual exceedances of the $O_3$ NOM, the results from Table 5 were depicted in Fig. 14.

[Figure]

**Fig. 14.** Annual exceedances of the $O_3$ NOM for 1-h averages (110 ppb) and 8-h running averages (80 ppb) at the 5 monitoring sites within the MMA from 1993 to 2014.

Figure 2: Please remove the wind rose plot from OBI and add predominant wind direction for each site as a single arrow for each site.

*Response:* As requested by the reviewer, Figure 2 (now Fig. 1) was modified.

[Figure]

**Fig. 1(a).** The MMA, MCMA and GMA in the national context. **(b).** Topography of the MMA and distribution of the 5 monitoring sites over the area. **(c).** The 5 monitoring sites in relation to primary and secondary motorways, industries and major residential areas. The red arrows show the predominant wind direction at each site from 1993 to 2014.

Figure 9: Legend is missing STA. There are two GPE s instead.
*Response:* The legend in Fig. 9 (now Fig. 6) was corrected.

[Figure]

**Fig. 6.** O$_3$ de-trended daily profiles by season observed within the MMA during 1993-2014. De-trended O$_3$ daily cycles were constructed by subtracting daily averages from hourly averages to remove the impact of the long-term trends.

Figure 10: Please add a zoom into one of the years.

*Response:* Figure 10 was included as a panel of Fig. 8. Additionally, instead of including a zoom in a given year, the average annual cycle for O$_3$ and meteorological variables reported to be associated with O$_3$ seasonal variations was included in Fig. 8.

[Figure]

**Fig. 8a).** Annual cycles of $O_3$, temp., rain, RH and SR constructed by averaging records from 1993 to 2014 for a 1-year period. **b).** Average seasonal cycles in $O_3$ and SR within the MMA, constructed from monthly averages filtered with the STL technique developed by Cleveland et al. (1990). **c).** Trends in $AV_s$ of $O_3$ recorded at the 5 monitoring sites within the MMA from 1993 to 2014. The decline in $AV_s$ observed is due to the economic crisis experienced at the country during 1994-1996, followed by persistent increases in $AV_s$ since 1998. **d).** Annual rates of change in $O_3$ $AV_s$ by site, before and after the economic crisis within the country.

References:
Castellanos, P. and Boersma, K. F.: Reductions in nitrogen oxides over Europe driven by environmental policy and economic recession, Scientific Reports, 2, 265, 1–7, doi:10.1038/srep00265, 2012.
Russell, A. R., Valin, L. C., and Cohen, R. C.: Trends in OMI NO2 observations over the United States: effects of emission control technology and the economic recession, Atmos. Chem. Phys., 12, 12197-12209, doi:10.5194/acp-2-12197-2012, 2012.

---

## Author Response (AR2)

We thank the Co-Editor for the thorough revision and for providing constructive comments on our manuscript, "**Observed trends in ground-level O$_3$ in Monterrey, Mexico during 1993-2014: Comparison with Mexico City and Guadalajara".** We are pleased that the Co-Editor editor's perspective on addressing O$_3$ long-term trends in Mexican urban areas is in agreement with our own views on the issue. We have addressed the concerns and recommendations received, and we believe that these helped to improve significantly the quality of our manuscript. Please find below our detailed response to the comments received, which are also highlighted in yellow in the revised version of the manuscript, submitted along with this response. Please note that sections 3 and 4 have few marks since there were included during the last revision as requested by the Co-Editor.

**Co-Editor Decision: Reconsider after major revisions** (19 Feb 2017) by Sally E. Pusede Comments to the Author: Review Hernandez Paniagua et al.

I appreciate that the authors have responded to the comments of the referees, but I have some major concerns that need to be addressed prior to publication.

**1.      My primary issue** is the paper lacks focus and, after reading, I am not sure what the authors were trying to communicate scientifically about Ox trends in the urban areas in Mexico studied. Instead, the paper is largely descriptive, presenting many trends, but without clear purpose. It is insufficient for publication in ACP to describe a wide array of measurements and then speculate on the causes of the observed patterns. While there are few studies on O$_x$ in Mexico, too much is known about urban O$_x$ chemistry generally for pages of text to describe O$_x$ seasonal trends and diurnal patterns. If the O$_x$ AVd metric is being interpreted as a proxy for O$_3$ production, then say this explicitly. Trends in O$_3$ AVd may simply indicate changes in NO$_2$.

**Response:** We highlight that our primary aim is to address the effect of the few controls introduced to control precursor emissions on ground-level O$_3$ long-term trends and on the air quality within the MMA. Second, we also investigate if the strategies designed to control O$_3$ precursor emissions within the MCMA, which have been introduced at the GMA and at the MMA have resulted in improvements in terms of decreasing the O$_3$ levels. We have clarified that very different air quality control strategies have been considered at each city studied here, which has led to different air pollution scenarios at each one. For example, while the Mexico City has been subject of numerous measures focused on emissions from on-road, point and area emissions, the measures implemented at Monterrey has been focused mostly on on-road sources with less consideration to other emissions sources, which has occurred also at the GMA. Furthermore, although on-road sources are reported to contribute with more than 50 % of total primary emissions at Monterrey, the location of the largest industrial area upwind the urban core has caused that growing industrial emissions offset the reductions in emissions from on road sources.

To estimate the absolute changes in net O$_3$ production, first, we describe how primary emissions impact the net O$_3$ production at each monitoring site at different time-scales. We use daily and annual cycles to interpret how the levels of O$_3$ and O$_X$ vary with the levels of NO$_X$, and interpret the long-term trends as the response to changes in precursor emissions. We also compared the trends for O$_3$ precursors derived from available ground-based measurements at each city, with those determined from emission estimates, showing that significant improvements in such estimates are required to better inform the current air quality policies. Finally, we revised the number of O$_3$ annual exceedances to the O$_3$ official standards, and show that if primary emissions maintain the current trends, the number of annual exceedances to the official standards will very likely increase.

**2.    Second,** the Introduction states that the authors are concerned with $O_x$ trends based on health-based metrics (page 4, line 135). It is not obvious to me then that a discussion of wintertime (and possibly springtime) trends is even warranted.

**Response:** This statement was requested by a referee during the last revision, however, after the major changes have been made it has been deleted. We have clarified that our main objective is the assessment of trends in $O_3$ and $O_X$ in response to changes in precursor emissions. The compliance of the Mexican standards for $O_3$ is carried out to show that if $O_3$ levels continue increasing more exceedances will occur.

**3.    Additionally,** annual average assessments obscure summertime differences, which, if concerned with high O3 (health-based metrics), should be the focus. To this aim, I see the presentation of trends in annual average Ox and AVd to be a distraction, as they may track wintertime changes (generally always VOC-limited chemistry), rather than changes in summertime O3 production (potentially VOC or NOx limited). Consideration of the 95th percentile O3 trends is likely a de facto summertime trend, but this is never stated nor are the data analyzed in this context.

**Response:** We have separated the analysis of $O_3$ long-term trends, considering now seasonal trends as reported by Parrish et al. (2009), and discussed the results accordingly. Briefly, we show that the significant trends observed are consistent with the transport of emissions from the industrial area, enhanced during spring and summer.

**4.    I recommend** the authors provide the reader with a focused statement of purpose and then only present observations relevant to this task.

**Response:** As requested, the objectives description paragraph of our study were modified accordingly Additionally, a paragraph describing the paper organisation was also included. Text modified: " To our knowledge, no previous study has address trends in $O_3$ and odd oxygen in urban areas of Mexico. In this study, we describe trends in ground-level $O_3$ within the MMA, and its response to changes in precursor emissions during 1993-2014. Long-term and high-frequency measurements of $O_3$ were recorded at 5 air quality monitoring stations evenly distributed within the MMA. In order to better assess photo-chemical production of $O_3$, odd oxygen defined as ($[O_X] = [O_3] + [NO_2]$) was also considered, as $O_3$ and $NO_2$ are rapidly interconverted. Diurnal and annual cycles of $O_3$ and $O_X$ are used to interpret net $O_3$ production within the MMA. We show that air mass origin influences strongly the $O_3$ annual growth rates. The trends in $O_3$, $O_X$ and precursor emissions are compared with those observed within the MCMA and GMA. Finally, we describe that NEI emission estimates for $NO_X$ and VOCs disagree in the trend magnitudes with ground-based $NO_X$ and VOCs measurements made at the urban areas studied here.

This paper is organised as follows: Section 2 presents the data quality and methodology used to derived the different trends presented. Section 3 describes in detail the $O_3$ and $O_X$ diurnal and annual cycles, and, annual and seasonally averaged trends. Section 4 discusses the origin of the $O_3$ and $O_X$ diurnal variations and trends in the light of changes in precursor emissions. Finally, Section 5 provides some conclusions regarding the trends observed at the studied urban areas.".

**5.** **For me,** an inter-annual high-Ox trend analysis would be an incredibly valuable contribution to the literature.

**Response:** As requested, we have also included the analysis of seasonal high $O_3$ trends. Briefly, we show that if $O_3$ precursor emissions continue increasing the daily maxima will likely increase and that the number of annual exceedances to the $O_3$ official standard will also increase.

**6.** **I also recommend** the authors present separate Results and Discussion sections. This would ensure the paper escapes the trap of presenting an observation and then immediately speculating on the cause. For example, this is done three times in the paragraph on page 9, lines 321–330, but is ubiquitous throughout the manuscript.

**Response:** As requested, we now present results and discussion in separate sections. See sections 3 and 4.

**Minor comments:**

**Abstract, line 15.** "In developed countries, long-term trends in $O_3$ have been studied extensively." I would not say this is true.

**Response:** We rephrased the sentence to clarify that high $O_3$ levels have been an historic problem at large Mexican urban areas. Text modified: "The largest urban areas in Mexico have experienced historically high ambient $O_3$ levels." See line: 17.

**Page 2, line 45.** "being VOC-limited" should say "being called VOC-limited."

**Response:** The statement was modified. Text modified: " The system of $O_3$ production is not linear, and is termed $NO_X$-limited, when $O_3$ production increases in response to increasing $NO_X$ emissions, and termed VOC-limited when it responds positively to emissions of VOCs (Monks et al., 2015; Pusede et al., 2015).". See lines: 45-47.

**Page 2, line 55.** The reaction between NO and $O_3$ is not an $O_3$ loss process, as $NO_2$ immediately photolyzes to yield $O_3$. At night $NO_2$ simply stores $O_3$.

**Response:** As requested, the sentence was modified. Text modified: " By contrast, the main removal processes for tropospheric $O_3$ are photochemical loss and dry deposition (Atkinson, 2000; Jenkin and Clemitshaw, 2000).". See lines: 55-56.

**Page 2, line 64–75.** Discussion of trends in background and annual $O_3$ is irrelevant to this analysis. Remove from Introduction.

**Response:** The lines describing trends in annual background $O_3$ were removed as requested.

**Page 3, lines 81–82.** Are you quoting trends in emission inventories or in actual trends? Make this clear. Should also say that there is variability between cities.

**Response:** We clarified that the decrease in $NO_X$ and VOCs emissions quoted was derived from emissions estimates, and also that this decline was estimated at national scale despite the variability from city-to-city. Text modified: "Emission estimates suggest an overall national scale decrease during 1980-2008 in US $NO_X$ and VOCs emissions of 40 % and 47 %, respectively, with city-to-city variability (EPA, 2009; Xing et al., 2013). " See lines: 75-76.

**Page 3, line 96.** A decrease of 33% in what?

**Response:** We clarified that the 33 % decrease described corresponds to that in $O_3$ annual averages. Text modified: "…with reports of a decrease in $O_3$ annual averages of ca. 33 % during the last two decades" See lines: 90-91.

**Page 3, line 96.** Delete "By contrast."

**Response:** As requested, "By contrast," was deleted. Text modified: "$O_3$ has received less consideration at other large metropolitan areas, where Mexican air quality standards are frequently exceeded (Table 1)." See lines: 91-93.

**Page 3, line 97. Delete "relatively."**

**Response:** As requested, "relatively" was deleted. See previous comment. See line: 92.

**Page 3, lines 100–101.** Not clear what this means: "has breached the 1-h average standard of 110 ppb $O_3$ by up to 80 %."

**Response:** The sentence was rephrased to clarify that the $O_3$ mixing ratios recorded at the GMA and at the MMA have exceeded the $O_3$ official standards by more than 50 %. Text modified: "Indeed, since 2000, recorded $O_3$ mixing ratios have exceeded Mexican official standards for $O_3$ 1-h average (110 ppb) and 8-h running average (80 ppb) by more than 50 % at the Guadalajara metropolitan area (GMA, the second most populated city) and at the Monterrey metropolitan area (MMA, the third most populated city (INE, 2011; SEMARNAT, 2015)." See lines: 93-96.

**Page 3, lines 104–106.** Critical without point. Remove or rephrase.

**Response:** The sentence was rephrased to clarify that the ordinary linear regression used by Benítez-García et al. (2014) is not suitable for determining long-term trends, because this can be biased by the presence of extreme data. Text modified: "However, it should be noted that the ordinary linear regression analysis used by Benítez-García et al. (2014) may be biased by extreme values and is therefore not suitable to determine $O_3$ long-term trends with significant confidence." See lines: 98-101.

**Page 3, line 109.** What is meant by "data?"

**Response:** The sentence was rephrased to clarify that the implemented initiatives have been designed on the basis of emission estimates reported in the NEI. Text modified: "The NEI suggest that from 1999 to 2008, anthropogenic $NO_X$ emissions decreased at the MCMA by 3.8 % $yr^{-1}$, but increased at the GMA and the MMA by 1.9 % $yr^{-1}$, and by 4.0 % $yr^{-1}$, respectively (Fig. S1) (SEMARNAT, 2006, 2011, 2014)." See lines: 105-107.

**Page 4, lines 137–140.** This statement is problematic: "The data sets contain features representative of industrial, urban-background and urban monitoring sites, which allow assessment of $O_3$ trends and dynamics, pollutant emissions and their contribution to the atmospheric composition depending on local meteorology and air mass transport." Really the data allow you to - assess ozone trends in locations with mixed sources and variable meteorology.

**Response:** We modified the objectives paragraph and removed the sentence regarding the dataset description. Text modified: "To our knowledge, no previous study has address trends in $O_3$ and odd oxygen in urban areas of Mexico. In this study, we describe trends in ground-level $O_3$ within the MMA, and its response to changes in precursor emissions during 1993-2014. Long-term and high-frequency measurements of $O_3$ were recorded at 5

air quality monitoring stations evenly distributed within the MMA. In order to better assess photo-chemical production of $O_3$, odd oxygen defined as ($[O_X] = [O_3] + [NO_2]$) was also considered, as $O_3$ and $NO_2$ are rapidly interconverted. Diurnal and annual cycles of $O_3$ and $O_X$ are used to interpret net $O_3$ production within the MMA. We show that air mass origin influences strongly the $O_3$ annual growth rates. The trends in $O_3$, $O_X$ and precursor emissions are compared with those observed within the MCMA and GMA. Finally, we describe that NEI emission estimates for $NO_X$ and VOCs disagree in the trend magnitudes with ground-based $NO_X$ and VOCs measurements made at the urban areas studied here" See lines: 130-139.

**Page 4, lines 141–142.** Change "oxidants" to "odd oxygen." A clearer way to say - $O_x$ include $O_3$ stored as $NO_2$.
**Response:** As requested: total oxidants as replaced with "odd oxygen". See line: 134.

**Page 4, lines 142–143.** Sentence is not meaningful.
**Response:** We modified the paragraph describing the aims of this study to make clear that $O_3$ diurnal and annual cycles are interpreted as proxy of net $O_3$ production. Text modified: "Diurnal and annual cycles of $O_3$ and $O_X$ are used to interpret net $O_3$ production within the MMA." See lines: 135-136.

**Page 6, line 207.** This subtitle should not be "Analysis of data," which implies scientific consideration of the data.
**Response:** The subtitle was changed to "Analytical methods". See line: 207.

**Page 9, lines 307–309.** Irrelevant, delete: "A study conducted among asthmatic children resident in the MCMA revealed an increase in coughing and wheezing rates, associated with cumulative exposure to high 1-h averages mixing ratios of $O_3$ and NO (Escamilla-Nuñez et al., 2008)."
**Response:** As requested, the sentence was deleted and the whole section split into Results and Discussion sections.

**Page 9, 313–314.** Speculation at best, "likely influenced by the significant ($p<0.05$) annual growth of 1.90 ppb y-1 in NO in levels as shown in Fig. 4." Best to have a separate Results and Discussion section.
**Response:** As requested, Results and Discussion are presented in separate sections. See sections 3 and 4.

**Page 9, 322–323.** Speculation - "which arise either from an increment in $NO_x$ or $O_3$ levels as shown in Fig. 4." Either give evidence for this statement or delete.
**Response:** We provide evidence to confirm that the increment in daily maximum 1-h $O_3$ arises from increases in $NO_X$. See lines: 483-493.

**Page 9, 324–325.** Speculative: - "$O_x$ trend is likely due to the decreasing levels of NOx." Give evidence or delete.
**Response:** We provide evidence to confirm that at OBI, $O_X$ has decreased in response to decreases in $NO_X$ emissions from the on-road sources, which only can be appreciated at OBI, since at the rest of the sites, $NO_X$ industrial emissions offset those reductions. See lines: See section 3.6 and 4.4.

**Page 9, 328–330.** Speculation - "This could be due to the arrival at OBI and at STA of chemically processed air masses with decreased VOC/NOx ratios, compared with those arriving at SNN loaded with fresh emissions from the nearby industrial area."
**Response:** See the previous comment. See section 4.4 and Fig. S10.

**Page 14, line 519.** Need to verify there is also no weekend effect in Ox.

**Response:** The existence of weekend effect in $O_X$ was tested for all urban areas, and shown in Fig. 8. We describe that no significant differences were observed both in $O_3$ and $O_X$ in each city. See lines: 395-411.

[revised manuscript text omitted]

---

## Author Response (AR3)

Dear Dr. Pusede

Thank you for providing the last comments on our manuscript, "**Observed trends in ground-level O$_3$ in Monterrey, Mexico during 1993-2014: Comparison with Mexico City and Guadalajara**".

We also thank to the helpful comments of all the referees during the revision process. We hope that you will now consider it suitable for publication at ACP.

Please find below our responses to your comments.

**1. The abstract is not well focused.** It should be streamlined, conveying just the key summary information.

**Response:** As requested, the abstract has been streamlined to convey only a summary of results. Text modified: "Here, we present an assessment of long-term trends in O$_3$ and odd oxygen (O$_3$ + NO$_2$) at the industrial Monterrey metropolitan area (MMA) in NE Mexico. Diurnal amplitudes in O$_X$ (AV$_d$) are used as a proxy for net O$_3$ production, which is influenced by the NO$_2$ photolysis rate. No significant differences in the AV$_d$ are observed between weekends and weekdays, although the largest AV$_d$ are observed at sites downwind of industrial areas. The highest O$_3$ mixing ratios are observed in spring, with minimum values in winter. The largest annual variations in O$_3$ are typically observed downwind of the MMA, with the lowest variations generally recorded in highly populated areas and close to industrial areas. A wind sector analysis of mixing ratios of O$_3$ precursors revealed that the dominant sources of emissions are located in the industrial regions within the MMA and surrounding area. Significant increasing trends in O$_3$ in spring, summer and autumn are observed depending on site location, with trends in annual averages ranging between 0.19 and 0.33 ppb yr$^{-1}$. Overall, during 1993 to 2014, within the MMA, O$_3$ has increased at an average rate of 0.22 ppb yr$^{-1}$ ($p<0.01$), which is in marked contrast with the decline of 1.15 ppb yr$^{-1}$ ($p<0.001$) observed in the Mexico City metropolitan area (MCMA) for the same period. No clear trend is observed during 1996 to 2014 within the Guadalajara metropolitan area (GMA)." See lines: 17-30.

**2. Line 56:** chemical loss, not photochemical loss.

**Response:** Sentence has been modified as requested. See line: 46.

**3. Line 136:** In sentence - "We show that air mass origin influences strongly the O$_3$ annual growth rates." It is not clear what "growth rates" means.

**Response:** We have replaced growth rates for increases. See line: 126.

**4. Line 285:** Revise - "photochemical season."

**Response:** As requested, photochemical season has been deleted.

**5. Line 306:** Delete sentence - "Diurnal variations in O$_3$ arise from the balance between its net production and destruction."
**Response:** As requested, the sentence has been deleted.

**6. Line 320:** Delete sentence - "O$_3$ and O$_X$ levels depend strongly on the photochemical processing of NO$_X$ and VOCs emissions."

**Response:** As requested, the sentence has been deleted.

7. $AV_d$ $O_x$ may be a proxy for $PO_3$, but not $AV_d$ $O_3$.

**Response:** The co-editor is right. We have rephrased the paragraph to describe that only $AV_d$s can be used as a proxy to assess net $O_3$ production. Text modified: "$O_x$ amplitude values ($AV_d$) derived from normalised daily cycles were used as a proxy to assess differences in the net $O_3$ production from site-to-site within the MMA. The normalised daily cycles were constructed by subtracting daily averages from hourly averages. Figure 4 shows normalised $O_x$ daily cycles. The lowest $AV_d$s in $O_x$ occur in winter consistent with reduced SR and low photolysis rates, with the largest values observed in summer. It is clear that during the year, the largest $AV_d$s are recorded at sites downwind of industrial emission sources, in particular at STA, while the lowest $AV_d$s are observed at sites upwind. The larger $AV_d$s at downwind sites are interpreted to indicate higher net $O_3$ production, derived from the occurrence of photochemical processed air masses from the E sector. The $AV_d$s at upwind sites are less affected by emissions from the MMA, and especially the industrial area." See lines: 308-316.

**8. The authors are not convincing that** "seasonal amplitude values ($AV_s$) provide insight into inter-annual variations in the net $O_3$ production in response to changes in precursor emissions and meteorology." Low $O_3$ in winter may be simply $O_3$ loss to NO, not low $PO_3$. This would be consistent with the later discussion of substantial $NO_x$ emission reductions in 1994-1996.

**Response:** We have rephrased the sentence to describe that $AV_s$ provide insights of inter-annual variations in $O_3$. Text modified: "The seasonal amplitude value ($AV_s$) provide insight into inter-annual variations in net $O_3$ production in response to changes in precursor emissions, meteorology, and $O_3$ chemistry."  See lines: 329-330.

**9. Line 413:** "Stephens et al. (2008) suggested that the most plausible explanation for the lack of weekend $O_3$ effect at MCMA during 1987-2007, is that weekday $O_3$ production is limited by VOCs and inhibited by $NO_x$." As described, this would cause weekday-weekend $O_3$ differences.

**Response:** The paragraph has been rephrased, stating that the most plausible explanation is the simultaneous decrease in both $NO_x$ and VOCs emissions during weekends, since the sole decrease in $NO_x$ emissions under VOC-limited conditions would lead to an increase in $O_3$, which is not observed. Text modified: "Stephens et al. (2008) suggested that the most plausible explanation for the lack of weekend $O_3$ effect at MCMA during 1987-2007 is a simultaneous decrease in $NO_x$ and VOCs emissions during weekends, since the sole decrease in $NO_x$ emissions under VOC-limited conditions would lead to an increase in $O_3$ not observed. Similarly, a VOC-limited $O_3$ production regime was reported for the MMA by Sierra et al. (2013), whereas Kanda et al. (2016) reported that at the GMA the $O_3$ production lies in the region between VOC- and $NO_x$-sensitivity. Therefore, it can be suggested that simultaneous decreases in $NO_x$ and VOCs emissions during weekends at the GMA and MMA explain the similar behaviour in $O_3$ and $O_x$ as at the MCMA. Moreover, a change to a $NO_x$-limited $O_3$ production regime during weekends at the three urban areas seems unlikely, since this would result in lower $O_3$ levels during weekends, which is not observed at any of the studied urban areas (Torres-Jardon et al., 2009)." See lines: 399-408.

**10. Figure 9:** Colors for $O_x$ and $O_3$ in MCMA and MMA are difficult to distinguish.

**Response:** As requested, Fig. 9 has been modified.

**11. Lines 610-618:** Delete. No need to compare to cities globally, keep the focus on cities in Mexico.

**Response:** Deleted, it was there to provide a wider context.

**12. Lines 643-646:** Delete - not relevant.

**Response:** Deleted.

**13. Lines 648-649:** Delete.

**Response:** Deleted.

**14. Line 659:** Replace "It is clear that" with "It has been shown that" and add citations.

**Response:** The sentence has been rephrased and citations have been added. Text modified: It has been shown that $O_3$ and $O_X$ decreases within the MCMA have been driven by reductions in $NO_X$ and VOCs emissions, and that the implemented strategies described in Sect. 4.1 have proved to be effective in controlling primary emissions (ProAire-MCMA, 2011; Jaimes-Palomera et al., 2016)." See lines: 629-631.

**15. Can the authors recommend NOx versus VOC control?** Or do the authors advocate for both and why?

**Response:** We have included a statement suggesting VOCs reductions alone, since reductions in $NO_X$ may increase $O_3$. Text modified: "
[revised manuscript text omitted]